# Selective weakening of population-coupled synaptic activity in vivo in a mouse model of amyloid-beta pathology

Leire Melgosa-Ecenarro [1,6], Carola I. Radulescu [1,6], Nazanin Doostdar[1,6], Joe Airey[1], Francesca A. Chaloner [1], Nawal Zabouri[2], Giada Pedretti[1], Francesca Osso[1], Leire Garrido Perez [1], Kjara S. Pilch [1], Xingjian Wang[1], Anna Mallach [1], Sadra Sadeh[3,4], Johanna Jackson [1], Paul M. Matthews [1,5] & Samuel J. Barnes [1] ✉

Synaptic dysfunction in Alzheimer's disease (AD) may drive synapse loss and cognitive impairment. Whether AD-related synaptic pathophysiology occurs globally, or in specific synapses, is unclear. We investigate in vivo AD-related synaptic dysfunction during early-stage amyloidosis in $App^{NL-G-F}$ mice. We find reduced presynaptic GABAergic proteins at c-Fos-positive excitatory neurons and increased calcium-mediated activity at excitatory and inhibitory neuronal assemblies. In vivo synaptic structure/function imaging finds reduced density and calcium-mediated activity of GABAergic axonal boutons. Rather than occurring globally, reduced synaptic activity is focused at GABAergic boutons strongly coupled to population activity in the amyloid microenvironment. The selective weakening of population-coupled synaptic activity also occurs in excitatory dendritic spines. Spatial transcriptomics finds parvalbumin-positive inhibitory neurons show differential gene expression associated with down-regulated GABAergic synaptic transmission at early stages. We propose that early-stage AD-related synaptic pathophysiology is focused at population-coupled synapses, with molecular measures implicating abnormal synaptic processing as an early-stage feature in parvalbumin-positive interneurons.

Synapse loss is a pathological correlate of impaired cognition in AD[1]. However, post-mortem brain tissue from AD patients suggests that only ~30–40% of synapses are lost across the course of the disease[2,3]. As such, synapse loss could be considered a late-stage endpoint of synaptic vulnerability, which may be preceded by earlier phases of synaptic dysfunction[1]. AD-related synaptic pathophysiology could either be a global phenomenon, impacting all synapses in the local amyloid microenvironment, or preferentially impact certain subsets of synapses[4]. For instance, synapses at discrete subsets of neurons with a specific molecular composition may be most vulnerable to synaptic pathophysiology. Alternatively, vulnerability may exist across a range of molecularly related, or independent, neuronal cell types. If synaptic pathophysiology is focused at a subset of molecularly similar and functionally specialised neurons, this may result in specific forms of circuit-related disruption (e.g., impaired feedback inhibition)[5]. However, synaptic vulnerability may also be specific to the complex

[1]UK Dementia Research Institute Centre, Department of Brain Sciences, Imperial College London, Hammersmith Hospital Campus, London, UK. [2]Department of Biomedical Engineering, Imperial College London, South Kensington Campus, London, UK. [3]Centre for Developmental Neurobiology, King's College London, New Hunt's House, Guy's Campus, London, UK. [4]Francis Crick Institute, London, UK. [5]The Rosalind Franklin Institute, Harwell Science and Innovation Campus, Didcot, Oxon, UK. [6]These authors contributed equally: Leire Melgosa-Ecenarro, Carola I. Radulescu, Nazanin Doostdar. ✉e-mail: samuel.barnes@imperial.ac.uk

molecular diversity of the synapse itself, rather than its home neuron[4]. Therefore, understanding the early-stage functional activity and molecular profile of synaptic populations that survive disease progression, as well as those that are lost, could drive targeted interventions to modify early-stage synaptic dysfunction.

Functional measures of synaptic pathophysiology are typically estimated from somatic electrophysiological recordings, which have reported mixed findings[6]. A highly reductionist summary points to a general disruption in the balance of excitatory and inhibitory synaptic activity with AD-related pathology[7–9]. However, this is an over-simplification, which misses many of the important mechanistic and cell-type-specific changes that are likely to have consequences for circuit function and the design of future therapeutic interventions. Studies of AD-related neuronal and micro-circuit pathophysiology, which can be used to approximate synaptic pathophysiology, have suggested that early-stage hyperactivity[10–12] is later followed by hypoactivity[13,14]. In addition, studies with AD patients have found evidence for subclinical epileptiform activity associated with cognitive impairment[15–17]. Paradoxically, while anti-seizure medications (ASMs) such as levetiracetam alleviate some of these deficits[15–18], other ASMs aggravate the condition[15,16,19]. Therefore, while early-stage amyloid-related changes have been associated with neuronal hyperactivity[20–23], it is unclear how this relates to the underlying synapse dysfunction and loss. Early work reported GABAergic circuitry to be relatively resilient to amyloid-beta challenge[24,25], so that major synaptic changes were focused in excitatory neurons[26–28]. However, more recent investigations have reported compelling evidence for GABAergic changes associated with neuronal hyperactivity in early stages of AD[29–31]. Micro-circuit investigations in preclinical amyloid models have implicated impaired glutamate re-uptake[27], astrocytic changes[32], modified GABAergic tone[9,19,33–35], and/or interactions between GABA-receptors and amyloid[36] as players in AD-related neuronal activity dysfunction. However, few studies have directly investigated subcellular single-synapse pathophysiology and its association with vulnerable/resilient cellular populations in the in vivo amyloid microenvironment[37,38]. In fact, AD-related cell-type vulnerability, particularly in the context of synaptic pathophysiology, remains poorly understood.

Here, we investigated synaptic pathophysiology in early-stage amyloidosis to assess the degree to which it occurs globally or in specific subsets of synapses. We first used mesoscopic calcium imaging to determine the spatiotemporal onset of dysregulated neuronal activity in the superficial cortex during early-stage amyloidosis in the $App^{NL-G-F}$ mouse model. We found a broad mesoscopic circuit, centred on posterior cortical regions, exhibits early-stage increases in the resting-state activity of excitatory neurons and abnormal oscillatory activity measured with electroencephalography (EEG). In vivo 2-photon cellular imaging revealed increased resting-state activity at excitatory (E) and inhibitory (I) neurons, with follow-up measures implicating both parvalbumin- (PV) and somatostatin-positive (SST) interneurons. These changes occur with a functional uncoupling of E-I assemblies[39], measured as a reduction in the number of E-E and E-I neuron pairs showing correlated activity. Measures of glutamatergic and GABAergic synaptic markers showed evidence for a loss of pre-synaptic GABAergic puncta, which is spatially focused at c-Fos-positive (c-Fos +) excitatory neurons. We then tested for synaptic pathophysiology using in vivo 2-photon functional imaging of both excitatory and inhibitory synaptic compartments in the amyloid microenvironment. In regions with dysregulated mesoscopic-level activity, we found a functional weakening of GABAergic axonal boutons, while excitatory dendritic spines showed only a modest destabilisation of activity levels. Weakening was greatest for subsets of GABAergic boutons and dendritic spines proximal to plaques and strongly coupled to population activity. In contrast, activity levels in GABAergic boutons and dendritic spines with low levels of population coupling were similar to wild-type (WT) animals, even if proximal to plaques. We used spatial transcriptomics to investigate cell-type-specific molecular processes associated with GABAergic synaptic pathophysiology in $App^{NL-G-F}$ mice. Of the tested GABAergic subclasses, PV interneurons showed the earliest and proportionally greatest differential gene expression. These differentially expressed genes (DEGs) were associated with GABAergic synaptic transmission and the synaptic vesicle cycle. In summary, our results suggest that early-stage synaptic pathophysiology in cortical regions with dysregulated network activity is associated with a selective vulnerability of strongly population-coupled inhibitory presynaptic inputs, with molecular measures implicating abnormal synaptic processing as an early-stage feature in PV interneurons.

## Results
### Dysregulated resting-state activity in a mouse model of amyloidosis
To test for early stages of AD-related synaptic pathophysiology, we investigated periods of dysregulated neuronal circuit activity, as this has been proposed to be an early-stage tipping point from healthy to aberrant network function in AD[15,18,40,41]. We first tested the spatiotemporal emergence of amyloid-associated activity dysregulation using calcium imaging approaches (Fig. 1a–g, Supplementary Fig. S1a–h) in a second-generation knock-in mouse model of amyloidosis ($App^{NL-G-F}$ mouse)[42]. We studied mice at 3-4 and 6-8 months (m) of age for two reasons. First, in vivo methoxy-X04 measures of amyloid plaque load found this to be a period of increasing amyloidosis in $App^{NL-G-F}$ mice (Supplementary Fig. S1i). Second, cognitive impairments have been reported to emerge between these two timepoints in homozygous $App^{NL-G-F}$ mice[43]. We crossed $App^{NL-G-F}$ mice with mice expressing GCaMP6s under the $Thy1$ promoter and used 1-photon mesoscopic imaging to measure calcium-mediated neuronal activity across the dorsal surface of the cortex in anaesthetised animals[44] (see Methods, Fig. 1a, b). We then performed a coarse parcellation to group functionally related regions with high interconnectivity[44,45]: visual/parietal (VIS), retrosplenial (RSC), somatosensory (SS), and motor (MO) cortices (Fig. 1c). Within those regions, we measured the activity (transient amplitude x transient frequency of the calcium signal) at the animal level during resting state (collected when mice were at rest in the dark[46]) (see Methods, Fig. 1d–g and Supplementary Fig. S1a–h). Using this approach, we found resting-state activity to be elevated in a broad mesoscopic circuit centred on posterior cortices in $App^{NL-G-F}$ x $Thy-1-GCaMP6s$ mice (Fig. 1d–g). Early changes in activity at 3-4 m were observed in VIS (Fig. 1d) and RSC (Fig. 1e) and were driven by changes in the frequency (Supplementary Fig. S1a, b) rather than amplitude (Supplementary Fig. S1e, f) of mesoscopic calcium events. Increases in frequency were also observed in SS, as well as in MO at later stages (Supplementary Fig. S1c, d, g, h). However, these changes were not sufficient to produce a significant rise in overall activity levels (Fig. 1f, g). Using immunofluorescence measures, we also observed that posterior regions (VIS), which exhibit earlier signs of hyperactivity in $App^{NL-G-F}$ mice, showed increased markers of both putative neuronal activity (c-Fos) and amyloidosis (methoxy-X04 and MOAB-2) in comparison to frontal regions (MO) (Supplementary Fig. S1j–m). Together, our data suggest that the most pronounced cortical activity changes in $App^{NL-G-F}$ mice occur in a broad posterior area.

### Functional uncoupling of excitatory and inhibitory assemblies in $App^{NL-G-F}$ mice
Elevated resting-state activity in excitatory neurons (Fig. 1d–g) may be driven by reductions in inhibitory neuronal activity[9]. To better understand the cellular origins of the activity changes we observed with mesoscopic calcium imaging, we made in vivo 2-photon calcium imaging measurements from functional assemblies comprising excitatory and inhibitory neurons in anaesthetised $App^{NL-G-F}$ mice (see Methods, Fig. 1h, i). Mice were co-injected with two viral constructs in

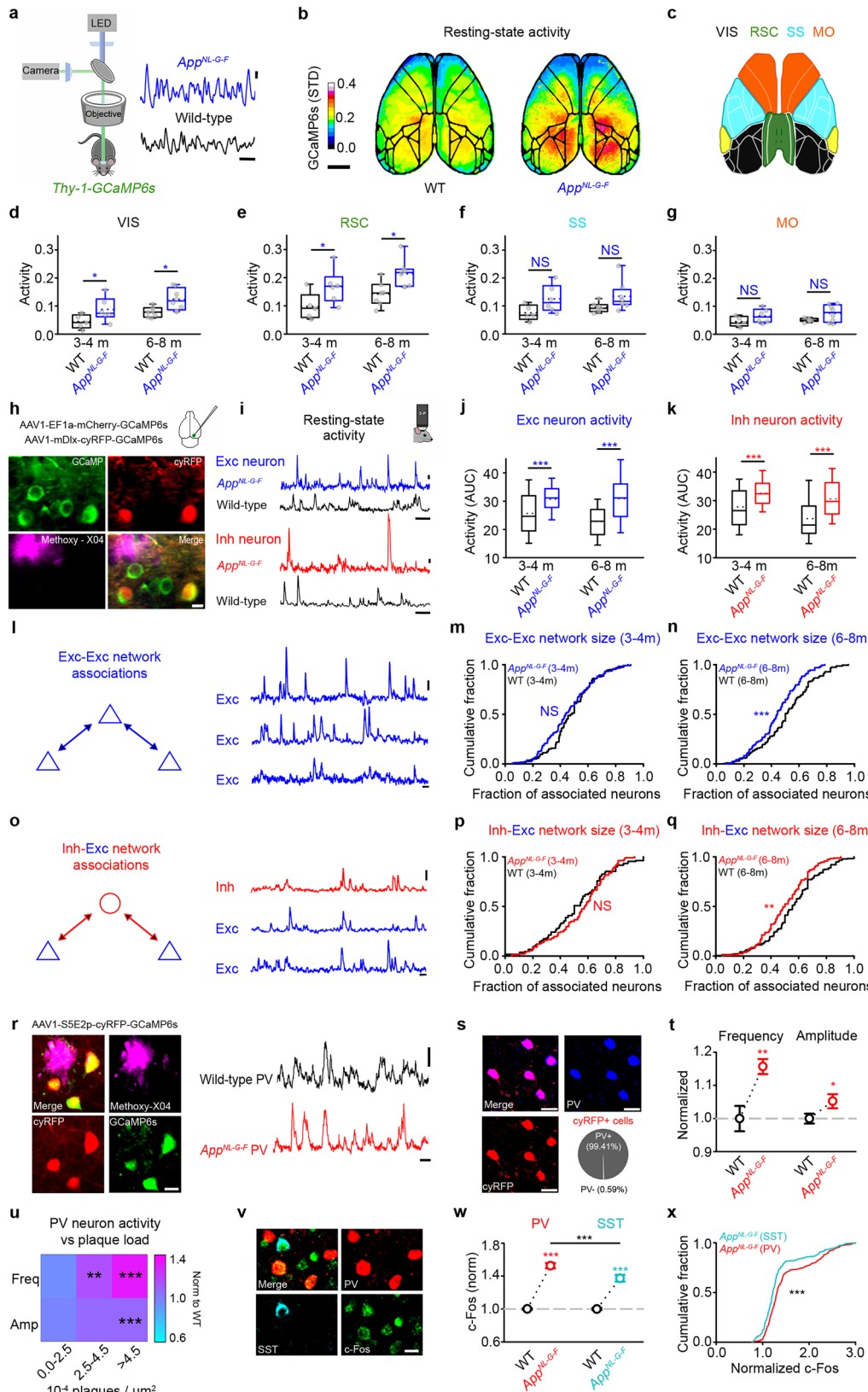

superficial layers of the primary visual cortex (V1), a cortical region showing early-stage dysregulation of resting-state activity (Fig. 1d). The first construct used the mDlx enhancer system to label a heterogeneous population of inhibitory neurons with both a functional (GCaMP6s) and a structural (cyRFP) marker (AAV1-mDlx-Kz-f-cyRFP1-GSG-P2A-GCaMP6s-WPRE-pA) (Fig. 1h)[47,48]. The second construct

labelled excitatory neurons with GCaMP6s and mCherry (AAV1-EF1a-mCherry-GSG-P2A-HIS-GCaMP6s-WPRE) (Fig. 1h)[49]. In previous work, we found less than one percent (0.37 %, 2/539 neurons) of labelled neurons expressed both constructs[39], suggesting labelling of separate neuronal populations[47,48]. Using these constructs, we simultaneously imaged, and separated, the calcium-mediated neuronal activity of

**Fig. 1 | Dysregulated resting-state activity at excitatory and inhibitory neurons during amyloidosis. a** Mesoscopic calcium-imaging (left) and resting-state ΔF/F$_0$ traces (right) from *App$^{NL-G-F}$*x*Thy-1-GCaMP6s* (blue) and *Thy-1-GCaMP6s* (black) mice. Scale bars: 0.2 ΔF/F$_0$ and 10 s. **b** Standard deviation maps from *Thy-1-GCaMP6s* (left) and *App$^{NL-G-F}$*x*Thy-1-GCaMP6s* (right) mice. **c–g** Dorsal cortex parcellation (**c**) and average activity (transient amplitude x frequency) per mouse at 3-4 m (left) and 6-8 m (right) in WT (black) and *App$^{NL-G-F}$* (blue) mice for visual/parietal (VIS, **d**), retrosplenial (RSC, **e**), somatosensory (SS, **f**) and motor (MO, **g**) regions. **h** Two-photon imaging of V1 excitatory (green) and inhibitory (red and green) neurons and methoxy-X04 (magenta). Scale bar: 10 μm. **i** Resting-state ΔF/F$_0$ traces from excitatory (top) and inhibitory (bottom) neurons in *App$^{NL-G-F}$* (blue, red) and WT (black) mice. Scale bars: 1.0 ΔF/F$_0$, 20 s. **j, k** AUC of ΔF/F$_0$ traces from excitatory (**j**, blue; MWRST, *p* < 0.001) and inhibitory (**k**, red, MWRST, *p* < 0.001) neurons in *App$^{NL-G-F}$* and WT (black) mice. **l, o** Schematics (left) and calcium trace (right) correlations from excitatory (**l**, blue), or excitatory and inhibitory (**o**, red) neuronal assemblies. Scale bars: 10 μm, 1.0 ΔF/F$_0$, 20 s. **m, n, p, q** Functionally associated neurons in 3-4 m (**m**, Student's *t* test, *p* = 0.111; **p** MWRST, *p* = 0.300) and 6-8 m (**n**, MWRST, *p* < 0.001; **q**, Student's *t* test, *p* = 0.003) *App$^{NL-G-F}$*(blue, red) and WT (black)

assemblies comprising excitatory neurons (**m, n**, blue), and inhibitory and excitatory neurons (**p, q**, red). **r** Two-photon imaging of PV neurons in V1 showing structural (cyRFP, red) and functional (GCaMP6s, green) markers with methoxy-X04 (magenta). Scale bar: 10 μm. Resting-state ΔF/F$_0$ traces from PV neurons in 4-6 m *App$^{NL-G-F}$* (red) and age-matched WT (black) mice. Scale bars: 1.0 ΔF/F$_0$, 10 s. **s** Immunofluorescence for PV (blue) in cyRFP-expressing neurons (red) and co-expression quantification. Scale bar: 20 μm. **t, u** Normalised transient frequency (left) and amplitude (right) in PV neurons from 4-6 m *App$^{NL-G-F}$* (red) and WT (black) mice (**t**, Welch's test) and versus methoxy-X04 plaque density (**u**, Student's *t* test; frequency, bin2, p = 0.005, bin3, p < 0.001; amplitude, bin3, p < 0.001). **v** Immunofluorescence for c-Fos (green), PV (red), and SST (cyan) in V1. Scale bar: 10 μm. **w,x** Normalised c-Fos intensity in PV (red) and SST (cyan) neurons in 4-6 m *App$^{NL-G-F}$* (red, cyan) and age-matched WT (black) mice (Two-Way ANOVA, HS, *p* < 0.001). Animals used: 26 (**a-g**), 19 (**h-q**) and 7 (**r-x**). Error bars: median (dotted: mean) with IQR (**d-k**), mean ± SEM (**t-w**), **p* < 0.05,***p* < 0.01,****p* < 0.001; NS, not significant. Two-sided tests used; detailed statistics: Supplementary Data S1. Source data provided as Source Data file.

neighbouring excitatory and inhibitory neurons during resting-state activity (Fig. 1i). We combined our 2-photon calcium imaging approach with live labelling of amyloid plaques using methoxy-X04[50] (Fig. 1h and Supplementary Fig. S1i). We then estimated somatic resting-state activity levels using the area under the curve (AUC) of the ΔF/F$_0$ calcium signal in excitatory and inhibitory cells (**see Methods**). We found increased levels of excitatory resting-state activity in *App$^{NL-G-F}$* mice compared to age-matched WT mice at the cellular and animal levels (Fig. 1j and Supplementary Fig. S1n). Changes in resting-state activity at excitatory neurons were mirrored by similar increases in the resting-state activity of inhibitory neurons (Fig. 1k and Supplementary Fig. S1n). This suggests that the greater levels of activity in excitatory cells are unlikely to be driven by population-level reductions in the suprathreshold activity of inhibitory cells in *App$^{NL-G-F}$* mice. We found the increased AUC in excitatory and inhibitory neurons to be driven by an increase in the amplitude of calcium events at both 3-4 and 6-8 months (Supplementary Fig. S1o–q). In excitatory neurons, the frequency of calcium events was elevated at 3-4 m, but lower than WT levels at 6-8 m (Supplementary Fig. S1r, t). In inhibitory neurons, the frequency of events was similar to WT levels at 3-4 m but then dropped to levels lower than WT at 6-8 m (Supplementary Fig. S1s, t). We investigated event-amplitude distributions to understand how reductions in the frequency of events at later timepoints could map to increases in the average event amplitude, and ultimately the total AUC. We found a reduction in the percentage of small-amplitude events and an increase in the percentage of large-amplitude events at both excitatory and inhibitory neurons at 6-8 m of age (Supplementary Fig. S1u, v), resulting in a significant increase in the total AUC. Given that the increased activity of excitatory and inhibitory neurons was driven by increases in calcium event amplitude (Supplementary Fig. S1o–q), we next tested the extent to which increases in amplitude were associated with plaque load at the animal level (Supplementary Fig. S1w). We found that higher plaque load was associated with increased amplitude in both excitatory and inhibitory neurons, while *App$^{NL-G-F}$* mice with low plaque densities were comparable to WT mice (Supplementary Fig. S1w). Excitatory neurons showed increased amplitude at all measured distances (Supplementary Fig. S1x). In contrast, inhibitory cells only showed increased activity within 20 μm of a plaque (Supplementary Fig. S1x).

Increased resting-state activity at excitatory neurons could occur in conditions of elevated inhibitory neuronal activity if the functional connectivity between assemblies of excitatory and inhibitory neurons (E-I assemblies) is reduced[30]. We therefore tested whether functional E-I assemblies were modified in the *App$^{NL-G-F}$* model (Fig. 1l–q). To do this, we used a previously published approach to estimate the functional association of E-I assemblies based on pairwise correlations

between somatic calcium traces (Fig. 1l, o)[46,48,51]. Using this method, positive and significant correlation values are thought to reflect mutual connectivity or shared inputs[52]. First, for each excitatory neuron, we calculated the fraction of associated excitatory neurons in the local cortical region (**see Methods**, Fig. 1l). We found fewer functional associations between excitatory neurons during resting-state activity by 6-8 m in *App$^{NL-G-F}$* mice compared to age-matched controls (Fig. 1m, n). We next calculated the fraction of excitatory neurons associated with local inhibitory neurons (**see Methods**, Fig. 1o) and found fewer functional associations between excitatory and inhibitory neurons in 6-8 m *App$^{NL-G-F}$* mice than in age-matched controls (Fig. 1p, q). These results suggest that resting-state neuronal activity is increased during amyloidosis and functional associations between E-E and I-E assemblies are disrupted.

The mDlx-based labelling approach targets a heterogeneous population of inhibitory neurons[47]. However, our previous work[39] and that of others[48] found a bias toward SST neurons using this enhancer system (~22 % of GCaMP-expressing neurons were PV positive and ~ 60 % were SST positive[39]). Therefore, to test for the involvement of other subclasses, we next conducted 2-photon imaging experiments using a PV-specific viral targeting strategy (AAV1-S5E2p-cyRFP-GSG-P2A-HIS-GCaMP6s-WPRE)[53,54] to label PV neurons with GCaMP6s and cyRFP in V1 (Fig. 1r). Immunofluorescence measures validated that 99.4 % (505/508 cells) of cyRFP positive neurons also expressed PV (Fig. 1s). Using 2-photon imaging, we found evidence for an increased frequency and amplitude of calcium events in PV neurons in 4-6 m *App$^{NL-G-F}$* mice relative to age-matched controls (Fig. 1t). In addition, elevated PV neuron activity was associated with greater local amyloid plaque load (Fig. 1u). We next conducted immunofluorescence experiments labelling PV, SST, and c-Fos in the visual cortex of WT and *App$^{NL-G-F}$* mice (Fig. 1v). Using this approach, we found increased c-Fos expression at both PV and SST neurons in *App$^{NL-G-F}$* mice by 4-6 m, with PV neurons showing a relatively greater increase than SST cells (Fig. 1w, x). Together, our data suggest that both PV and SST neurons show hyperactivity, based on increased c-Fos in both GABAergic subclasses, and elevated calcium activity measured with the mDlx construct (biased toward SST) and the S5E2 construct (PV-specific).

Our mesoscopic imaging data suggested that frontal regions, such as the motor cortex, are more resilient to hyperactivity than posterior regions (Fig. 1a–g and Supplementary Fig. S1a–h). To test whether activity changes are detectable at the cellular level in these frontal regions, we repeated our cellular imaging experiments by measuring the calcium activity of excitatory and inhibitory neurons, as well as methoxy-X04 plaque density, in the motor cortex at 4-6 months (Supplementary Fig. S1y). Consistent with our mesoscopic imaging data, activity levels in both excitatory and inhibitory cells were

comparable between $App^{NL-G-F}$ and WT mice (Supplementary Fig. S1y). We also found in vivo plaque load to be lower in the motor cortex than age-matched timepoints in the visual cortex (Supplementary Fig. S1z). Analysis of functional assemblies found a reduction in the size of E-E (Supplementary Fig. S1aa), but not I-E (Supplementary Fig. S1ab) assemblies. Together, this suggests that cellular and assembly-level inhibition is relatively intact at earlier timepoints in the motor cortex of $App^{NL-G-F}$ in comparison to more posterior regions.

## Presynaptic glutamatergic and GABAergic loss during amyloidosis

Weakening of functional neuronal assemblies may be due to synaptic changes[55]. We therefore tested for alterations in synaptic protein levels during amyloidosis (3-8 m) in the $App^{NL-G-F}$ mouse. We targeted superficial layers in posterior cortical regions identified as exhibiting dysregulated resting-state activity in vivo (visual and retrosplenial cortices) (Fig. 1d, e). We conducted immunofluorescence experiments to measure colocalisation of pre- and post-synaptic markers of glutamatergic (pre: VGLUT1 and post: PSD-95) and GABAergic (pre: VGAT and post: gephyrin) synapses in regions surrounding plaques in $App^{NL-G-F}$ mice and in WT controls (see Methods, Fig. 2a, b). In $App^{NL-G-F}$ mice, the normalised density of both glutamatergic and GABAergic colocalised synaptic puncta was reduced relative to WT controls (Fig. 2c, d), with GABAergic puncta showing earlier changes (Supplementary Fig. S2a, b). To determine the locus of synaptic density changes, we analysed pre- and post-synaptic puncta in isolation (Fig. 2e, f). We found a reduction in the density of VGLUT1, but no change in PSD-95 density in 3-8 m $App^{NL-G-F}$ mice when compared to WT (Fig. 2e). For GABAergic puncta, the density of VGAT was lower in $App^{NL-G-F}$ mice than WT mice, while gephyrin density was similar to WT levels (Fig. 2f). We compared the normalised change in VGLUT1 to VGAT in $App^{NL-G-F}$ mice and found more pronounced reductions in VGAT density (Supplementary Fig. S2c), pointing towards a greater vulnerability of the GABAergic presynapse in cortical regions with dysregulated resting-state activity. We next tested the extent to which synapse loss in the $App^{NL-G-F}$ mouse was also evident in other areas known to be sites of early-stage AD-related pathology, such as the entorhinal cortex[56,57]. Similar to measures in V1, we found evidence for the loss of both excitatory and inhibitory colocalised puncta in entorhinal cortex (Supplementary Fig. S2d). Glutamatergic changes were associated with a loss of both VGLUT1 and PSD-95 (Supplementary Fig. S2e), suggesting that the postsynaptic compartment also shows vulnerability in the entorhinal cortex. However, GABAergic changes were again associated with reductions in presynaptic VGAT levels without changes in gephyrin density (Supplementary Fig. S2f). These results suggest that presynaptic vulnerability is a common feature in the $App^{NL-G-F}$ mouse across the investigated regions, and is more prominent at GABAergic compared to glutamatergic synapses.

## Functional weakening of population-coupled GABAergic boutons in $App^{NL-G-F}$ mice

Our immunofluorescence measures found evidence for a loss of presynaptic GABAergic proteins (Fig. 2). However, these measures are blind to accompanying changes in synaptic activity. For example, GABAergic protein loss may be part of a global process of synaptic weakening or, alternatively, persistent synapses may exhibit compensatory changes in strength to offset local synapse loss[46,58]. To test between these scenarios, we measured in vivo calcium signals from presynaptic GABAergic boutons in the amyloid plaque microenvironment using 2-photon microscopy. To do this, we again utilised the mDlx enhancer strategy to express both cyRFP and GCaMP6s in superficial axons and boutons of V1 in 3-4 m and 6-8 m $App^{NL-G-F}$ and WT mice (see Methods, Fig. 3a). Using this approach, we obtained

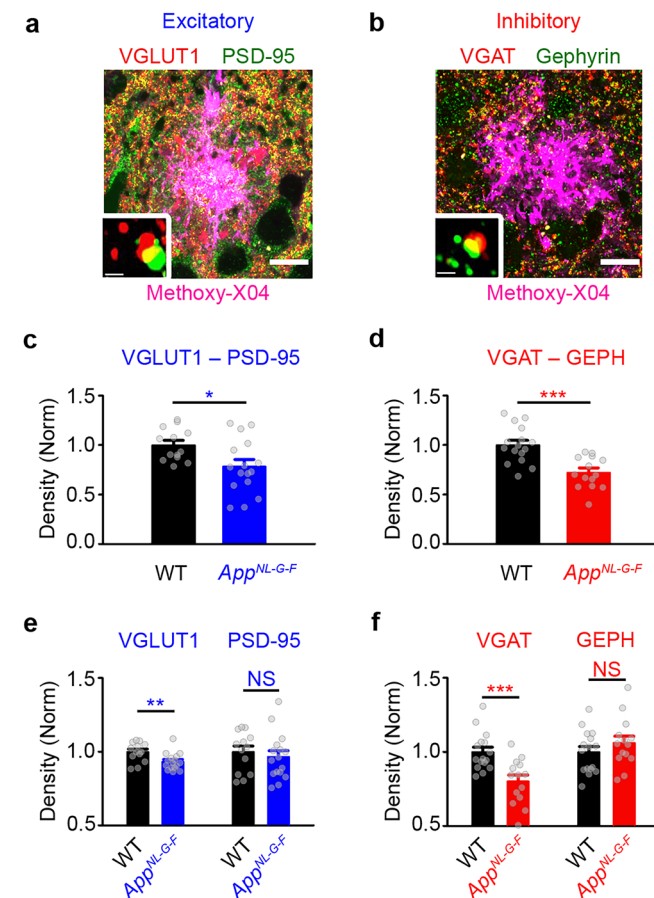

**Fig. 2 | Presynaptic GABAergic vulnerability in $App^{NL-G-F}$ mouse cortex.**
**a, b** Cortical region showing immunofluorescence labelling of presynaptic (red), postsynaptic (green), and colocalised (yellow) glutamatergic (**a**) and GABAergic (**b**) markers in the vicinity of amyloid plaques (magenta) in $App^{NL-G-F}$ mice. Scale bar: 10 μm. Insets show colocalization between the pre- and post-synaptic markers. Scale bar: 1 μm. **c, d** Normalised (to WT) colocalised glutamatergic (**c**, Student's $t$ test, $p = 0.018$) and GABAergic (**d**, Student's $t$ test, $p < 0.001$) synaptic puncta in 3-8 m $App^{NL-G-F}$ (blue, red) and age-matched WT (black) mice. **e, f** Normalised (to WT) density of excitatory (**e**, Student's $t$ test, left, $p = 0.007$; right, $p = 0.553$) and inhibitory (**f**, Student's $t$ test, left, $p < 0.001$; right, $p = 0.278$) pre- (left) and post- (right) synaptic proteins in 3-8 m $App^{NL-G-F}$ (blue, red) and age- matched WT mice (black). Data obtained from 17 animals. Error bars: mean ± SEM, *$p < 0.05$, **$p < 0.01$, ***$p < 0.001$; NS, not significant. Two-sided tests used; detailed statistics: Supplementary Data S2. Source data provided as Source Data file.

structural and functional measures of GABAergic axonal boutons in vivo and spatially related them to local amyloid pathology via methoxy-X04 labelling (see Methods).

We first measured the density of GABAergic boutons in vivo (see Methods) and found that bouton density was stable at 3-4 m (Fig. 3b) but reduced by 6-8 m (Fig. 3c) in $App^{NL-G-F}$ mice when compared to WT. GABAergic bouton density showed the greatest reduction in areas with high plaque load (Fig. 3d). We next measured calcium-mediated activity at GABAergic boutons during resting-state activity at 3-4 m and 6-8 m, and extracted bouton responses based on published methods[44,49,59] (see Methods, Supplementary Fig. S3a). We found GABAergic bouton activity was reduced at both 3-4 m and 6-8 m in $App^{NL-G-F}$ mice when compared to WT (Fig. 3e–h and Supplementary Fig. S3b, c). At 3-4 m, reduced GABAergic bouton activity was associated with reductions in the frequency and amplitude of calcium events (Fig. 3e, f), while reductions in amplitude were more prominent

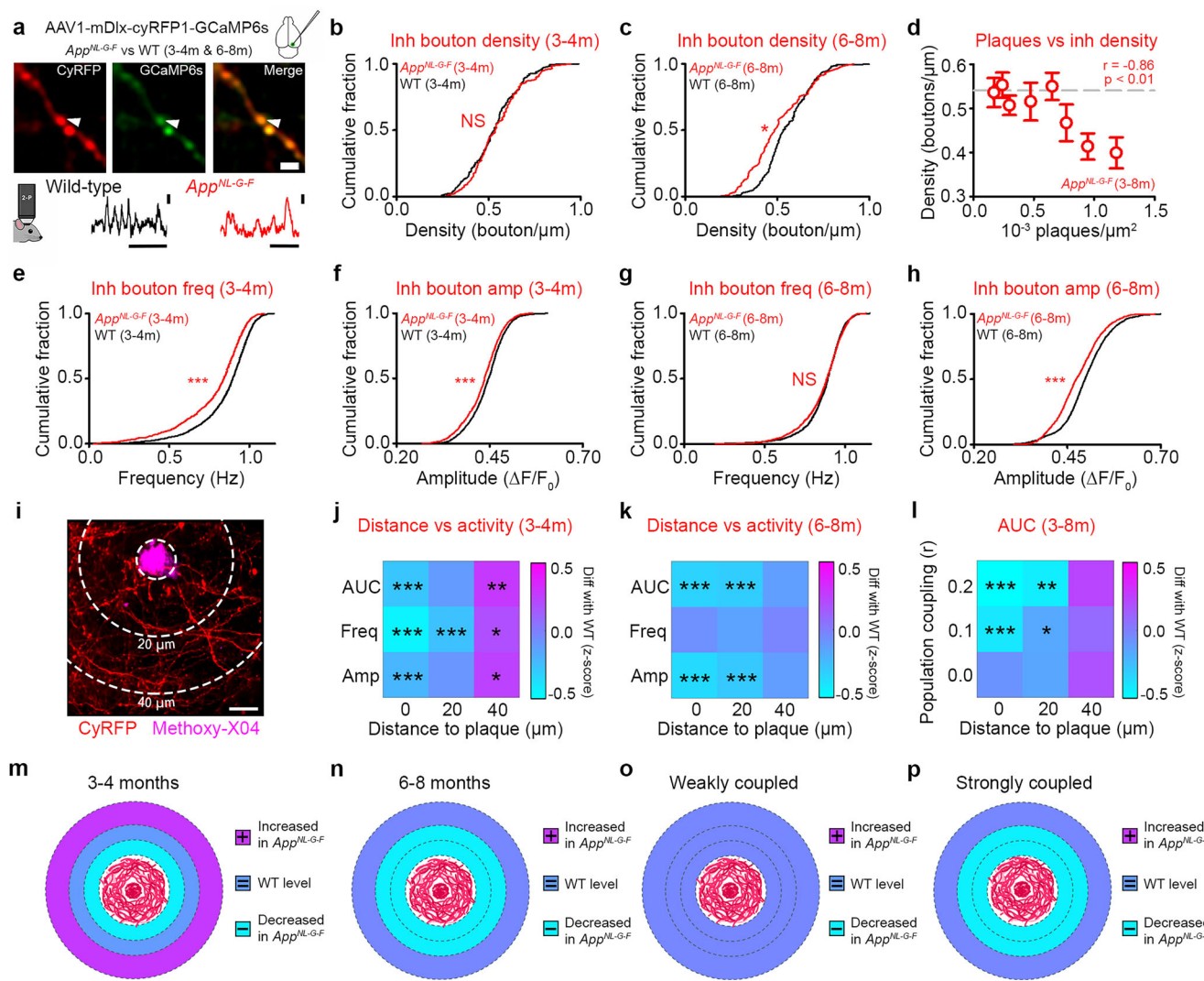

**Fig. 3 | Functional signatures of dysregulated bouton activity in $App^{NL-G-F}$ mice.**
**a** Top: GABAergic axon expressing cyRFP (red) and GCaMP6s (green) in superficial V1 imaged in vivo with 2-photon microscopy, giving structure/function merge (right). Bottom: example WT (black) and $App^{NL-G-F}$ (red) ΔF/F$_0$ traces from boutons. Scale bars: 2 μm, 0.5 ΔF/F$_0$ and 10 s. **b, c** Density of GABAergic boutons in 3-4 m (**b**) and 6-8 m (**c**) $App^{NL-G-F}$ (red) and WT (black) mice. **d** Bouton density versus methoxy-X04-labelled plaque density (Pearson correlation, boutons density vs plaques density, $r = -0.86$, $p = 0.006$). **e–h** GABAergic bouton frequency (**e**, MWRST, $p < 0.001$; **g**, MWRST, $p = 0.116$) and amplitude (**f**, MWRST, $p < 0.001$; **h**, MWRST, $p < 0.001$) in $App^{NL-G-F}$ (red) and age-matched WT (black) animals at 3-4 m (**e**, **f**) and 6-8 m (**g**, **h**). **i** Superficial GABAergic axons expressing cyRFP (red) and a methoxy-X04-labelled plaque (magenta). Dashed circles represent plaque edge and 20 μm increments. Scale bar: 10 μm. **j, k** Heatmaps showing difference ($App^{NL-G-F}$-WT) in the z-scored (to WT) average area under curve (AUC), frequency (Freq), and amplitude (Amp) values of GABAergic boutons versus distance to methoxy-X04-labelled plaques in 3-4 m (**j**, One-Way ANOVA; AUC, bin = 0, $p < 0.001$, bin = 40, $p = 0.007$; frequency, bin = 0, $p < 0.001$, bin = 20, $p < 0.001$, bin = 40, $p = 0.045$; amplitude,

bin = 0, $p < 0.001$, bin = 40, $p = 0.019$) and 6-8 m (**k**, One-Way ANOVA; AUC, bin = 0, $p < 0.001$, bin = 20, $p < 0.001$; amplitude, bin = 0, $p < 0.001$, bin = 20, $p < 0.001$) $App^{NL-G-F}$ mice. Values on the x-axis represent the minimum distance for each bin. **l** Heatmaps showing difference ($App^{NL-G-F}$-WT) in the z-scored (to WT) AUC values of GABAergic boutons with increasing population coupling values (y-axis) and distance from methoxy-X04-labelled plaques (x-axis) in 3–8 m $App^{NL-G-F}$ mice (Two-Way ANOVA, $p < 0.001$; HS, bin(0.2,0), $p < 0.001$; bin(0.2,20), $p = 0.005$; bin(0.1,0), $p < 0.001$; bin(0.1,20), $p = 0.022$). Values on the x- and y-axes represent the minimum distance and population coupling score for each bin. **m–p** Schematics of peri-plaque regions showing increased (magenta), unchanged (blue), or decreased (cyan) activity (AUC) of GABAergic boutons in $App^{NL-G-F}$ mice compared to WT in 3-4 m (**m**) vs 6-8 m (**n**), and for boutons showing weak (**o**) versus strong (**p**) population coupling. Circles around the plaque represent areas at 0–20, 20–40, and > 40 μm. Data obtained from 27 animals. Error bars: mean ± SEM, *$p < 0.05$,**$p < 0.01$,***$p < 0.001$; NS, non-significant. Two-sided tests used; detailed statistics: Supplementary Data S3. Source data provided as Source Data file.

at 6-8 m (Fig. 3g, h). At both timepoints, GABAergic boutons with reduced and increased activity could be found within the same axon in $App^{NL-G-F}$ mice (Supplementary Fig. S3d), suggesting that functional alterations are bouton specific. However, the fraction of axons which exclusively housed GABAergic boutons with reduced activity more than doubled between 3-4 m and 6-8 m (Supplementary Fig. S3d). Our data suggest a complex activity profile for GABAergic boutons that varies with both age and distance to plaques. More distal GABAergic boutons initially map to elevated GABAergic cellular activity at early

time points but then show a decrease in activity with both time and proximity to pathology (Fig. 3i-k). To understand how the elevated cellular activity could map to reductions in bouton activity, we examined the event-amplitude distributions of boutons with low activity levels ($App^{NL-G-F}$). We found a shift in the distribution of calcium-event sizes, so that small amplitude events in $App^{NL-G-F}$ mice were more frequent, while medium- and high-amplitude events became more infrequent than in WT mice (Supplementary Fig. S3e–g). Together, these results show that synaptic and cellular hyper- and hypo-activity

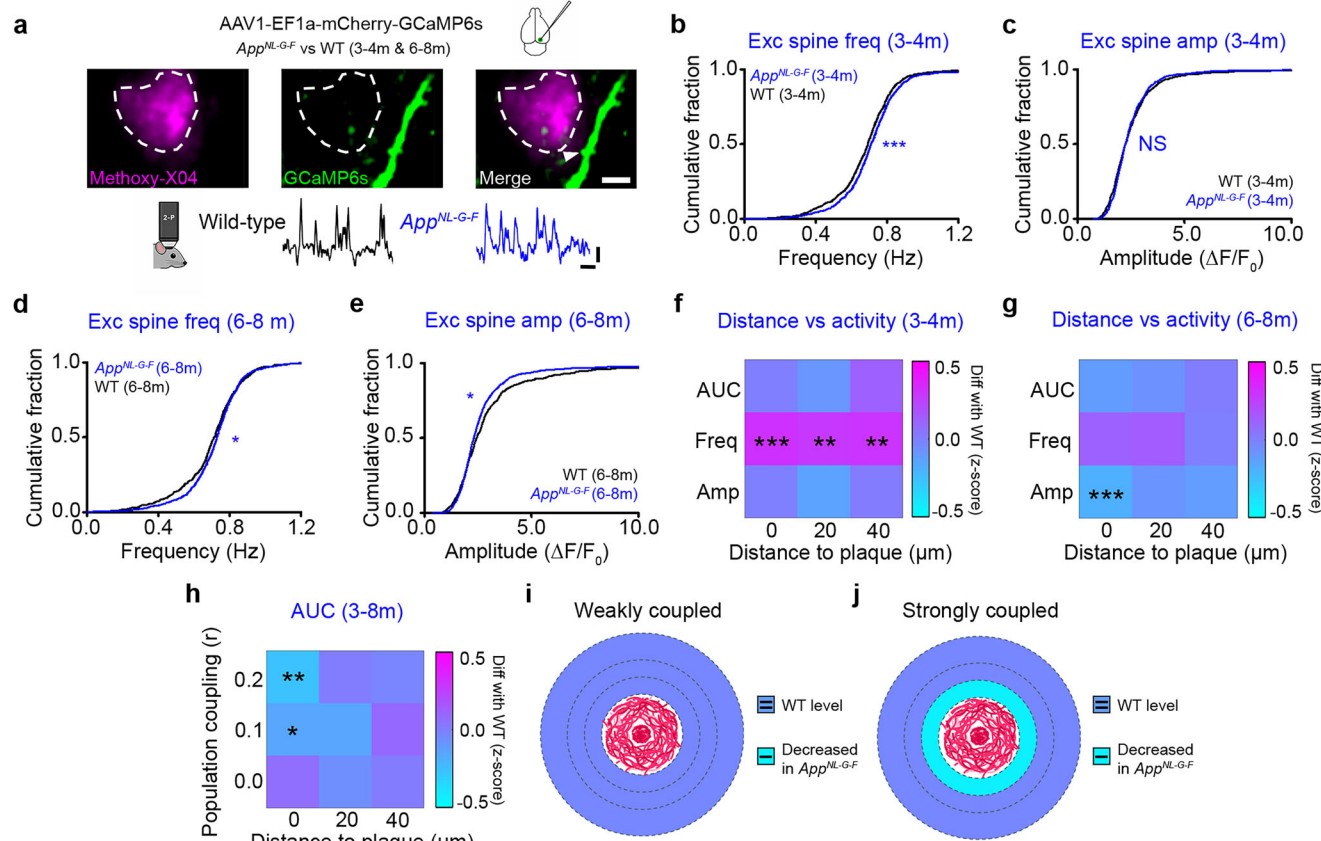

**Fig. 4 | Functional signatures of dysregulated spine activity in $App^{NL-G-F}$ mice.**
**a** Top: Example region depicting methoxy-X04-labelled amyloid plaque (magenta) and GCaMP6s-expressing dendritic branch (green). Bottom: Example activity trace from WT (black) and $App^{NL-G-F}$ (blue) spines. Scale bars: 5 μm, 1.0 ΔF/F$_0$ and 1 s. **b**–**e** In vivo dendritic spine frequency (**b**, MWRST, p < 0.001; **d**, MWRST, p = 0.618) and amplitude (**c**, MWRST, p = 0.042; **e**, MWRST, p = 0.021) in $App^{NL-G-F}$ (blue) and age-matched WT (black) animals at 3-4 m (**b**, **c**) and 6-8 m (**d**, **e**). **f**, **g** Heatmaps showing difference ($App^{NL-G-F}$-WT) in the z-scored (to WT) average AUC, frequency (Freq), and amplitude (Amp) values of excitatory spines located at increasing distances from methoxy-X04-labelled plaques in 3-4 m (**f**, One-Way ANOVA; frequency, bin = 0, p < 0.001, bin = 20, p = 0.003, bin = 40, p = 0.006) and 6-8 m (**g**, One-Way ANOVA; amplitude, bin = 0, p < 0.001) $App^{NL-G-F}$ mice. Values on the x-axis represent the minimum distance for each bin. **h** Heatmap showing difference ($App^{NL-G-F}$-WT) in the z-scored (to WT) AUC values of spines with increasing population coupling values (y-axis) and distance from methoxy-X04 labelled plaques (x-axis) in 3-8 m $App^{NL-G-F}$ mice (Two-Way ANOVA, p = 0.002; HS, bin(0,2,0), p = 0.002; bin(0,1,0), p = 0.026). Values on the x- and y-axes represent the minimum distance and population coupling score for each bin. **i**, **j** Schematic of peri-plaque regions showing decreased (cyan) or unchanged (blue) activity (AUC) in $App^{NL-G-F}$ mice compared to WT for excitatory spines showing weak (**i**) vs strong (**j**) population coupling. Circles around the plaque represent areas at 0–20, 20–40, and > 40 μm. Data obtained from 21 animals. *p < 0.05, **p < 0.01, ***p < 0.001; NS, not significant. Two-sided tests used; detailed statistics: Supp. Data S4. Source data provided as Source Data file.

may initially coexist in early-stage amyloidosis, and that periods of increased GABAergic bouton activity may precede functional weakening in more distal areas.

Some GABAergic neurons are known to play an important role in modulating the synchrony of population-level resting-state activity[60], which we found to be disrupted at the mesoscopic and cellular levels (Fig. 1). Therefore, we next tested if GABAergic boutons with varying degrees of population coupling exhibited different changes in activity levels. Based on previous work, we quantified population coupling as the functional association between each bouton's activity and the average activity of its home region[61,62] (**see Methods**). We found GABAergic boutons with patterns of activity that were strongly coupled to the population activity showed the greatest reductions in activity (Supplementary Fig. S3h). In addition, there was an interplay between population coupling and proximity to plaque for GABAergic bouton activity (Fig. 3l). Within GABAergic boutons that were proximal to plaques, only those with population-coupled activity showed weakening, while proximal boutons with low levels of coupling had activity levels that were similar to WT (Fig. 3l). These results suggest that the functional profile, combined with the proximity to amyloid

plaque pathology, may be a key feature of synaptic pathophysiology (Fig. 3m–p).

## Functional weakening of population-coupled dendritic spines in $App^{NL-G-F}$ mice

Excitatory synaptic dysfunction in vivo may be detectable ahead of prominent changes in protein markers (Fig. 2). We therefore tested the extent to which the calcium-mediated activity of dendritic spines was modified during increasing amyloidosis (Fig. 4). To do this, we used in vivo 2-photon imaging to measure calcium signals at dendritic spines of L2/3 excitatory cells in V1 of 3-4 m and 6-8 m $App^{NL-G-F}$ mice and age-matched WT controls (**see Methods**, Fig. 4a). We imaged and extracted spine activity using published methods[44,49,59] (**see Methods**). We found a small but significant increase in the frequency of calcium events at excitatory dendritic spines in $App^{NL-G-F}$ mice at 3-4 m (Fig. 4b, c and Supplementary Fig. S4a). At 6-8 m, while the frequency of spine events was still elevated (Fig. 4d), the amplitude was reduced (Fig. 4e), resulting in a net reduction of overall spine activity levels (Supplementary Fig. S4b). The initial increase in spine

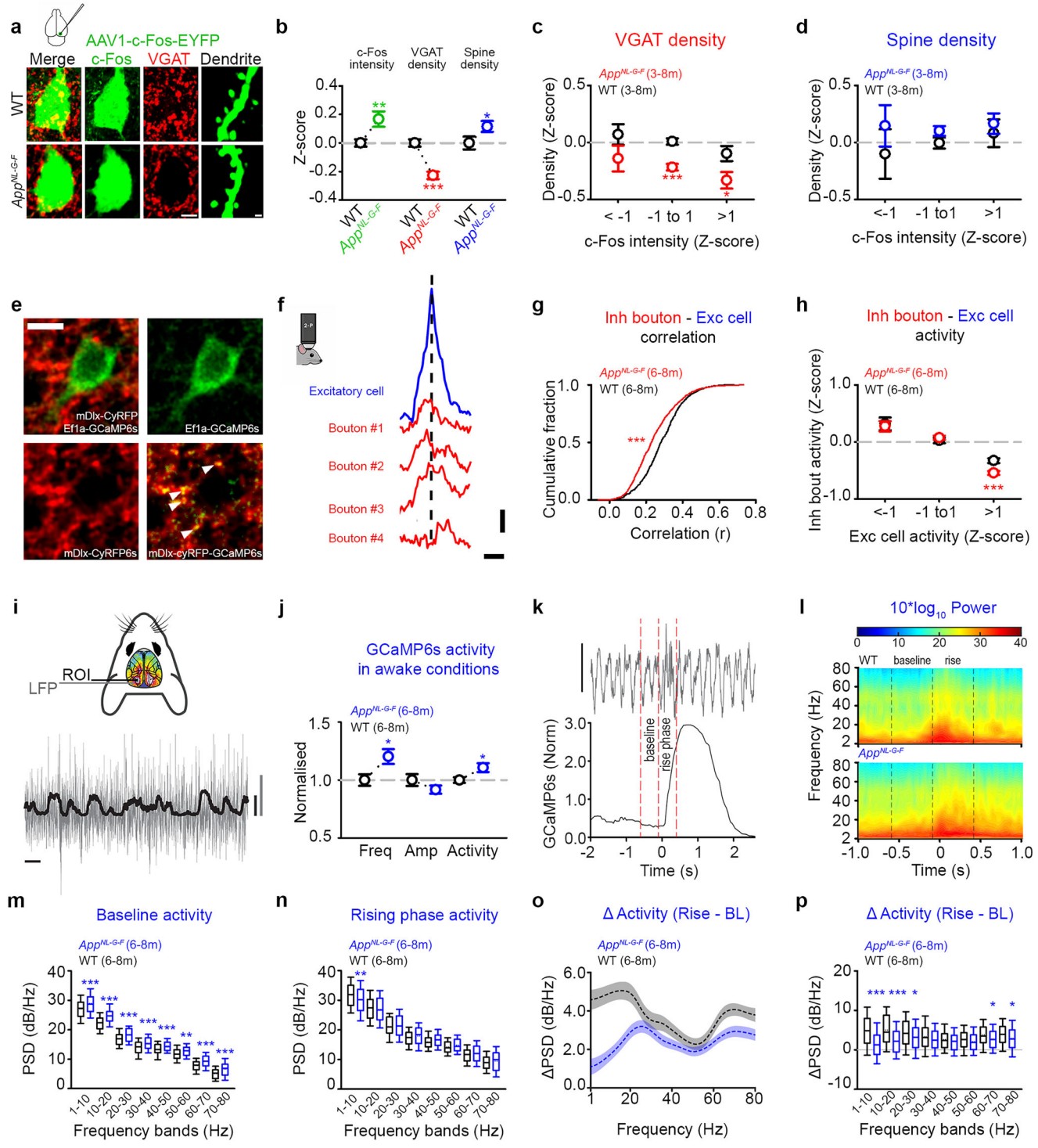

frequency occurred both in proximal and distal regions at 3-4 m (Fig. 4f), whilst the reduction in spine amplitude at 6-8 m was more localised to spines proximal to plaques (Fig. 4g). Following the pattern observed at GABAergic boutons (Fig. 3l), population-coupled spines that were proximal to plaques showed the greatest reductions in activity, whilst proximal spines with low population coupling had activity levels that were similar to WT animals (Fig. 4h). Together with the findings from GABAergic bouton measures, these results suggest that the functional profile, rather than the proximity to amyloid plaques alone, may be a key feature of both excitatory and inhibitory AD-related synaptic pathophysiology (Figs. 3m–p, 4i, j). Compared to activity changes in

excitatory spines, functional weakening at GABAergic boutons occurs earlier (Supplementary Figs. S3b, c, S4a, b) and is more prominent (Supplementary Fig. S4c, d). In addition, changes in GABAergic bouton, but not spine, activity correlated with plaque load at the animal level (Supplementary Fig. S4e, f). Together, this highlights the early-stage vulnerability of cortical GABAergic circuitry to amyloid-related pathology in this model in vivo.

## Presynaptic GABAergic loss and dysfunction are associated with highly active excitatory neurons

Reductions in inhibitory bouton density and/or activity (Fig. 3) may partly mediate functional uncoupling of E-I assemblies (Fig. 1l–q),

**Fig. 5 | Hyperactivity is associated with GABAergic dysfunction and abnormal oscillatory activity in a mouse model of amyloidosis. a** Viral-mediated c-Fos-based activity-tagging. Immunofluorescence for VGAT (red) with c-Fos-EYFP-positive cells (left) and dendrites (right) in 3-8 m WT (top) and $App^{NL-G-F}$ (bottom) mice. Scale bars: 5 μm (left), 2 μm (right). **b** Normalised c-Fos intensity (green; Welch's $t$ test, $p = 0.005$), VGAT (red; Student's $t$ test, $p < 0.001$) and dendritic spine (blue; Welch's $t$ test, $p = 0.048$) density in 3-8 m $App^{NL-G-F}$ and WT (black) mice. **c, d**,Normalised excitatory c-Fos expression versus VGAT (**c**, Student's $t$ tests; bin1, $p = 0.145$; bin2, $p < 0.001$; bin3, $p = 0.018$) and spine (**d**, Student's $t$ tests; bin1, $p = 0.391$; bin2, $p = 0.102$; bin3, $p = 0.567$) density in 3–8 m $App^{NL-G-F}$ (red, blue) and WT (black) mice. **e** Top left: GABAergic mDlx-CyRFP (red) boutons proximal to GCaMP6s-expressing excitatory neuron (green). Top right: GCaMP6s excitatory cell. Bottom left: mDlx-cyRFP boutons. Bottom right: Merged structural bouton (red) with mDlx-GCaMP6s (green) channels showing bouton-specific activity (arrowheads). Scale bar 10 μm. **f** Event-triggered averaging gives excitatory neuronal somatic waveform (blue) and proximal inhibitory bouton activity (red). The dashed line is the time of peak excitatory activity. Scale bars: 0.2 ΔF/F₀ and 0.5 s. **g** Correlation of GABAergic bouton activity with local excitatory neuron activity in 6-8 m $App^{NL-G-F}$ (red) and WT (black) mice (MWRST, $p < 0.001$). **h** Normalised excitatory neuronal activity in 6-8 m $App^{NL-G-F}$ (red) and WT (black) mice versus activity (amplitude) from spatially-temporally associated GABAergic boutons (Student's $t$ tests; bin1, $p = 0.776$; bin2, $p = 0.394$; bin3, $p < 0.001$.). **i** Schematic (top) and of simultaneous superficial local field potential (LFP, grey) and 1-photon mesoscopic calcium-mediated (black) recordings (bottom) in awake mice. Scale bars: 2.0 ΔF/F₀, 400nV, and 5 s. **j** Normalised frequency ($p = 0.032$), amplitude ($p = 0.199$), and total activity (freq×amp; $p = 0.045$) per mouse in awake $Thy-1-GCaMP6s$ (black) and $App^{NL-G-F}×Thy-1-GCaMP6s$ (blue) visual cortex (Student's $t$ test). **k** LFP trace (top) associated with calcium transient (bottom). **l** Time-frequency power plot in WT (top) and $App^{NL-G-F}$ (bottom) mice. Dashed lines denote baseline and rising-phase (**k, l**). **m, n** Power spectral density (PSD) estimates in 10 Hz bands during baseline (**m**, Two-Way ANOVA, genotype, $p < 0.001$) and rising-phase (**n**, Two-Way ANOVA, genotype×band, $p = 0.014$). **o, p** Change in PSD estimate between baseline and rising-phase (**o**) in 10 Hz bands (**p**, Two-Way ANOVA, genotype×band, $p < 0.001$). Animals used: 20 (**b-d**), 15 (**g, h**) and 13 (**j-p**). Error bars: mean ± SEM (**b–j**), median (dotted = mean) ± IQR ± 5th-95th percentile (**m–p**). *$p < 0.05$,**$p < 0.01$,***$p < 0.001$; NS, non-significant. Two-sided tests used; detailed statistics: Supplementary Data S5. Source data provided as Source Data file.

ultimately contributing to amyloid-associated increases in resting-state activity (Fig. 1a–k). We therefore tested if the loss and dysfunction of GABAergic presynaptic compartments may be spatially associated with elevated neuronal activity in $App^{NL-G-F}$ mice. We first examined whether VGAT loss was evident at neurons with elevated expression of c-Fos, an immediate early gene known to correlate with neuronal activity levels[63]. To do this, we used a viral strategy to express a c-Fos-EYFP construct in regions identified as exhibiting dysregulated activity (visual and retrosplenial cortices) in 3-8 m $App^{NL-G-F}$ mice and age-matched WT controls (**see Methods**, Fig. 5a). The c-Fos-EYFP construct uniformly labelled the entire dendritic arborisation of neurons expressing c-Fos, including dendritic spines (Fig. 5a and Supplementary Fig. S5a). Following viral expression, we prepared brain slices for immunofluorescence labelling to measure both VGAT puncta and dendritic spines at c-Fos+ neurons (**see Methods**, Fig. 5a). In agreement with our in vivo imaging (Fig. 1), we found c-Fos levels to be greater in $App^{NL-G-F}$ mice than WT controls (Fig. 5b). In $App^{NL-G-F}$ mice, there was a broad reduction in VGAT density at c-Fos positive neurons, and a small but significant increase in spine density (Fig. 5b). Reductions in VGAT density were greater in cells with higher c-Fos levels (Fig. 5c), suggesting that loss of GABAergic presynaptic puncta may be associated with elevated neuronal activity levels in early-stage amyloidosis. In contrast, dendritic spine density (Fig. 5d) and size (Supplementary Fig. S5b, c) were similar to controls across cells with different c-Fos expression levels.

We next investigated the relationship between changes in the activity of GABAergic boutons and spatially proximal excitatory neurons in vivo. To do this, we returned to our co-expression experiments, which used the mDlx enhancer system to label a heterogeneous population of inhibitory neurons with both a functional (GCaMP6s) and a structural (cyRFP) marker[47,48] and a second construct to label excitatory neurons with GCaMP6s and mCherry (**see Methods**, Fig. 1h, i). Using this approach, we were able to observe putative baskets of GABAergic boutons (expressing both cyRFP and GCaMP6s) surrounding the soma of excitatory neurons (expressing GCaMP6s and mCherry) in superficial layers of visual cortex (Fig. 5e). We used in vivo 2-photon calcium imaging to simultaneously image and separate the calcium-mediated neuronal activity of excitatory neurons and their adjacent inhibitory boutons in 6-8 m WT and $App^{NL-G-F}$ mice (Fig. 5e, f). We first used correlation-based analysis to test the functional properties of GABAergic boutons in relation to local excitatory neuronal activity (**see Methods**). We found the activity of GABAergic boutons at axonal baskets was less correlated with its spatially associated excitatory neuron activity in $App^{NL-G-F}$ mice compared to age-matched WT animals (Fig. 5g). We next used event-triggered averaging to test for changes in GABAergic bouton activity that were spatially and temporally associated with activity levels in the adjacent excitatory neuron (**see Methods**, Fig. 5f). We identified calcium transients in excitatory neurons and generated average inhibitory bouton signals time-locked to the transients in excitatory neurons (Fig. 5f). We found that, in $App^{NL-G-F}$ animals, the activity of inhibitory boutons that were spatially and temporally associated with highly active excitatory neurons was reduced relative to equivalent values in WT animals (Fig. 5h). These data suggest that reductions in inhibitory bouton activity are associated with elevated neuronal activity levels in early-stage amyloidosis.

## Amyloid-associated hyperactivity is accompanied by deficits in specific low and high frequency neuronal oscillations

Our data suggest that synaptic dysfunction is evident during amyloid-associated hyperactivity (Figs. 3, 4). We wondered whether other bio-markers of pathophysiology, which may have greater translational relevance, are also evident during this time. To address this issue, we combined widefield 1-photon mesoscopic calcium imaging with simultaneous EEG recordings in awake $Thy-1-GCaMP6s$ mice and $App^{NL-G-F}×Thy-1-GCaMP6s$ crosses (**see Methods**, Fig. 5i). In agreement with our earlier mesoscopic calcium imaging work (Fig. 1d, e), we found increased resting-state calcium-mediated neuronal activity only in the posterior cortical regions (VIS and RSC) of awake $App^{NL-G-F}$ mice when compared to controls (Fig. 5j and Supplementary Fig. S5d). We then used event-triggered averaging of mesoscopic calcium events from the visual cortex and measured temporally associated local field potential (LFP) recordings from a superficial electrode (Fig. 5k). We investigated the power of EEG frequency bands during the pre-event baseline and the rising phase of the calcium event (Fig. 5k, l). Following our imaging results, we observed increased power in $App^{NL-G-F}$ mice, spanning all measured frequency bands in the baseline phase immediately before the onset of a calcium event (Fig. 5m). During the rising phase, we found evidence for reduced power in the 1-10 Hz range (Fig. 5n). To better understand the oscillatory dynamics associated with calcium events, we then measured the difference in the power of the frequency bands from baseline to the rising phase. Here, we found a reduced response in $App^{NL-G-F}$ mice, specifically in lower (1-30 Hz) and higher (60-80 Hz) frequency bands (Fig. 5o, p). Our data suggest that despite the global increase in power across all bands, the onset of amyloid-associated hyperactivity in $App^{NL-G-F}$ mice coincides with deficits in more specific low and high-frequency neuronal oscillations in vivo.

## Modified gene expression relating to GABAergic synaptic transmission in the $App^{NL-G-F}$ mouse

Our data suggest that GABAergic microcircuitry is dysfunctional in the $App^{NL-G-F}$ mouse. To determine the relative contribution of different

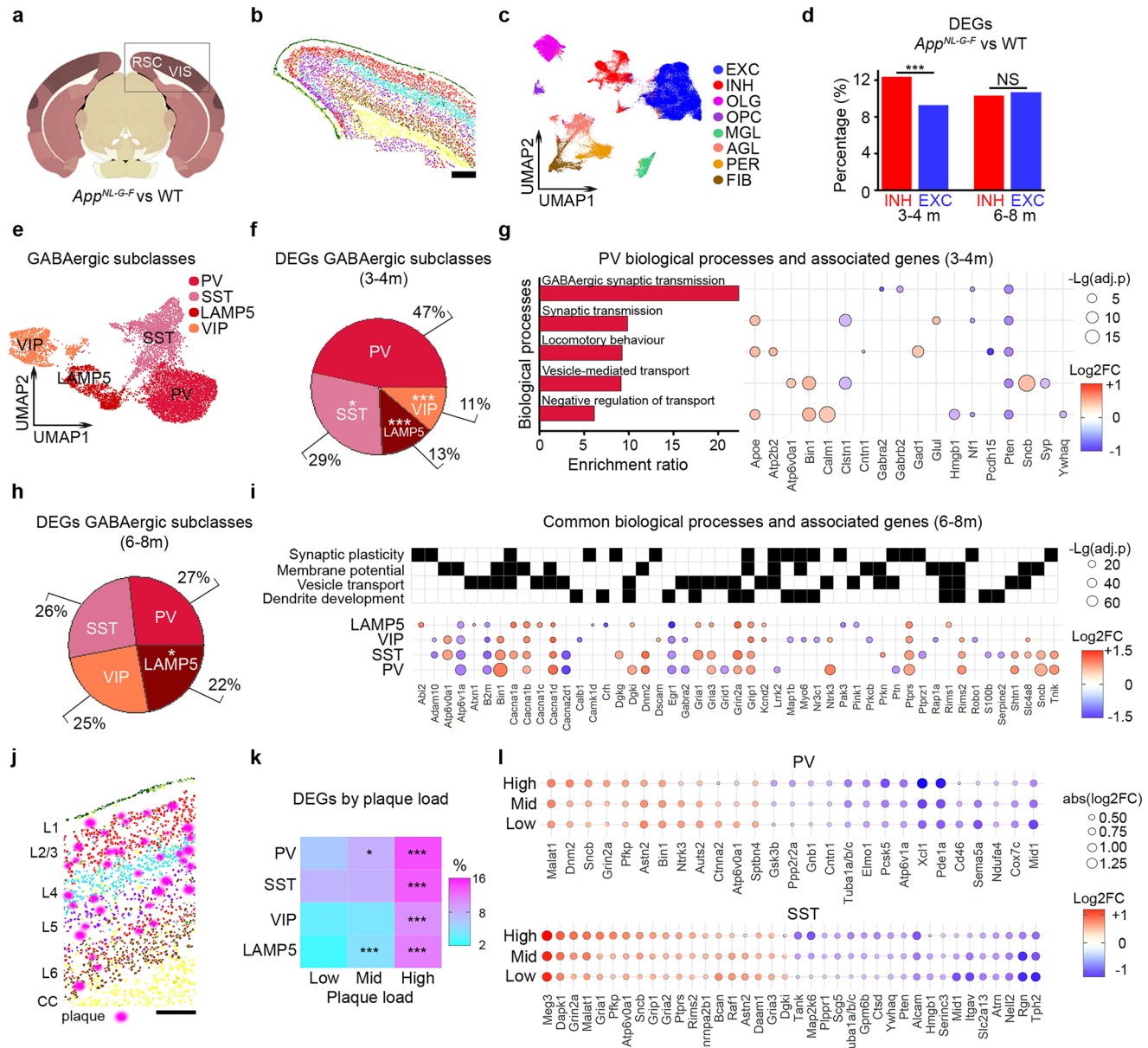

**Fig. 6 | Modified synaptic transmission at GABAergic neurons in the $App^{NL-G-F}$ mouse. a** Brain section schematic showing regions of interest (visual, VIS; retrosplenial, RSC) for spatial transcriptomics (Allen CCFv3). **b** Example section showing laminar expression profile of excitatory neurons. Scale bar 300 μm. **c** UMAP for main cell-types: excitatory (EXC) and inhibitory (INH) neurons, oligo-dendroglial cells (OLG), oligodendrocyte precursor cells (OPC), microglia (MGL), astroglia (AGL), pericytes (PER), and fibroblasts (FIB). **d** Percentage of DEGs for INH (red) and EXC (blue) cells when comparing $App^{NL-G-F}$ and WT at 3–4 m (left, z-test, $p < 0.001$) and 6-8 m (right, z-test, $p = 0.711$). **e** UMAP of GABAergic subclasses (PV, SST, LAMP5, and VIP). **f, h** Percentage of DEGs across GABAergic subclasses when comparing $App^{NL-G-F}$ and WT at 3-4 m (**f**, chi-squared test, $p < 0.001$) and 6-8 m (**h**, chi-squared test, $p = 0.130$). Comparison against PV shown for each subclass. **g** Biological processes and associated genes in PV interneurons following pathway enrichment at 3-4 m. **i** Subset of DEGs from GABAergic subclasses (dot plot) and top biological processes (grid plot) at 6-8 m. **j** Example cortical image showing coordinates for cells and plaques in V1 (layers, L1-6, and corpus callosum, CC). Scale bar:

200 μm. **k** Percentage of DEGs for inhibitory subclasses in $App^{NL-G-F}$ mice versus areas with low, mid, and high plaque number. Comparisons against low plaque areas shown for each subclass (z-test; PV, mid, $p = 0.012$; high, $p < 0.001$; SST, high, $p < 0.001$; VIP, high, $p < 0.001$; LAMP5, mid, $p < 0.001$; high, $p < 0.001$). **l** DEGs that covary with plaque load for PV (top) and SST (bottom) in $App^{NL-G-F}$ mice (Pearson correlation; PV: negFC: $r = -0.38$, $p = 0.038$; posFC: $r = 0.62$, $p = 0.015$; SST: negFC: $r = -0.67$, $p < 0.01$; posFC: $r = 0.38$, $p = 0.016$). All panels DEGs: MAST, adj. $p < 0.05$ and abs(log2FC) > 0.25. Biological processes: ORA, BH- FDR < 0.05 (**g, i**). Dot plots, size proportional to -Lg(FDR) (**g, i**) or the abs(Log2FC) (**l**); colour bars show Log2FC (**g, i, l**). Grid plot (**i**), black, association of genes (x-axis) with biological processes (y-axis). For panels (**b** and **j**), colour coding based on Louvain clustering and showing, from pia, vascular cells (green); EXC cells: L2/3 (red), L4 (turquoise), L5 (purple), L6 (brown); and CC OLG (yellow). Data obtained from 16 animals.

*$p < 0.05$,***$p < 0.001$; NS, not significant. Two-sided tests used; detailed statistics: Supp. Data S6. Source data provided as Source Data file.

GABAergic subclasses to abnormal synaptic processing, we tested for cell-type-specific molecular processes associated with early-stage amyloidosis. To do this, we used the CosMx™ (Nanostring) spatial transcriptomic system to investigate cellular changes in posterior cortical regions (visual and retrosplenial cortices) identified as expressing

dysregulated resting-state activity in 3-4 m and 6-8 m $App^{NL-G-F}$ mice (Fig. 1). We analysed data from 63.5 mm² equating to 112 fields of view (FOVs) in $App^{NL-G-F}$ mice and age-matched controls (**see Methods**, Fig. 6a, b). The CosMx™ system uses protein labelling (DAPI, 18 s RNA, Histone and GFAP) to support cell segmentation, combined with spatial

transcriptomics and mapping to amyloid pathology (via monoclonal Aβ antibody MOAB-2)[64] (**see Methods**). Using this approach, in combination with well-established cell-type markers[65–67], we were able to cluster and identify major cell types which expressed a laminar expression profile in line with previous reports[67,68] (Fig. 6b, c and Supplementary Fig. S6a, Supplementary Data S6, S12). We first tested for differentially expressed genes (DEGs) at identified cell types between $App^{NL-G-F}$ and WT mice at 3-4 m and 6-8 m (Fig. 6d and Supplementary Fig. S6b-c) during a period of increasing amyloidosis (Supplementary Fig. S6d). In agreement with previous work using the CosMx™ system[68], we found evidence for the greatest number of DEGs at microglial cells across all tested ages (Supplementary Fig. S6b, c and Supplementary Data S12). Using over-representation analysis (ORA), we found DEGs at microglia to be associated with synaptic regulation in $App^{NL-G-F}$, with ligand-receptor interaction analysis finding most prominent interactions between microglia and excitatory neurons (**see Methods**, Supplementary Fig. S6e–h). For neuronal cells, GABAergic neurons showed the greatest number of DEGs at 3-4 m (Fig. 6d and Supplementary Fig. S6b), supporting early-stage transcriptomic changes in these cells. At 6–8 m, both excitatory and inhibitory neurons showed similar levels of DEGs (Fig. 6d and Supplementary Fig. S6c). Consistent with the functional uncoupling we observe in in vivo recordings (Fig. 1q), we also found a decrease in the strength of putative ligand-receptor interaction probability between inhibitory and excitatory neurons in $App^{NL-G-F}$ mice at this age (Supplementary Fig. S6h).

Our synaptic measures had identified early-stage GABAergic weakening to occur at population-coupled boutons (Fig. 3l and Supplementary Fig. S3h). Therefore, we next tested whether DEGs were focused at a particular GABAergic cellular subclass (Fig .6e, Supplementary Fig. S6i and Supplementary Data S6 and S12), as different GABAergic subclasses are known to express varying degrees of population coupling and state modulation[69]. For instance, PV neurons are thought to play a central role in modulating population activity[69]. We found the greatest level of DEGs at 3-4 m were focused at PV neurons, followed by SST neurons, with much lower levels in LAMP5 and VIP neurons (Fig. 6f, Supplementary Fig. S6j and Supplementary Data S6). We next conducted an ORA[70] to investigate the putative functions of DEGs at GABAergic subclasses (**see Methods**). At 3-4 m, we found enriched biological processes exclusively in the PV population (Fig. 6g and Supplementary. Data S6). Relevant enriched categories were mostly associated with synaptic processes, including the downregulation of transcripts linked to GABAergic receptors and transmission (*Gabra2*, *Gabrb2*, *Nf1*, *Pten*), and the upregulation of genes involved in GABA synthesis (*Gad1*, *Glul*), synaptic vesicle cycle (*Apoe*, *Bin1*, and *Sncb*), and calcium signalling (*Atp2b2* and *Calm1*) (Fig. 6g and Supplementary Data S6). At 6-8 m, changes across GABAergic subclasses were more uniform, so that all subclasses showed a similar fraction of DEGs (Fig. 6h and Supplementary Data S6) and included common biological processes associated with the regulation of synaptic plasticity, membrane potential, vesicle-mediated transport in synapses, and dendrite development (Fig. 6i and Supplementary Data S6). These included upregulation of multiple transcripts linked to voltage-gated calcium channels (*Cacna1a*, *Cacna1b*, *Cacna1c* and *Cacna1d*) and glutamatergic receptors and transmission (*Gria1*, *Gria2*, *Gria3*, *Gria4*, *Grin2a Grin2b*, *Grik2*, *Grip1*, *Grm5*, *Grm7*, *Grm8*, *Nrg2*, *Syt1*), as well as downregulation of transcripts linked to GABAergic receptors and transmission (*Calb1*, *Cck*, *Cckbr*, *Chrna7*, *Gabra2*, *Gabrb2*, *Gphn*, *Pvalb*, *Slc32a1*, *Sst*, *Sstr2*, *Sstr4*, *Vipr2*) (Fig. 6i and Supplementary Data S6). Together, these results suggest that early synaptic processing-related changes in PV neurons are followed by more widespread alterations across all tested GABAergic subclasses.

We next tested the extent to which transcripts from GABAergic subclasses covaried with local amyloid pathology. To do this, we pooled age groups and quantified amyloid plaque load for each FOV to define regions with low, medium, and high plaque load (**see Methods**, Fig. 6j, k).

We found that all GABAergic subclasses showed increased levels of DEGs with high plaque load (Fig. 6k). However, PV and SST interneurons were associated with more than double the percentage of DEGs that covaried with regional plaque load compared to VIP and LAMP5 subclasses (Fig. 6l, Supplementary Fig. S6k–m, Supplementary Data S6, S12). In PV interneurons, DEGs covarying with plaque load were linked to processes such as the synaptic vesicle cycle and vesicle-mediated transport (*Atp6v0a1*, *Atp6v1a*, *Bin1*, *Dnm2*, *Sncb*) (Supplementary Fig. S6n and Supplementary Data S12). In SST interneurons, processes associated with DEGs that covaried with plaque load included postsynaptic membrane potential and synaptic plasticity regulation (*Dgki*, *Gria1*, *Gria3*, *Grin2a*, *Pten*, *Rims2*) (Supplementary Fig. S6o and Supplementary Data S12). Taken together, our results suggest that PV interneurons show some of the earliest transcriptomic changes in cortical regions with dysregulated activity in $App^{NL-G-F}$ mice, which are linked to a downregulation of GABAergic synaptic transmission. However, both PV and SST interneurons show the highest number of transcripts that covary with local plaque load, which are associated with synaptic processes.

## Discussion

We investigated synaptic pathophysiology during early-stage amyloidosis and tested the degree to which it occurs globally or in specific subsets of synapses. GABAergic presynaptic axonal boutons exhibited early-stage molecular, structural, and functional weakening in regions with dysregulated activity in $App^{NL-G-F}$ mice, whilst excitatory dendritic spines showed more modest changes. Functional synaptic weakening was most evident for subsets of GABAergic boutons and dendritic spines that were strongly coupled to population activity and located near plaques. In contrast, synaptic compartments proximal to plaques but with low levels of population coupling had similar activity to WT controls. Synaptic changes occurred in concert with abnormal oscillatory activity and increased resting-state activity, which was associated with reduced VGAT puncta at c-Fos-positive excitatory cells, as well as a functional uncoupling of both E-I assemblies and putative GABAergic baskets in vivo. Using spatial transcriptomics, we found early changes associated with GABAergic synaptic transmission in PV interneurons. Both PV and SST interneurons exhibited signs of hyperactivity in imaging data and had the greatest number of DEGs that covaried with local plaque load. These DEGs were also associated with synaptic processes, suggesting abnormal synaptic regulation in these GABAergic subclasses during amyloidosis. We therefore propose that early-stage synaptic pathophysiology in $App^{NL-G-F}$ mice is not a global event but driven by a subset of strongly population-coupled GABAergic presynaptic inputs in the amyloid microenvironment, with molecular measures implicating abnormal synaptic processing at early stages in PV interneurons.

### Dysfunction of population-coupled synaptic compartments

Synapses exhibit a heterogeneous molecular composition, which is thought to support specialised functionality[4,69]. However, much of the work investigating amyloid-associated dysregulation lacks single-synapse resolution and is therefore unable to capture functional heterogeneity[6,30]. We recorded calcium imaging data from individual GABAergic axonal boutons and excitatory dendritic spines in vivo in $App^{NL-G-F}$ mice, and found synaptic compartments with strong population coupling showed the most dysregulated activity when proximal to plaques. Population-coupled synapses may be vulnerable to dysfunction because they come from cellular populations that regulate network activity (e.g., PV neurons)[69], which may be sensitive to amyloid-related processes[71]. Alternatively, synaptic compartments with strong levels of population coupling may have a molecular composition that is more sensitive to amyloid toxicity[4,36]. The micro-circuit characteristics of neurons innervated by population-coupled synapses may also impact their vulnerability. For example, in vivo calcium imaging and

electrophysiology have shown that cells with strong levels of population coupling are associated with more synaptic inputs and larger changes in activity following optogenetic or sensory modulation[61,72]. In contrast, weakly coupled cells are more resistant to population-level changes in activity[61]. Highly active neurons are also thought to produce the most amyloid[73]. Therefore, it is possible that early-stage amyloidosis induces global changes in activity which are primarily propagated through strongly coupled cells and synaptic connections. This could then lead to local increases in soluble amyloid levels, perpetuating a vicious cycle of amyloid-driven neuronal hyperactivity[73]. Homeostatic plasticity processes, which are known to be impaired in AD models[74,75], have also been suggested to localise to population-coupled synaptic compartments[62]. Future work may use repeated longitudinal in vivo approaches to track the activity of individual GABAergic boutons across AD-related pathology. This approach may test if population-coupled boutons with weaker activity are ultimately lost and investigate if surviving boutons show homeostasis-like (e.g., increases in size/strength[76]) or maladaptive responses.

### Amyloid-associated cortical circuit dysregulation

Previous work investigating AD-related pathophysiology has shown that periods of neuronal hyperactivity are followed by hypoactivity in both excitatory and inhibitory cells[8,9,33,77]. However, the synaptic-level alterations that map to changes in neuronal and micro-circuit activity remain unclear[27,30]. Early work using amyloid overexpression models reported decreases in excitatory synaptic function[26–28]. Recent studies in *App*[NL-G-F] mice[78,79] and younger APP/PS1 mice[80] found evidence for impaired glutamate reuptake and increased synaptic excitation. In agreement, we observed increased calcium-event frequency in excitatory dendritic spines. At later timepoints, this increase in frequency occurred alongside reductions in event amplitude, resulting in a net decrease in spine activity (AUC). The greatest reductions in spine activity occurred in population-coupled spines proximal to plaques. This suggests that both excitatory synaptic hyper- and hypo-activity may co-exist during different stages of amyloidosis. However, our data suggest that such excitatory synaptic changes are preceded and exceeded by reductions in GABAergic bouton activity, potentially disrupting the local excitatory/inhibitory balance and contributing to microcircuit hyperactivity. We observed VGAT reductions around c-Fos-positive excitatory cells together with a functional uncoupling of E-I assemblies and GABAergic baskets, supporting impaired inhibition as an early mediator of amyloid-associated hyperactivity[81–84].

AD-related pathology has been associated with the loss and dysfunction of dendritic spines[85–89] and axonal boutons[38,90–93], as well as asymmetries in pre- and post-synaptic dynamics[94]. We found loss of excitatory and inhibitory presynaptic markers across all studied brain regions, supporting the earlier vulnerability of this compartment. Consistent with previous work[27,80,95], PSD-95 loss and functional alterations at excitatory spines were also observed in our data, although less pronounced. Importantly, we only investigated a small fraction of synaptic proteins[4,96], so future studies with greater multiplexing capability may reveal additional synaptic vulnerabilities. Nevertheless, our measures of synaptic puncta, structure, and function suggest presynaptic GABAergic vulnerability as an early feature of amyloid-associated pathophysiology in line with previous work[97,98]. This conclusion is also supported by work in other knock-in mouse models and human *postmortem* samples, showing reductions in presynaptic density and function, as well as Aβ accumulation at inhibitory boutons[34,78].

Multiple cellular and micro-circuit changes in our data correlated with amyloid plaque load. For example, our spatial transcriptomics analysis identified molecular processes in PV and SST cells that covaried with regional plaque load. In contrast, other inhibitory subclasses, such as VIP and LAMP5, had far fewer DEG signatures that covaried with amyloid pathology. In addition, increased activity of both excitatory and inhibitory neurons, as well as reductions in GABAergic bouton density and calcium activity, were more evident in regions with higher plaque levels. Recent work has shown no changes in activity in superficial layers (L1-4) in *App*[NL-G-F] mice at 1.5 m, when plaque levels are very low in this mouse model[31]. Consistent with this, we found that mice with low plaque load were similar to WT, suggesting an early-stage resilience which may involve homeostatic processes[40,74]. However, it is important to note that we imaged at later timepoints and exclusively in regions where plaques were evident, to specifically study the impact of the plaque micro-environment on micro-circuit activity. This is not the case in recent work, which focused on pre-plaque periods[31]. Others have reported synaptic deficits in similar knock-in models in the entorhinal cortex by 2 m[78], suggesting that dysfunctional synaptic activity may be detectable during very early stages of amyloid-beta pathology. Taken together, our distance and plaque density measurements suggest an association between micro-circuit dysregulation and amyloid pathology. However, this does not imply causation, and further work is needed to determine whether the effects we observe are directly driven by soluble or aggregated forms of Aβ, or whether other mechanisms, potentially initiated by Aβ, or independent of it, are involved[99].

### Cell-type-specific synaptic vulnerability

Traditionally, excitatory synapses have been regarded as the most susceptible to AD-related pathology[27,80], but it is now increasingly evident that early-stage GABAergic dysfunction plays a central role in disease development[6,19,29,30,98,100,101]. The nature and locus of this vulnerability is less clear[30]. Our data suggest an increased vulnerability of population-coupled presynaptic GABAergic axonal boutons. Our spatial transcriptomic analyses found that, although DEGs are detectable across different inhibitory populations, some subclasses, such as PV and SST interneurons, show greater changes in initial disease stages and stronger putative molecular covariation with plaque-rich environments. Our data suggest that early molecular changes associated with synapse dysfunction may occur first in PV cells, but that cellular hyperactivity is likely a feature of both PV and SST interneurons during amyloid-associated pathology. Several studies have implicated PV- and/or SST-positive interneurons in early-stage amyloidosis[30,31]. However, their involvement appears to be brain region- and timepoint-dependent, with evidence pointing to a complex interplay of potentially compensatory and pathogenic roles across disease progression[77,102–105]. In contrast, LAMP5 and VIP cells showed lower levels of DEGs that covaried with plaque load. Interestingly, recent work has suggested that LAMP5 and VIP neurons are more active during certain brain states (e.g., locomotion), whilst PV interneuron activity better maps to resting-state synchronised oscillations[69]. Thus, one possibility is that early-stage PV interneuron dysfunction may contribute to amyloid-associated hyperactivity by failing to modulate resting-state activity levels[71]. In support of this, dysfunction in PV interneurons associated with network, plasticity, and memory impairments has been widely linked to AD[30,71]. However, interneuron subclasses (such as PV cells) can be further divided into diverse subtypes, which are sensitive to specialised brain-state modulation[69,106]. Therefore, determining how disease processes drive synaptic pathophysiology across such diverse neuronal populations will require deeper synaptic and cellular classification[5,69]. In addition, further studies are needed to establish rigorous theoretical models that can capture and describe healthy synaptic activity in vivo. For example, although the population coupling metric we used is a well-established and useful measure[61,62], it is a deliberate oversimplification of population-level

dynamics. Therefore, more advanced computational approaches[107] may be necessary to capture the functional heterogeneity of synaptic populations across diverse brain states (e.g., sensory processing, learning, sleep). For instance, previous work has found that circuit dysfunction in AD models can be influenced by anaesthesia[108]. Here, we found hyperactivity in posterior cortical regions (VIS and RSC) to be present in both awake and anaesthetised conditions. In contrast, we did not find robust evidence of hyperactivity in frontal regions of anaesthetised or awake $App^{NL-G-F}$ mice. Further work is required to understand the extent to which brain-state and behavioural features modulate AD-related pathophysiology.

## Possible molecular mechanisms of synaptic and micro-circuit vulnerability

Our data suggest that, while interneuron somas show signs of hyperactivity in $App^{NL-G-F}$ mice, activity levels in GABAergic boutons decrease with time and proximity to plaques. These changes in activity can be reconciled by considering bouton-level event distributions, which suggest that, while inhibitory cells show greater calcium activity in $App^{NL-G-F}$ mice, these somatic events may recruit small-amplitude synaptic events at downstream GABAergic boutons. Future work may investigate the underlying mechanisms. For example, presynaptic calcium channels and/or other synaptic machinery may be directly or indirectly disrupted by amyloidosis[4]. In support of this, we found early-stage changes in molecular processes involving GABAergic synaptic transmission at PV interneurons, suggesting abnormal synaptic processing in this GABAergic subclass during early amyloidosis[4,6]. We found DEGs linked to GABAergic synaptic transmission (*Gabra2, Gabrb2, Nf1, Pten*), GABA synthesis (*Gad1, Glul*), synaptic vesicle cycle (*Apoe, Bin1, Sncb*) and calcium signalling (*Atp2b2, Calm1*). Impairments in the expression and function of some of these molecules have previously been reported in mouse models and AD patient tissue, and their pharmacological modulation has shown therapeutic potential[30,109,110]. Further work is required to investigate how the early molecular changes that we highlight link to the downregulation of GABAergic synaptic protein expression and, critically, in vivo synaptic activity. We measured in vivo synaptic activity separately and could not definitively link changes to any specific inhibitory neuronal subclass. Such work may focus on PV and SST interneurons, as these subclasses showed the greatest number of DEGs that covaried with local amyloid load. In addition, studies using unbiased multi-omics approaches and synaptic compartment-specific analyses will be crucial to further elucidate the nature of the mechanistic processes involved[111]. A major technical challenge, however, lies in linking in vivo functional recordings with molecular measures at the single-synapse level, and understanding the evolution of such changes over time, which can be challenging in cross-sectional studies.

We propose that early-stage synaptic pathophysiology in $App^{NL-G-F}$ mice is not a global event. Instead, it is driven by a selective vulnerability of strongly population-coupled GABAergic presynaptic inputs in the amyloid microenvironment, leading to a failure to modulate resting-state network activity. Our data suggest that PV and SST interneurons show the greatest differential gene expression, with transcriptomic signatures relating to abnormal synaptic function during amyloidosis. Understanding how brain state-specific synaptic pathophysiology maps to synaptic molecular composition in conditions of AD-related pathology may support more targeted interventions in the future.

## Methods
### Animals
Experiments were conducted according to the United Kingdom Animals (Scientific Procedures) Act 1986 and were approved by the Home Office under the project licence PP8529297. We used adult (P90-P240) animals male and female $App^{NL-G-F}$ homozygous mice, which carry a humanised amyloid β (Aβ) sequence with the Swedish (KM670/671NL), Beyreuther/Iberian (I716F), and Arctic (E693G) [E22G in humanised Aβ] mutations, in the mouse *App* gene[42] on a C57BL/J6 background. Age-matched C57BL/J6 mice served as controls for all experiments other than 1-photon mesoscopic imaging. For these experiments, $App^{NL-G-F}$ mice were crossed with *Thy-1-GCaMP6s*[112] mice (RRID: IMSR_JAX:024275), and so the *Thy-1-GCaMP6s* line served as controls. Mice were matched for sex and age within experimental groups and housed in a temperature (21 ± 2 °C) and humidity (55 % ± 10 %) controlled facility on a 12 h light/dark cycle. Imaging data were matched for time within that animal's light cycle. All mice had access to water and food ad libitum.

### Clear skull preparation
We used a skull optical clearing technique[44] in which mice were anaesthetised with 3 % isoflurane followed by intraperitoneal (i.p.) injection of ketamine (0.10 mg/g) and xylazine (0.01 mg/g), and subcutaneous injection of carprofen (5 mg/kg), buprenorphine (0.1 mg/kg) and dexamethasone (1.26 mg/kg). Fur was removed from the scalp using hair removal cream and the skin cut away to expose a large section of the skull. The skull was thinned using a scraper, and a thin layer of cyanoacrylate glue was applied directly to the skull surface to facilitate optical clearance. A protective layer of clear nitrocellulose was applied on top of the glue layer. A custom-made head-bar was then affixed to the skull using dental cement. Following head bar placement, burr holes were drilled ±3.1 mm lateral to lambda (to target binocular V1). For superficial recordings, a tungsten recording electrode was implanted at coordinates [+3.1 ML, −0.3 AP, −0.1 DV] at a -30-degree angle. For the reference electrode, a burr hole was drilled at +3.3 anterior to bregma, and −1.5 to −2.1 lateral, and a silver wire was placed at the brain surface. All electrodes were fixed in place using super glue (ethyl cyanoacrylate). Gold pins and the head-bar were fixed in place with dental cement. Mice recovered for a minimum of two weeks ahead of imaging.

### Mesoscopic calcium imaging
Widefield calcium imaging was conducted as before[44] using a custom-built tandem lens epifluorescence mesoscope. Imaging was conducted using the *Thy-1-GCaMP6s* line[112] either alone or crossed with $App^{NL-G-F}$ mice[42]. In the *Thy-1-GCaMP6s* line, deep layer 5 excitatory neurons and superficial layer 2/3 excitatory neurons express GCaMP6s[112]. 1-photon widefield mesoscopic imaging is restricted to the most superficial areas of the cortex. As such, our 1-photon mesoscopic imaging experiments are likely sampling population-level signals from L2/3 excitatory neurons and the dendrites of L5 neurons that project into superficial layers of the cortex.

Excitation light from two LEDs (470 nm (M470L3, Thorlabs, with excitation filter DC/ET470/40x, Chroma), and 405 nm (M405L3, Thorlabs, with excitation filter DC/ET405/10x, Chroma)) was combined using a beam-splitter (DC/ T425lpxr, Chroma) and delivered through a dichroic mirror (DC/ET525/50 m, Chroma). Images were acquired through an emission filter (DC/ET525/50 m, Chroma) using a CMOS camera (Grasshopper3 GS3- U3-23S6, FLIR) at 50 Hz. The two excitation wavelengths were temporally interleaved at a frequency of 25 Hz each. For data shown in Fig. 1, 4 × 30 s trials of resting-state activity were collected in the dark for each animal and timepoint under continuous 0.5 % isoflurane anaesthesia. For data shown in Fig. 5, five minutes of resting-state activity in the dark were collected in awake mice placed in a restraining tube, with 4 × 4 pixel binning to result in a 288 × 300 frame.

### Mesoscopic and EEG signal analysis
Mesoscopic imaging data were processed as before[44]. Acquired time-series stacks were full-frame registered to an average intensity projection of both channels recorded at the first imaging session using motion correction software[113] in ImageJ (US National Institute of

Health, v1.53t). Epochs with large movement-related artifacts (where the signal exceeds 5x the RMS of the entire signal) in the EEG recordings were excluded from analysis. Subsequent analysis used custom-written MATLAB (Mathworks, R2021b) code. For recordings made under anaesthesia, stacks were manually aligned to an Allen Mouse Brain Atlas template (https://atlas.brain-map.org/) using the posterior sutures, bregma, and olfactory bulb as alignment points. Custom masks were generated corresponding to distinct functional cortices in the mouse (Fig. 1c). Channels were deinterleaved into 405 nm and 470 nm channels and separately bleach corrected by subtracting a temporally smoothed trace generated for each pixel with a window width of 3 s. The 405 nm channel was subtracted from the 470 nm channel to remove hemodynamic absorption signals. For data collected under anaesthesia, stacks were downsampled in $2 \times 2$ bins using bilinear interpolation. Images in awake conditions were already binned during acquisition and therefore did not require additional downsampling. The median value ($F_O$) for each pixel was then subtracted to generate $\Delta F$ and the product divided again by $F_O$ to give $\Delta F/F_O$. Large regions of interest were then drawn around brain areas (VIS, RSC, SS, MO) and the average $\Delta F/F_O$ signal was extracted. Peak detection was used to estimate amplitude, frequency, and total activity levels (frequency x amplitude). For simultaneous measures of mesoscopic calcium events and temporally associated EEG recordings, we used an event triggered averaging approach. First, calcium events were identified from a ROI proximal to the EEG recording electrode in the visual cortex using peak detection (MATLAB) and a threshold that was twice the average $\Delta F/F_O$ activity across the entire recording epoch. Identified calcium events were then used to define temporally associated electrophysiological data in a 4 s window that was 2 s before and 2 s after the calcium event onset. The baseline and rising phases were defined as −600 to −100 ms and −100 to 400 ms relative to the event onset, respectively. For Fig. 5l, we performed wavelet convolution[114] on each event separately and then computed the power across all events for each animal. We then averaged the power values for each frequency band across animals. For spectral analysis in Fig. 5m–o, power spectral densities (PSDs) were computed from 500 ms epochs during baseline and the rising phase using Welch's method (0.1-s Hann, 25 % overlap), cropped to 1–80 Hz for all events across all animals. Across all events, the difference PSD was calculated by subtracting the baseline PSD from the rising phase PSD.

## Craniotomy and viral construct delivery

Cranial windows were surgically implanted over the right hemisphere of the monocular visual cortex as described previously[39,44,76,115,116]. Briefly, mice were anaesthetised via an i.p. injection of ketamine/xylazine (0.1 mg/g and 0.01 mg/g, respectively) or 1.5 % isoflurane. This was followed by a subcutaneous injection of the analgesics carprofen (5 mg/kg) and buprenorphine (0.1 mg/kg), and the anti-inflammatory dexamethasone (1.26 mg/kg). Following a craniotomy, viral constructs were injected using a sharp glass micropipette. Where described in the text, mice were injected with: AAV1-c-Fos-EYFP (Vigene Biosciences, $1.67 \times 10^{13}$ GC/ml), AAV1-S5E2p-cyRFP-GSG-P2A-HIS-GCaMP6s-WPRE (Vector Biolabs, 1.98 ×1013 GC/ml), AAV1-EF1a-mCherry-GSG-P2A-HIS-GCaMP6s-WPRE (Vector Biolabs, 7.0 ×1012 GC/ml) and/or AAV1-mDlx-Kz-f-cyRFP1-GSG-P2A-GCaMP6S-WPRE-pA (Vigene Biosciences, $8.97 \times 10^{12}$ GC/ml) into superficial layers of the motor, visual, and/or the retrosplenial cortex. Following virus delivery, the skull was replaced with a double glass coverslip, and a head-bar was attached to the bone with dental cement. Mice recovered for a minimum of two weeks ahead of imaging.

## 2-photon structural/functional imaging

Functional calcium cellular and synaptic imaging was performed on a custom built 2-photon microscope (INSS) with a resonance scanner (Cambridge Technology) and a high-power objective Z-piezo stage

(Physik Instrumente), using a Chameleon Vision II laser (Coherent) set to 920 nm and a water immersion 16x, 0.8 NA, 3.0 mm WD objective (Nikon), as before[39]. Data were acquired with an 800 M Hz digitiser (National Instruments) and pre-processed with a custom-programmed field programmable gate array (FPGA) (National Instruments). The average laser power delivered to the brain was < 50 mW. Scanning and image acquisition were conducted using Scanimage freeware (Vidrio-Tech). Image acquisition parameters for all experiments were: 512 × 512 px$^2$, $130 \times 130 \mu m^2$, 30 Hz. For functional 2-photon imaging of both excitatory and inhibitory neurons and synaptic compartments, the excitation wavelength was set to 920 nm to excite GCaMP6s in both cellular populations. At an excitation wavelength of 920 nm, cyRFP (only expressed in inhibitory neurons) is also excited. Emitted light from the cyRFP fluorophore was filtered at an ET605/70 m bandpass filter and collected at a second PMT. When relevant, the excitation wavelength was set to 1080 nm at the end of the imaging session in order to excite the mCherry fluorophore (only expressed in excitatory neurons)[49]. For both cellular and synaptic imaging, three trials ($3 \times 85$ s) of resting-state activity were measured in the dark under constant 0.5 % isoflurane anaesthesia. Data were collected at 3-4 m and 6-8 m for separate cohorts of mice to mitigate risks relating to virus overexpression.

## Cellular calcium signal extraction and analysis

Cellular functional calcium imaging data were processed using ImageJ and custom-written MATLAB code as previously described[39,44,46,58,117]. First, a 15-frame running z-projection was generated from each time series, and mean 2 x 2 x 2 filter applied. Images were then full-frame registered using the motion correction software 'Moco' in ImageJ[113]. Cellular and background regions of interest (ROIs) were then drawn using ImageJ by at least three experimenters, blind to the experimental condition and timepoint of imaging. Extracted signals were processed in custom-written MATLAB code to remove slow signal changes in raw fluorescence traces. Cellular calcium traces were then normalised to the background fluorescence signal, and a 25 % threshold was applied. A peak detection algorithm was used to estimate the frequency and average amplitude of calcium events. For both excitatory and inhibitory neurons, calcium-mediated neuronal activity was estimated as the area under the curve of the $\Delta F/F_O$ trace as in previous work[44,46,58,117]. For animal-level analyses, each mouse's median activity was calculated across all imaged regions and normalised to the age-matched WT group median. For distance analyses, measures were z-scored for each cell using the mean and standard deviation of the age-matched WT distribution. E-I neuronal assemblies were estimated using positive and significant pairwise correlations between thresholded calcium signals based on previously published approaches[39,46,48,51,117]. For each neuron of interest, network size was estimated as the fraction of correlated neurons out of all neurons in each cortical region[46]. In Supplementary Fig. 1u, v, calcium events were binned by amplitude (bin size: 0.5 $\Delta F/F_O$), and their fractions were calculated relative to the total number of events per imaged region. For PV-specific data analysis, measures were normalised to the mean of the WT cell distribution.

## GABAergic bouton signal extraction

Data were analysed blind to experimental condition by at least two researchers. First, a 20-frame running z-projection was generated from each time series, and mean 2 x 2 x 2 filter applied. Recordings were full-frame registered in ImageJ using 'Moco'[113] and manually aligned to each other. Stacks were visually inspected for movement or artefacts and discarded if movement was detected in the z-axis. Axonal sections were excluded if there was evidence of crossing axons or dendrites. Swellings along the axon which were present in both channels (cyRFP and GCaMP6s) were identified as active boutons. Circular ROIs were drawn around boutons and immediately adjacent axons. Traces from each

bouton and axon were extracted and analysed using custom-written MATLAB code. Traces were normalised to the background fluorescence (300 frame running average) to calculate $\Delta F/F_0\_bouton$ and $\Delta F/F_0\_axon$ (Supplementary Fig. S3a). To isolate bouton-specific responses, the $\Delta F/F_0\_axon$ was subtracted from each $\Delta F/F_0\_bouton$, and a 20% threshold was applied (Supplementary Fig. S3a). Bouton activity was z-scored to the mean and standard deviation of the age-matched WT population. Axonal stretches (>10 μm) visualised in the cyRFP channel were used to quantify bouton density. 1-pixel-thick lines were drawn over each axonal stretch, and their fluorescence intensity profiles were used to estimate bouton numbers using a peak detection algorithm in MATLAB.

## Spine signal extraction

Spine signal extraction was performed as previously described[39,44,49,59]. All data were analysed blind to time and experimental condition by at least three independent analysts. To estimate calcium-mediated dendritic spine activity, individual single time-series stacks were first full-frame registered using the motion correction software 'Moco' in ImageJ[113]. Stacks were then visually inspected for movement in the z-axis and discarded if a stable dendrite was not evident. Dendritic sections were excluded if there was evidence of crossing axons or dendrites. Calcium responses from individual spines were isolated from global dendritic signals using a subtraction procedure described previously[44,49,59]. To measure spine signals, circular ROIs were first drawn over individual dendritic spines to measure spine fluorescence and compute $\Delta F/F_0\_spine$. Circular ROIs of the same size were then drawn around the adjacent parent dendrite to calculate the local dendritic signal, $\Delta F/F_0\_dendrite$. Spine signals were isolated by subtracting the dendritic signal from the spine signal, and a 25% $\Delta F/F_0$ threshold was applied.

## Synaptic activity analysis

A peak detection algorithm was used to estimate the frequency and average amplitude of calcium events. Total activity was measured as the area under the curve of the $\Delta F/F_0$ trace. To enable pooling and comparisons, values were z-scored within each experimental group using the mean and standard deviation of the corresponding WT distribution. In Supplementary Fig. 3d, boutons with negative and positive z-score values were considered to have reduced and increased activity, respectively. Axons containing a combination of boutons with negative and positive z-scored values were classified as "mixed". Population coupling scores based on previous work[61] were calculated for each bouton/spine as follows. The average signal was calculated from all bouton/spine-specific $\Delta F/F_0$ traces in each imaged region, and a Pearson correlation value was calculated between this average signal and the bouton/spine-specific $\Delta F/F_0$ trace for each individual bouton/spine in that region. In Supplementary Fig. 3f–h, low-activity boutons (AUC: z-score< 0) were identified, and their calcium events were classified as small (<0.45 $\Delta F/F_0$), medium (0.45–0.65 $\Delta F/F_0$), or high (>0.65 $\Delta F/F_0$). Their relative fractions were then calculated based on the total number of calcium events per bouton.

## Basket signal extraction and analysis

Inhibitory baskets around excitatory cells were visually identified, and circular ROIs were placed around the central excitatory cell and adjacent putative inhibitory boutons. Bouton traces were normalised to the background fluorescence (300 frame running average), and a 15% threshold was applied to the $\Delta F/F_0$ signal. The activity from each excitatory cell was calculated as described in **Cellular calcium signal extraction and analysis**. Each bouton's signal was compared to the excitatory cellular activity to calculate a Pearson correlation coefficient. Events in the excitatory cell were defined as peaks with a >0.25 prominence in the normalised $\Delta F/F_0$ signal. Bouton signals were temporally aligned to excitatory cellular events and 2 s epochs (1 s before to 1 s after each peak) were isolated for further analysis.

For each bouton, the mean amplitude of its responses was calculated from the average signal of all its epochs. Excitatory cellular frequency and GABAergic bouton amplitude values in Fig. 5h were z-scored using the mean and standard deviation of the age-matched WT distribution.

## Amyloid-beta plaque quantification in vivo

At the end of each imaging session, mice were injected (i.p.) with methoxy-X04 (8% vol of 10 mg/mL methoxy-X04 in DMSO (≥99.9%, Sigma-Aldrich) and 15.4% vol Kolliphor®EL (Sigma-Aldrich) in 76.6% vol sterile saline (0.9%, Thermo Scientific)). The following day, methoxy-X04 signal was imaged at 720 nm. To estimate amyloid load, plaques were manually identified based on size and morphology[50,118]. Subsequent immunofluorescence measures (**see Immunofluorescence**) showed that more than 70% of MOAB-2 positive plaques were also labelled by methoxy-X04 in all analysed mice and brain regions. Distances were measured from the centre of each cell/synapse to the edge of its nearest methoxy-X04 labelled plaque. For animal-level analyses, the median plaque density for each mouse was calculated from the number of plaques detected across all imaged regions.

## Immunofluorescence

Animals were terminally anaesthetised (ketamine: 0.1 mg/g; xylazine: 0.01 mg/g i.p.) and transcardially perfused using 10 mL of oxygenated ice-cold artificial cerebrospinal fluid (108 mM choline-Cl, 3 mM KCl, 26 mM $NaHCO_3$, 1.25 mM $NaHPO_4$, 25 mM D-glucose, 3 mM Na pyruvate, 2 mM $CaCl_2$ and 1 mM $MgCl_2$ saturated with 95% $O_2$ / 5% $CO_2$). Brains were post-fixed in 4% paraformaldehyde overnight at 4 °C and washed in 1xPBS. Following immersion in a cryoprotectant solution (30% sucrose and 0.1% $NaN_3$) for at least 48 h, a freezing microtome was used to obtain 40 or 80 μm-thick coronal sections containing V1, RSC, and ENT. A similar protocol was followed for all immunolabelling (Supplementary Data S13, S14). Sections were washed in buffer solution and, when applicable, incubated in blocking solution for 1 h at room temperature (RT). Primary antibodies were incubated overnight at 4 °C and washed in a buffer containing 0.25% Triton-X100. Secondary antibodies were incubated for 2 h at RT. For methoxy-X04 staining, sections were washed in buffer, incubated at RT for 30 mins with the dye, and washed again in a buffer containing 0.25% Triton-X100. Samples were mounted on microscope slides using Vectashield® Non-Hardening Mounting Media and stored at 4 °C for at least 24 h prior to imaging. An SP8 Lightning confocal microscope and the DMI8 platform from Leica Microsystems were used for imaging. Diode 405 & 638 and OPSL 488 & 552 lasers were combined as necessary, together with hybrid detectors or photomultiplier tubes. For each set of experiments, specific HC PL APO CS2 objectives (Leica), dimensions, digital zoom, and z-steps were used to acquire z-stacks of: c-Fos/MOAB-2/methoxy-X04 co-stain (40x/1.30 oil, 1040×1040 px², 291.42 × 291.42 μm², 1%, z = 1 μm), c-Fos levels in PV/SST cells and S5E2 virus validation (20x/0.75 dry, 2048 × 2048 px², 580.67 × 580.67 μm², 1%, z = 1 μm), excitatory and inhibitory synaptic puncta (63x/1.40 oil, 2264 × 2264 px², 102.51 × 102.51 μm², 1.8%, z = 0.25 μm), and c-Fos-based activity tagging (40x/1.30 oil, 2760 × 2760 px² 387.5 × 387.5 μm², 0.75%, z = 0.5 μm). All images acquired by confocal microscopy were further processed and analysed using ImageJ software and custom-written scripts in MATLAB as described in the following sections.

## Quantification of amyloid-beta and activity markers in cortical regions

To assess whether local variations in amyloid deposition (MOAB-2, methoxy-X04) in motor and visual cortices corresponded to differences in neuronal activity levels (c-Fos), images were subdivided into nine 90 × 90 μm² regions. Mean fluorescence intensity was extracted for all markers and normalised to the median of the V1 distribution.

## Quantification of c-Fos in interneuron subclasses

Neurons expressing PV or SST (defined as ≥10 % above background) were identified, and circular ROIs were drawn around each cell. Mean c-Fos fluorescence intensity was extracted for every cell and normalised to the mean of the subclass-matched WT distribution.

## S5E2 virus validation

Neurons expressing cyRFP (defined as ≥10 % above background) were identified in both WT and $App^{NL-G-F}$ mice injected with the AAV1-S5E2p-cyRFP-GSG-P2A-HIS-GCaMP6s-WPRE and stained with an anti-PV antibody (Supplementary Data S13, S14). Circular ROIs were drawn around each cell and the mean PV fluorescence intensity was quantified for each ROI. Cells were considered PV-positive if fluorescence levels were ≥10 % above background.

## Quantification of synaptic puncta

Putative excitatory synapses were identified by measuring the colocalisation between vesicular glutamate transporter 1 (VGLUT1) and postsynaptic density protein 95 (PSD-95) (Fig. 2a), while vesicular GABA transporter (VGAT) and gephyrin were used for putative inhibitory synapses (Fig. 2b). In $App^{NL-G-F}$ mice, the image was centred around a methoxy-X04-labelled amyloid plaque (Fig. 2a, b). An average projection was generated from six consecutive z-slices, and four bisecting lines (one vertical, one horizontal, and two diagonal) were drawn to extract fluorescence intensity profiles. A custom-written MATLAB code was used to normalise each fluorescence intensity profile. A threshold was applied to identify regions with postsynaptic marker expression (i.e., PSD-95 and gephyrin) and generate a binary mask. This mask was superimposed to the presynaptic signal (i.e., VGLUT1 and VGAT), and points of colocalisation were identified using a peak detection algorithm in MATLAB. Pixel intensity values extracted from each marker (VGLUT1, PSD-95, VGAT, gephyrin) were also processed independently to estimate the density of pre-and post-synaptic puncta using a peak detection algorithm. Mean synapse/puncta density was estimated for each image as an average of its four bisecting stretches.

## Quantification of c-Fos and synaptic measures in virally injected brains

To estimate the levels of c-Fos expression in V1 and RSC, we quantified EYFP fluorescence intensity in virally injected brains. Pyramidal neurons exhibiting c-Fos expression (defined as 10 % above background) were identified and included in the analysis. Ten-pixel-thick lines were manually drawn around c-Fos-positive cells to extract EYFP fluorescence intensity profiles, which were normalised to background values (25th percentile) using custom-written MATLAB code. For each cell, c-Fos expression was estimated as the average pixel intensity in each normalised trace. The same 10 pixel-thick lines were also utilised to calculate the density of VGAT-positive presynaptic puncta colocalising with those cells. Fluorescence intensity profiles were normalised, and a peak detection algorithm was applied. VGAT density was calculated from the number of identified supra-threshold peaks (>20 %). EYFP-expressing dendritic stretches (defined as 10 % above background) were identified at various depths of the z-stacks (Fig. 5a, right). Ten-pixel-thick lines were then manually drawn to extract fluorescence intensity profiles, which revealed uniform c-Fos expression along the dendritic length (Supplementary Fig. S5a). These traces were processed as described above to quantify spine density and size, estimated as the number and average height of all supra-threshold peaks per dendrite. The lines were then superimposed on the parent dendrite, and the fluorescence intensity profile was used to calculate the average c-Fos intensity. To enable pooling and comparisons, c-Fos and synaptic measurements were z-scored using the mean and standard deviation of the corresponding region- and age-matched WT distribution.

## Spatial transcriptomics using CosMx™

We used CosMx™ spatial molecular imaging with the 950-plex CosMx™ mouse neuroscience panel to define transcriptional signatures at single-cell resolution. We sampled the VIS and RSC regions from 8 $App^{NL-G-F}$ mice and 8 age-matched WT controls of mixed sex. Mice were perfused as described in the **Immunofluorescence** section. Fresh frozen 10 μm-thick coronal sections were collected using a cryostat. The sample processing, staining, imaging, and cell segmentation for CosMx™ were performed as previously described[64,119]. Previously published protocols were followed for tissue staining and probe acquisition (https://university.nanostring.com/page/document-library). Markers for morphology and cell segmentation were used: Histone H3: nucleus, 18 s rRNA: cytoplasm, GFAP: astrocytes; DAPI: nucleus. The monoclonal Aβ antibody MOAB-2 was used on adjacent sections[120] (Supplementary Data S13, S14). Data were acquired from 16 coronal sections, and 129 FOVs were selected, making a total area of 63.5 mm². Each FOV was 4480 × 4480 pixels (753 × 753 μm) with a pixel size of 0.168 μm/px. For data processing and analysis, we used Seurat v5.1[121,122]. The cell quality check parameters involved filtering out any cells that contained fewer than 20 transcripts and any cells that had an area greater than 5*geometric mean of all cells in the dataset, which resulted in a dataset containing 178,285 cells. The transcript matrices, cell expression matrices, cell metadata, FOV position, and cell polygon files were read into a Seurat object, where data were integrated and normalised (SCTransform (v2)[123]). Transcripts from negative probes were excluded from the count matrices. To identify the main cell types, clustering analysis was performed following integration; RunUMAP[124] (pca, dims = 1:30, n.neighbours = 20, umap.method = uwot, learning.rate = 1, min.dist = 0.1, spread = 1, set.op.mix.ratio = 1, local.connectivity = 1, repulsion.strength = 1, negative.sample.rate = 5), FindNeighbors (pca, dims = 1:30), FindClusters (res = 0.2), FindAllMarkers (min.pct = 0.25, logfc.threshold = 1). Main cell types were identified using an adapted automated cell-typing using the Expression Weighted Celltype Enrichment (EWCE)[66] algorithm against a generated cell-type dataset (CTD) reference from the Allen Mouse Whole Brain Atlas[67]. Our CTD reference was filtered to contain only cell types specific to VIS and RSC regions and only transcripts included in the Nanostring panel (NCTD). As previously described, we also found that highly expressed transcripts in the CosMx data may sometimes be mis-segmented[68]. We took several steps to avoid false positives in our results. First, for each cell type, we used the highest expressed markers as a condition to exclude potentially mislabelled cells (i.e., *Gad1*, *Gad2* and *Slc32a1* for inhibitory cell types). Second, cells expressing (over 2*SD) top markers of other cell types identified using our NCTD reference were filtered out (based on the max specificity score; e.g., *C1qa*, *C1qb*, *C1qc*, *Ctss*, etc., expressing cells were excluded from the inhibitory (INH) cluster). However, a low expression of non-representative transcripts (i.e., *Gfap* or *Mbp* in inhibitory cells) were still identified (https://nanostring-biostats.github.io/CosMx-Analysis-Scratch-Space/posts/segmentation-error/). We then used the NCTD reference to filter out transcripts with a top specificity score for other cell types prior to running differential gene expression (DGE) comparison between our groups of interest. For the GABAergic cell (INH) subclass DGE analysis, we generated a subset of the data and re-clustered the cell types using the same steps as above (FindClusters; res = 0.1), which resulted in 5 cell types, of which 4 were identified as inhibitory subclasses using the EWCE[66] package: PV, SST, VIP and LAMP5. FOVs not covering VIS or RSC regions were excluded from this analysis, resulting in 9643 GABAergic cells. To identify transcriptional differences between $App^{NL-G-F}$ and WT mice, we run DGE analysis using the Model-based Analysis of Single-cell Transcriptomics (MAST) test[125]. The models were fitted separately for each age-group and cell type. Genes expressed in at least 20 % of cells (minimum of 2 counts per cell) were evaluated for gene expression. We applied a thresholding method

based on previously published results[126–130], specifically an adj. p < 0.05 and a log2 fold change (log2FC) > abs(±0.25). The percentage DEGs for each cell type was calculated from the total number of DEGs expressed in all the cells (Fig. 6d, f, h, k and Supplementary Fig. S6b, c). To map gene IDs and annotate them to a functional category, we used overrepresentation analysis (ORA, WebgestaltR package, 2024) with the geneontology, 'Biological Process noRedundant' (Fig. 6g, i) and 'Biological Process' (Supplementary Fig. S6h, i) databases, with the following parameters: 5/2000 min/max analytes/category, FDR < 0.05 (using the Benjamini-Hochberg (BH) method[70]). The top 5 enriched categories (based on FDR) were selected (Fig. 6g, i and Supplementary Fig. S6h, i). We calculated putative ligand-receptor (LR) interaction probabilities using CellChat[131]. The interaction strengths for microglia (MGL), excitatory (EXC) and inhibitory (INH) cell types were computed separately for each genotype and age group. The LR interaction changes between *App*^NL-G-F vs WT mice were then obtained by calculating the percentage difference in weights between the two genotypes (Supplementary Fig. S6e, f). We calculated plaque load based on the average number of plaques per FOV. Plaque load was defined as low, medium, and high based on the 33$^{rd}$ and 66$^{th}$ percentiles of the average plaque number distribution for each age group. Data from both age groups were combined, and DEGs by plaque load were calculated separately for each GABAergic subclass using MAST. We then selected transcripts which showed a monotonically increasing or decreasing fold change (Spearman's rho = ± 1) as a function of plaque load (Fig. 6l and Supplementary Fig. S6f, g) and used them to identify functional categories (Supplementary Fig. S6h, i).

### Statistics & reproducibility

Statistical analyses were performed in either MATLAB (version R2021b) or SigmaPlot (version 14.0, Systat Software), or R (version 4.3.1). Two-sided comparisons were made using parametric or non-parametric tests, as necessary: Student's t-test, Mann-Whitney rank sum *t* test, Welch's test, Chi-squared test, z-test, One-Way and Two-Way ANOVA with Holm-Šidák post-hoc test, or Kruskal-Wallis One-Way ANOVA with Dunn's test. Correlation was calculated with a Pearson correlation coefficient. Specific statistical tests and the number of samples used for all figures can be found in Supplementary Data 1–14. Sample sizes were based on power calculations and pilot work. The allocation of animals to groups was randomised, and analyses were conducted blind to the experimental conditions by two or more independent analysts.

### Reporting summary

Further information on research design is available in the Nature Portfolio Reporting Summary linked to this article.

## Data availability

Data associated with this paper has been deposited at https://doi.org/10.5281/zenodo.18369811 and at the Gene Expression Omnibus (GEO) database (GEO accession code: GSE318590). These are publicly available as of the date of publication. Due to large file size, other datasets generated in this study are also available from the author upon request. Source data are provided in this paper.

## Code availability

Code that supports the analysis is available at https://doi.org/10.5281/zenodo.18370193.

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

## Acknowledgements
We thank members of the Barnes Lab, Stergios Tsartsalis, Arjune Sen, Nir Grossman, Amy Smith, and Nurun Fancy for comments on the manuscript and helpful discussions. We also thank the Nanostring (Bruker) team, particularly Michael McKenna for support with the CosMx™ experiments and analysis. This work is supported by the UK Dementia Research Institute through UK DRI Ltd, principally funded by the Medical Research Council (to S.J.B). This work is also supported by grants from The Brightfocus Foundation (A2022030S to S.J.B.), The Wellcome Trust (to S.J.B. and S.S.), The Uren Foundation (to S.J.B.), The Rosetrees Trust (to S.J.B. and J.J.), The Epilepsy Research Institute (to F.A.C.) and by the Imperial College Healthcare Trust - NIHR Biomedical Research Centre (to P.M.M) C.I.R., A.M. and P.M.M. are also supported by the Edmond J. and Lily Safra Foundation.

## Author contributions
L.M-E., C.I.R., N.D., J.A., S.S., J.J, P.M.M. and S.J.B designed the research, L.M-E., C.I.R., N.D., J.A., N.Z., F.A.C, G.P., F.O., L.G-P, K.P., X.W., A.M. and S.J.B performed the research, L.M-E. C.I.R. and S.J.B. wrote the paper.

## Competing interests
P.M.M. is Director of the Rosalind Franklin Institute, an independent charitable company. He receives investigator-initiated research funding unrelated to this work from Biogen and is a consultant to Biogen, GSK, Nimbus Therapeutics and Sudo Biosciences. The remaining authors declare no competing interests.
