## [Transparent Peer Review file · Nature Communications]

Selective weakening of population-coupled synaptic activity *in vivo* in a mouse model of amyloid-beta pathology

Corresponding Author: Dr Samuel Barnes

Version 0:

Reviewer comments:

Reviewer #1

(Remarks to the Author)

This manuscript by Melgosa-Ecenarro, Radulescu, Doostdar et al. describes novel insights into perturbed synaptic and network function in an APP mouse model. The study has several strengths, including sophisticated multiphoton calcium recordings from excitatory and inhibitory synapses, and attempts to link these changes with transcriptomic alterations. While identifying aberrant synaptic activity in APP mice or in response to A β is not novel, the main results are compelling and informative, the imaging is beautiful and the data are presented in a visually appealing manner. However, the manuscript also has severe weaknesses that need to be addressed to strengthen the conclusions and interpretations. A mechanism how Abeta targets selective synaptic populations is not described.

One-photon calcium imaging:

- It is not clear what the precise origin of the bulk calcium signal is in their one-photon mesoscopic imaging. While calcium signals are likely related to action potential firing, it is important to determine whether these signals mainly arise from somata, axons, dendrites, or synapses (or are a mixture of these). Multiphoton calcium imaging could help to better understand the (sub)-cellular basis of their signals. Additionally, performing local field potential recordings would validate the findings of increased population activity.
- Recent work, e.g. by the Slutsky group, has suggested that neuronal hyperactivity in APP mice is related to anesthesia. Therefore, it would be important to perform the one-photon imaging during wakefulness to rule out anesthesia artifacts. This consideration is also relevant as a possible limitation of the subsequent imaging of synapses, although it might be technically too challenging.
- In Figure 1a, they show an example trace but do not provide any comparison to the NLGF mice; the same issue applies to other figures across the manuscript where original data from NLGF mice are not shown, which is a weakness as it does not allow the reader to assess the quality of the data. They focus on AUC but do not discuss the rate or frequency of the calcium signals. Furthermore, it is unclear whether the analysis in Figures 1d–k is done on a single-animal basis or aggregated from individual spines. The authors should show analyses on a per-animal basis (which is standard in the field) to strengthen conclusions and allow readers to appreciate the likely variability in the data.

Assumption of A β causality:

- In the title and throughout the manuscript, there is an inherent assumption that the observed changes are due to A β pathology. However, there is no direct evidence provided to show that this is indeed the case. No manipulations are performed to reduce A β levels to see if this alters any of the phenotypes. In fact, some of their data (e.g., Figure 1) suggest there is no clear relationship between A β plaque pathology and the functional parameters measured.
- It is interesting that they find the first calcium population changes in the visual cortex. Is there a correlation with increased plaque load in the visual cortex compared to other regions? Why is there no difference in the motor cortex? Is this area devoid of plaques? These are important questions, especially since they imply that the functional changes observed are A β -related. There is a substantial body of literature (e.g., from the Stevens lab) suggesting that in APP models, microglia approach synapses and may contribute to synaptic dysfunction. The authors report in their spatial transcriptomics results that microglia display the greatest changes, so one wonders if they play a role in the early synaptic changes reported here?
- Regarding Methoxy-X04 labeling, can the authors quantify how many of the total plaques (as labeled by anti-A β antibodies) are Methoxy-X04 positive in the NLGF mice? This is important to understand if the phenotype they report is related to a particular plaque type.

Synapse imaging:

- In Figure 1i, the authors show examples of an excitatory and an inhibitory synapse but do not discuss their correlation. Would one not expect a correlation between overall inhibitory and excitatory synaptic activity based on electrophysiological recordings to maintain synaptic excitatory/inhibitory balance?
- In Figures 1j and 1k, they assess the AUC of calcium signals but do not examine the frequency of these signals. It would be beneficial to analyze whether the frequency differs between groups, as they did e.g. in analyses presented in Figure 3.
- The authors report that they label a “heterogeneous population of inhibitory neurons”, but it would be important to understand if this involves all interneuron subtypes to a similar degree or whether there is a preference for certain subtypes in their in vivo imaging. In the subsequent spatial transcriptomics results, they make claims that PV interneurons are affected first. The authors should clarify which interneuron subtypes are labeled and affected in their study.
- Is there a time point in NLGF mice when synaptic activity is normal?
- In Figure 2, they report primarily reductions in presynaptic markers, which is interesting because previous work from the Malinow group and others has shown reductions in postsynaptic markers, such as PSD-95, in both mouse models and human AD tissue. Simply attributing this difference to variations in mouse models is not convincing and does not help move the field forward. The authors should address this discrepancy by exploring possible reasons for the different findings.
- They show that the levels of c-Fos are elevated in NLGF mice but VGAT is reduced at c-Fos-positive neurons, suggesting that loss of GABAergic presynaptic puncta may be associated with elevated neuronal activity levels in early-stage amyloidosis. However, this is not convincing, as there are many reasons for c-Fos changes in NLGF mice. More direct analyses of neuronal activity, perhaps combining functional assays with post-hoc labeling, would strengthen this claim and provide clearer evidence for the relationship between VGAT reduction and elevated neuronal activity. Indeed, previous work has shown reduced or no changes in c-FOS in several APP models vs. controls, which suggest that this technique is quite variable and also brain region-dependent.
- In Figure 3d, it would be helpful if the authors could be more explicit about which specific areas are being analyzed. Presenting analyses on a single-animal basis would also allow readers to appreciate the variability in the NLGF models. They observe a surprising “normalization” in bouton frequency from 3–4 to 6–8 months, which is interesting. Why is that? Is there a subsequent reduction at later time points?

Spatial transcriptomics:

- On the surface the spatial transcriptomics results seem to nicely support the imaging data; however, there appears to be an overextension in using this data to infer mechanistic insights into the synaptic changes. In Figure 6e, it seems that they find almost as many SST and (slightly fewer) VIP neurons as PV neurons, but in reality, PV interneurons typically constitute a larger proportion of inhibitory neurons. Why is that? What is the independent validation that they have assigned cell types/classes correctly in the UMAP plots?
- They also claim that PV interneurons show the earliest and most pronounced changes compared with other cell types, as highlighted in the abstract. However, the figure suggests there are changes across all interneuron subtypes. Using the percentage of DEGs as the main criterion to claim that one cell type is more or earlier affected than others is not convincing. The authors need to temper these claims or provide additional experimental evidence to support the preferential vulnerability of PV interneurons.
- They have 8 mice per group and can they please clarify if groups were sex-balanced in the transcriptomics analyses?

Conclusion:

While the manuscript presents interesting novel data and offers some insights into synaptic and network dysfunction in an APP mouse model, several issues need to be addressed to strengthen the validity and impact of the findings. Providing more direct evidence for the role of A β , improving data analyses and statistics, and ensuring that the main conclusions are fully supported by the data would significantly improve the manuscript.

Reviewer #2

(Remarks to the Author)

This manuscript by Melgosa-Ecenarro et al. examined cell-type-specific single-synapse pathophysiology associated with amyloidosis in an Alzheimer's disease (AD) using a second-generation knock-in AD mouse model, AppNL-G-F. In cortical regions with dysregulated meso-scale activity (elevated resting-state activity) at early stages of amyloidosis, by using in vivo two-photon calcium imaging the authors found disrupted functional associations between excitatory and inhibitory neuron ensembles and selective functional weakening of GABAergic presynaptic activity proximal to plaques and showing strong coupling with population activity. Using spatial transcriptomics, they further discovered that parvalbumin (PV) interneurons among all subclasses of GABAergic neurons have the earliest transcriptomic changes that contribute to the dysregulation of synaptic processes. This study has thus provided important insights for understanding microcircuit-level mechanisms underlying aberrant network function in AD-related pathology, highlighting an early vulnerability of PV inhibitory neurons. The study is comprehensive, and the data are in general convincing. However, there are multiple issues that need to be clarified.

Major points:

1. Fig. 2d-e, the authors separately counted VGUT/VGAT and PSD95/GEPH puncta. Are they suggesting that some presynaptic functions (or structures) are lost without much affecting the postsynaptic function/structure? Please explain here.
2. Fig. 2f, please show example zoomed-in images with both cFos-EYFP and VGAT signals in wildtype and mutant mice.
3. Fig. 2h, it is not clear how c-Fos intensity was Z-scored.
4. Line 239, “...or, alternatively, persistent synapses may exhibit compensatory changes in strength to offset local synapse loss”. Wouldn't a general increase in the density of presynaptic puncta be expected if it is a compensatory change to offset some local synapse loss?
5. Line 259, “GABAergic boutons with reduced and increased activity could be found within the same axon...”, please clarify

compared to what reduced and increased activity could be found. Does this only reflect the fact that there were natural variations among different boutons within the same axon in terms of calcium activity rather than "bouton-specific" alterations? 6. Fig. 4 and S4 shows an overall decrease in the postsynaptic activity of excitatory synapses of excitatory neurons. However, this result is inconsistent with Fig.1 which shows a global increase in the network activity at 6-8m. Please explain. 7. Imaging was performed under anesthesia, which can affect cortical activity and neuronal synchrony. Discussion on this caveat should be added.

Reviewer #3

(Remarks to the Author)

Re: Melgosa-Ecenarro et al. , early-stage amyloidosis selectively weakens population-coupled synaptic activity in vivo.

The results presented here represent a nice advancement in our understanding of global circuit changes in an amyloidosis model. Also, as stated in the manuscript, synapse loss is a hallmark of cognitive decline in Alzheimer's disease (AD), but it represents only a partial effect, with perhaps 30–40% of synapses lost during the disease's progression. This late-stage manifestation likely follows earlier synaptic dysfunction phases, which may vary between global impacts on all synapses in amyloid-affected areas or targeted effects on specific synapse subsets. The authors observe early increases in resting state activity in the visual cortex, and later increases in the retrosplenial and somatosensory cortices, with increases in both inhibitory and excitatory neuron populations and interestingly, they observed reductions in both VGLUT1 and VGAT signals. They also found that greatest number of Differentially Expressed Genes were in PV neurons. The work contains intriguing findings and is well-written, but several concerns were noted.

Main points:

-Here the authors found broad changes in cortical circuit activity using 1P mesoscale GECI imaging in APP-KI mice. Interestingly, they found early/more robust increases in circuit activity in the posterior cortical regions, including V1, but resilience in the frontal/M1 regions. The authors also observed simultaneously increased activity of both excitatory and inhibitory neurons in the V1 region using 2P imaging (except not in their axons?). There are some concerns arising here, specifically:

- 1) Prior studies found that interneuron activity either increased or decreased in an amyloidosis model depending on the interneuron subtype (Algamil et al. 2022, Communications Biology). B/c the authors are averaging across all GABAergic types here, this could be misleading as to mechanisms at the single-cell level. Recommend repeating the V1 experiment using PV and/or Sst-specific tools.
- 2) Why are GABAergic bouton activity levels decreasing while soma level activity increases? It appears the axonal data 'i.e. activity' changes are related to proximity to plaques. Is this due to action potential failures in axons close to plaques, effects on presynaptic Ca²⁺ channels near plaques, other reasons (non AP-related signaling), etc.? Just saying presynaptic 'activity' is changing is inadequate. Understanding this seems critical to the author's interpretations. Another possibility is that different inhibitory axons (i.e., PV or Sst axons) signal differently (as shown previously for their somas) in an amyloidosis situation.
- 3) I think the authors have a fantastic opportunity to also look at the single-cell excitatory and inhibitory behavior (somatically) in the 'resilient' frontal/motor region. Perhaps global changes are stable but are coming at the cost of substantial homeostatic changes (and thus stress on the system) at the single-cell level. Thus, it would certainly increase the impact to consider these experiments in that region as well.

-For the synapse quantification, these are informative experimental designs. However, it is unclear why the authors have now chosen to lump all data from 3-8 months into these datasets. Are there correlations emerging with age here- authors mention 'earlier changes' in Fig. S2a, but no data related seems to be in the figure?

-Why would overexpression of c-fos in WT and APP mice result in differential amounts- is this truly activity-dependent? Is it surprising that c-fos activity levels can be sustained in an elevated fashion for months (i.e., in APP mice)? Wouldn't the system downregulate that signaling at some point? What is the threshold for a c-Fos positive neuron? Is c-Fos distributed uniformly in the neurites?

-The simultaneous imaging of excitatory neurons and presynaptic putative basket contacts is a very impressive experiment. However, it is unclear what functional property is being tested (relatedly, the authors rely on 'see Methods' far too much throughout the manuscript... tell us what is being tested in the results) and what is the relevance? One cannot infer changes in inhibitory synaptic strength from a change in Ca²⁺ amplitude alone. Also jitter of PV synapses with Ca²⁺ requires some ground truthing to see if temporal resolution is there (look at previous patch paired recordings perhaps)?

In the results and/or discussion, the authors should address the fact that GABAergic circuit dysfunction and, thus, hyperexcitability is likely a feature of APP-NL-G-F at very early time points (i.e., 1-2 months in Petrache et al. 2019 already cited) and has also been noted in very early APP/Abeta models (i.e., 3-4 weeks post expression; Goettemoeller et al. 2024; also Bushe & Kole 2012, showing hyperexcitability before extensive plaques). This is (importantly) complemented here by the authors' finding of little correlation between resting state activity and proximity to X04 staining (although perhaps not for the axonal regions).

Minor points:

-Line 79-80: there are many papers, including the 1.0 and 2.0 (KI) amyloid pathology mice, which show preferential pathophysiology (mostly presynaptic) in specific synapse types-notably PV interneurons...would recommend adding these additional citations in context(results and/or discussion)- Chen, Mody 2018 eNeuro; Sos, Mody 2020 PLOS ONE; Kumar,

Rangaraju 2024 Nature Comm.) which likely help with interpretations of the data here.

-Line 89: "which occurs with a functional uncoupling of E-I assemblies." Please add a reference here; it is an important point.

-EF1a is not known to be a promoter specific to excitatory neurons, or neurons alone. It is remarkably surprising that the authors did not find overlap between cells expressing their mDlx driven and EF1a driven constructs. Fortunately, this does not appear to affect most of the claims in the paper, thus making this a minor point.

Version 1:

Reviewer comments:

Reviewer #1

(Remarks to the Author)

I thank the authors for addressing my comments and questions with additional experiments, analysis and clearer language. Still, several central claims need tempering to avoid overstating causality beyond what the data support. Throughout the Abstract, Results, Discussion and Conclusions, the text implies that early synaptic pathophysiology is driven by or results from selective vulnerability of "strongly population-coupled inhibitory presynaptic inputs" and that PV interneurons are the direct locus of this effect. Because all bouton imaging comes from mDlx-labelled, SST-biased inhibitory axons (not PV-specific axons) this wording goes beyond what the data show. In addition, the manuscript currently implies temporal sequences in the transcriptomics (PV changes "followed by" broader GABAergic changes) that cannot be inferred from cross-sectional data. There also remain issues with regard to analysis definitions and statistical correctness. The population-coupling metric needs clearer specification because the current definition could introduce autocorrelation. Likewise, the manuscript does not explicitly state whether synaptic statistics treat boutons/spines as independent samples or whether per-animal summaries or mixed-effects models were used. If synapses are treated as independent n's, this constitutes pseudo-replication.

Minor points:

"However, more recent investigations have reported compelling evidence for GABAergic changes associated with neuronal hyperactivity in early stages of AD" - here the authors only cite themselves and a transcriptomics study but not the original studies that have shown this association more directly

"Several studies have implicated PV- and SST-positive interneurons in early-stage amyloidosis" - the authors are encouraged to cite the primary papers and not only their own review paper

Reviewer #2

(Remarks to the Author)

This manuscript by Melgosa-Ecenarro et al., has improved a lot through the revision, by adding multiple sets of new experiments. The reviewer only identified some small issues before publication.

1. In Introduction, line 71, "while early-stage amyloid-related changes have been associated with neuronal hyperactivity...", a highly relevant paper should be cited here (Zhang et al., Front Aging Neurosci 2023, doi: 10.3389/fnagi.2023.1213379), which reviews amyloid-associated hyperactivity, synaptic loss and excitation-inhibition imbalance, etc.

2. Figure 1m,n,p,q, how was the fraction of associated neurons quantified? What was the criterion for determining whether two neurons are associated? I could not find this information in the manuscript.

Reviewer #3

(Remarks to the Author)

This manuscript provides a compelling, technically sophisticated and comprehensive investigation into early synaptic and circuit-level dysfunction using App-KI model of amyloid pathology. Together, the data converge on a central conclusion that synaptic pathophysiology is not global, but rather selectively targets population-coupled inhibitory presynaptic boutons and excitatory dendritic spines, with PV interneurons disproportionately affected. The population assembly data is a highly novel for the field.

The manuscript was already visually beautiful and analytically rigorous. It is now strengthened by extensive revisions and new experiments detailed in the rebuttal. The authors responded thoroughly to reviewer concerns and provided added analyses, controls, and clarifications that greatly enhance the clarity and impact of the work. This includes interpretation of mesoscopic signals, new analyses, anesthesia-related concerns were addressed with new experiments- providing increased confidence results were not artifact related to anesthesia. The rebuttal is detailed and supported by substantial new experimental additions in general, resulting in a markedly strengthened manuscript.

REVIEWER COMMENTS

Reviewer #1 (Remarks to the Author):

This manuscript by Melgosa-Ecenarro, Radulescu, Doostdar et al. describes novel insights into perturbed synaptic and network function in an APP mouse model. The study has several strengths, including sophisticated multiphoton calcium recordings from excitatory and inhibitory synapses, and attempts to link these changes with transcriptomic alterations. While identifying aberrant synaptic activity in APP mice or in response to A β is not novel, the main results are compelling and informative, the imaging is beautiful and the data are presented in a visually appealing manner. However, the manuscript also has severe weaknesses that need to be addressed to strengthen the conclusions and interpretations. A mechanism how Abeta targets selective synaptic populations is not described.

One-photon calcium imaging:

It is not clear what the precise origin of the bulk calcium signal is in their one-photon mesoscopic imaging. While calcium signals are likely related to action potential firing, it is important to determine whether these signals mainly arise from somata, axons, dendrites, or synapses (or are a mixture of these).

We thank the reviewer for their comment. Our 1-photon widefield mesoscopic calcium imaging is conducted using the Thy-1-GCaMP6s line (Feng et al., 2000). In this genetic line, deep layer 5 excitatory neurons and superficial layer 2/3 excitatory neurons are labelled. In our preparation, 1-photon mesoscopic imaging is restricted to the most superficial areas of the cortex. As such, our 1-photon mesoscopic imaging experiments are likely sampling population level signals from L2/3 excitatory neurons and the dendrites of L5 neurons that project into superficial layers of the cortex. We have added text to the methods to clarify this point; the text now runs as follows:

Methods

Imaging was conducted using the Thy-1-GCaMP6s line¹¹⁰ either alone or crossed with *App*^{NL-G-F} mice⁴⁰. In the Thy-1-GCaMP6s line, deep layer 5 excitatory neurons and superficial layer 2/3 excitatory neurons express GCaMP6s¹¹⁰. 1-photon widefield mesoscopic imaging is restricted to the most superficial areas of the cortex. As such, our 1-photon mesoscopic imaging experiments are likely sampling population level signals from L2/3 excitatory neurons and the dendrites of L5 neurons that project into superficial layers of the cortex.

Multiphoton calcium imaging could help to better understand the (sub)-cellular basis of their signals.

As the reviewer quite rightly states, our population level mesoscopic imaging experiments sampled a mixture of cellular signals (**Fig 1a-g**). Next, as suggested by the reviewer, we used multiphoton imaging to probe sub-cellular signals at the soma and synapse (**see figures 1-5**, to better understand the origins of the changes in the mesoscopic signal. We have added text to the results to clarify this point.

Results

Functional uncoupling of excitatory and inhibitory assemblies in *App*^{NL-G-F} mice

Elevated resting-state activity in excitatory neurons (**Fig.1d-g**) may be driven by reductions in inhibitory neuronal activity⁹. To better understand the cellular origins of the activity changes we observed with mesoscopic calcium imaging, we made *in vivo* 2-photon calcium imaging measurements from functional assemblies comprising excitatory and inhibitory neurons in anaesthetised *App*^{NL-G-F} mice (**see Methods, Fig.1h-i**).

Additionally, performing local field potential recordings would validate the findings of increased population activity.

As suggested by the reviewer, we have now added electrophysiological recordings to better investigate the population-level response. Specifically, we developed an imaging and electrophysiology approach which combines cortical EEG measures with mesoscopic calcium imaging in awake animals (**Fig.5i**).

With this approach, we first confirmed that App^{NL-G-F} animals have greater levels of calcium activity in awake conditions (**Fig.5j, S5d**). We next used the event-triggered averaging epochs to test for differences in the EEG recordings during the pre-transient baseline period and rising phases (**Fig.5k-l**). We found a broad increase in EEG power across multiple frequency bands (**Fig.5m-n**). The onset of a calcium event correlated with EEG features that differed between wild-type and App^{NL-G-F} animals (**Fig.5o-p**). Specifically, we found a reduction in the power of low (1-30 Hz) and high (60-80 Hz) frequency bands in App^{NL-G-F} mice during the change from baseline to the rising phase (**Fig.5o-p**). We have now added additional text and figures relating to this work throughout the manuscript, which runs as follows – with figure 5 included after the next reviewer response:

Results

Amyloid-associated hyperactivity is accompanied by deficits in specific low and high frequency neuronal oscillations

Our data suggest that synaptic dysfunction is evident during amyloid-associated hyperactivity (**Fig.3-4**). We wondered whether other biomarkers of pathophysiology, which may have greater translational relevance, are also evident during this time. To address this issue, we combined widefield 1-photon mesoscopic calcium imaging with simultaneous EEG recordings in awake $Thy1-GCaMP6s$ mice and $App^{NL-G-F} \times Thy1-GCaMP6s$ crosses (see **Methods, Fig.5i**). In agreement with our earlier mesoscopic calcium imaging work (**Fig.1d-e**), we found increased resting-state calcium-mediated neuronal activity only in the posterior cortical regions (VIS and RSC) of awake App^{NL-G-F} mice when compared to controls (**Fig.5j, S5d**). We then used event-triggered averaging of mesoscopic calcium events from visual cortex and measured temporally associated local field potential (LFP) recordings from a superficial electrode (**Fig.5k**). We investigated the power of EEG frequency bands during the pre-event baseline and the rising phase of the calcium event (**Fig.5k-l**). Following our imaging results, we observed increased power in App^{NL-G-F} mice, spanning all measured frequency bands in the baseline phase immediately before the onset of a calcium event (**Fig.5m**). During the rising phase, we found evidence for reduced power in the 1-10 Hz range (**Fig. 5n**). To better understand the oscillatory dynamics associated with calcium events, we then measured the difference in the power of the frequency bands from baseline to the rising phase. Here, we found a reduced response in App^{NL-G-F} mice, specifically in lower (1-30 Hz) and higher (60-80 Hz) frequency bands (**Fig.5o-p**). Our data suggest that despite the global increase in power across all bands, the onset of amyloid-associated hyperactivity in App^{NL-G-F} mice coincides with deficits in more specific low and high frequency neuronal oscillations *in vivo*.

Recent work, e.g. by the Slutsky group, has suggested that neuronal hyperactivity in APP mice is related to anesthesia. Therefore, it would be important to perform the one-photon imaging during wakefulness to rule out anesthesia artifacts. This consideration is also relevant as a possible limitation of the subsequent imaging of synapses, although it might be technically too challenging.

We thank the reviewer for their useful suggestion and have now added widefield 1-photon mesoscopic calcium imaging with simultaneous EEG recording during wakefulness to test for anaesthesia effects (**Fig.5i-p**). As raised by the reviewer, previous work from the Slutsky group has shown that anaesthesia is associated with hyperactivity in the CA1 region of the hippocampus in AD-mouse models (Zarhin et al., 2022). We find increased calcium-mediated neuronal activity levels in the posterior cortical regions of both anaesthetised and awake

App^{NL-G-F} mice when compared to wild-type animals (**Fig.5j, S5d**). This suggests that the impact of anaesthesia on AD-related hyperactivity may exhibit regional variability. Our EEG electrode was implanted in superficial areas of visual cortex. Following our imaging results, we observed increased power in *App^{NL-G-F}* mice, spanning all measured EEG frequency bands in the baseline phase immediately before the onset of a calcium event (**Fig.5m**). The relevant text and figures are integrated into the manuscript and included below.

Amyloid-associated hyperactivity is accompanied by deficits in specific low and high frequency neuronal oscillations

Our data suggest that synaptic dysfunction is evident during amyloid-associated hyperactivity (**Fig.3-4**). We wondered whether other biomarkers of pathophysiology, which may have greater translational relevance, are also evident during this time. To address this issue, we combined widefield 1-photon mesoscopic calcium imaging with simultaneous EEG recordings in awake *Thy1-GCaMP6s* mice and *App^{NL-G-F}xThy1-GCaMP6s* crosses (see **Methods, Fig.5i**). In agreement with our earlier mesoscopic calcium imaging work (**Fig.1d-e**), we found increased resting-state calcium-mediated neuronal activity only in the posterior cortical regions (VIS and RSC) of awake *App^{NL-G-F}* mice when compared to controls (**Fig.5j, S5d**). We then used event-triggered averaging of mesoscopic calcium events from visual cortex and measured temporally associated local field potential (LFP) recordings from a superficial electrode (**Fig.5k**). We investigated the power of EEG frequency bands during the pre-event baseline and the rising phase of the calcium event (**Fig.5k-l**). Following our imaging results, we observed increased power in *App^{NL-G-F}* mice, spanning all measured frequency bands in the baseline phase immediately before the onset of a calcium event (**Fig.5m**).

Fig. 5 | Hyperactivity is associated with GABAergic dysfunction and abnormal oscillatory activity in a mouse model of amyloidosis.

[...] i, Schematic (top) and example traces (bottom) of simultaneous superficial local field potential recordings (LFP, light grey) and 1-photon mesoscopic imaging of calcium-mediated activity (black) in awake mice. Scale bars: 2.0 $\Delta F/F_0$, 400 nV, and 5 s. j, Normalised frequency, amplitude, and total activity (freq x amp) averaged per mouse in awake *Thy1-GCaMP6s* (black) and *App^{NL-G-F} x Thy1-GCaMP6s* (blue) visual cortex. k, Example LFP trace (top) associated with calcium transient (bottom). Red dashed lines denote time periods used for

baseline and rising phase. **l**, Time-frequency power plot using wavelet convolution and log values converted in WT (top) and *App^{NL-G-F}* (bottom) mice. Dashed lines denote baseline and rising phase described in **k**. **m-n**, Mean Welch's power spectral density (PSD) estimates in 10 Hz bands during the baseline (**m**) and rising phase (**n**) shown in **k-l**. **o**, Change in Welch's PSD estimate between baseline and rising phase. **p**, Mean PSD difference (shown in **o**) in 10 Hz bands.

Supplementary Fig.5 | *App^{NL-G-F}* mice show increased activity in posterior cortical regions.

[...] **d**, Normalised calcium-mediated activity (freq x amp) per mouse in visual (VIS), retrosplenial (RSC), somatosensory (SS) and motor (MO) areas at 6-8m in *Thy-1-GCaMP6s* (black) and *App^{NL-G-F} x Thy-1-GCaMP6s* (blue) awake mice.

In Figure 1a, they show an example trace but do not provide any comparison to the NLGF mice; the same issue applies to other figures across the manuscript where original data from NLGF mice are not shown, which is a weakness as it does not allow the reader to assess the quality of the data.

We thank the reviewer for their comment. We have now added comparative traces throughout the manuscript. Some examples are included below:

Fig. 1 | Dysregulated resting-state activity at excitatory and inhibitory neurons during amyloidosis.

a, Schematic depicting 1-photon mesoscopic imaging. Trace: resting-state calcium-mediated neuronal activity from visual/parietal cortex in *App^{NL-G-F} x Thy-1-GCaMP6s* (blue) and *Thy-1-*

GCaMP6s (black) mice. Scale bars: 0.2 $\Delta F/F_0$ and 10 s. [...] **h**, Example region (V1) from *in vivo* 2-photon imaging showing expression of co-injected viral constructs to label excitatory (green only) and inhibitory (red and green) neurons as well as amyloid plaques via methoxy-X04 (magenta). Scale bar: 10 μm . **i**, Example of $\Delta F/F_0$ calcium traces from excitatory (top) and inhibitory (bottom) neurons during resting-state activity in *App^{NL-G-F}* (blue, red) and WT (black) mice. Scale bars: 1.0 $\Delta F/F_0$, 20 s. [...] **r**, Example region (V1) from *in vivo* 2-photon imaging showing expression of structural (cyRFP, red) and functional (GCaMP6s, green) markers in PV neurons as well as amyloid plaques via methoxy-X04 (magenta). Scale bar: 10 μm . Traces: Example of $\Delta F/F_0$ calcium traces from PV neurons in 4-6m *App^{NL-G-F}* (red) and WT (black) mice during resting-state activity. Scale bars: 1.0 $\Delta F/F_0$, 10 s.

They focus on AUC but do not discuss the rate or frequency of the calcium signals.

We thank the reviewer for raising this issue. We have now added animal-level analysis of amplitude and frequency from our mesoscopic calcium imaging experiments (**Fig.1a-g**, **Fig.S1a-h**). We find an increase in total activity (amplitude x frequency) at the animal level in visual and retrosplenial cortices at 3-4 and 6-8 months (**Fig.1d-e**), which is consistent with our previous analysis. The increased activity is driven by an increased frequency of events (**Fig.S1a-b**), with no statistically significant changes in mesoscopic event amplitude at the animal level (**Fig.S1e-f**). We find increases in frequency in SS, and in MO at later stages (**Fig.Sc-d**, **g-h**), but these are not sufficient to significantly increase overall activity levels (**Fig.1f-g**). Therefore, our data suggest that the strongest changes in activity in *App^{NL-G-F}* mice are centred in posterior cortical regions. The relevant text and figures are shown below.

Results

We crossed *App^{NL-G-F}* mice with mice expressing GCaMP6s under the *Thy1* promoter and used longitudinal 1-photon mesoscopic imaging to measure calcium-mediated neuronal activity across the dorsal surface of the cortex in anaesthetised animals⁴² (**see Methods**, **Fig.1a-b**). We then performed a coarse parcellation to group functionally related regions with high interconnectivity^{42,43}: visual/parietal (VIS), retrosplenial (RSC), somatosensory (SS), and motor (MO) cortices (**Fig.1c**). Within those regions, we measured the activity (transient amplitude x transient frequency of the calcium signal) at the animal level during resting state (collected when mice were at rest in the dark⁴⁴) (**see Methods**, **Fig.1d-g**, **S1a-h**). Using this approach, we found resting-state activity to be elevated in a broad mesoscopic circuit centred on posterior cortices in *App^{NL-G-F} x Thy-1-GCaMP6s* mice (**Fig.1d-g**). Early changes in activity at 3-4m were observed in VIS (**Fig.1d**) and RSC (**Fig.1e**) and were driven by changes in the frequency (**Fig. S1a-b**) rather than amplitude (**Fig.S1e-f**) of mesoscopic calcium events. Increases in frequency were also observed in SS, as well as in MO at later stages (**Fig.S1c-d**, **g-h**). However, these changes were not sufficient to produce a significant rise in overall activity levels (**Fig.1f-g**).

Fig. 1 | Dysregulated resting-state activity at excitatory and inhibitory neurons during amyloidosis.

a, Schematic depicting 1-photon mesoscopic imaging. Trace: resting-state calcium-mediated neuronal activity from visual/parietal cortex in *App^{NL-G-F} x Thy-1-GCaMP6s* (blue) and *Thy-1-GCaMP6s* (black) mice. Scale bars: 0.2 $\Delta F/F_0$ and 10 s. **b**, Standard deviation (STD) of calcium-mediated activity ($\Delta F/F_0$) in example *Thy-1-GCaMP6s* (left) and *App^{NL-G-F} x Thy-1-GCaMP6s* (right) mice. **c**, Broad functional parcellation of the dorsal cortex into visual/parietal (VIS), retrosplenial (RSC), somatosensory (SS) and motor (MO) regions. **d-g**, Average activity (transient amplitude x frequency) per mouse measured at 3-4m (left) and 6-8m (right) in WT (black) and *App^{NL-G-F}* (blue) *Thy-1-GCaMP6s* mice from broad parcellated regions in panel c. [...]

Supplementary Fig. 1 | Dysregulated resting-state activity at excitatory and inhibitory neurons in a mouse model of amyloidosis.

a-h, Average frequency (**a-d**) and amplitude (**e-h**) per mouse measured at 3-4m (left) and 6-8m (right) in WT (black) and *App^{NL-G-F}* (blue) *Thy-1-GCaMP6s* mice from broad parcellated regions in Figure 1c. [...]

Furthermore, it is unclear whether the analysis in Figures 1d–k is done on a single-animal basis or aggregated from individual spines. The authors should show analyses on a per-animal basis (which is standard in the field) to strengthen conclusions and allow readers to appreciate the likely variability in the data.

We thank the reviewer for their comment. The new animal-level analysis of mesoscopic data is now included in the manuscript (see previous comment).

The data in figures 1j-k in the original manuscript are cellular measures and not dendritic spines; we have added text to better communicate this.

Results

We then estimated somatic resting-state activity levels using the AUC of the $\Delta F/F_0$ calcium signal in excitatory and inhibitory cells (see Methods).

The previous version of the manuscript did include animal-level analysis of the cellular data (AUC), which we now better signpost.

Results

We found increased levels of excitatory resting-state activity in App^{NL-G-F} mice compared to age-matched WT mice at the cellular and animal levels (Fig.1j, S1n). Changes in resting-state activity at excitatory neurons were mirrored by similar increases in the resting-state activity of inhibitory neurons (Fig.1k, S1n).

We also include the animal-level analysis of the amplitude and frequency of calcium events from the cellular data as suggested by the reviewer. The relevant figures have been added to the supplementary material and are shown below:

Supplementary Fig. 1 | Dysregulated resting-state activity at excitatory and inhibitory neurons in a mouse model of amyloidosis.

[...] n, Area under the curve (AUC) of the $\Delta F/F_0$ trace of excitatory (left) and inhibitory (right) cells per mouse in 3-8m in WT (black) and App^{NL-G-F} (blue, red) mice. [...] q, Amplitude of calcium events at excitatory (left) and inhibitory (right) cells per mouse in 3-8m in WT (black) and App^{NL-G-F} (blue, red) mice. [...] t, Frequency of calcium events at excitatory (left) and inhibitory (right) cells per mouse in 3-8m in WT (black) and App^{NL-G-F} (blue, red) mice. [...] t, Frequency of calcium events at excitatory (left) and inhibitory (right) cells per mouse in 3-8m in WT (black) and App^{NL-G-F} (blue, red) mice.

Assumption of A β causality:

In the title and throughout the manuscript, there is an inherent assumption that the observed changes are due to A β pathology. However, there is no direct evidence provided to show that this is indeed the case. No manipulations are performed to reduce A β levels to see if this alters any of the phenotypes. In fact, some of their data (e.g., Figure 1) suggest there is no clear relationship between A β plaque pathology and the functional parameters measured.

We thank the reviewer for their comment and have revised the title and the text, as well as adding a section to the discussion to address this issue. The new title now runs as follows: ‘**Selective weakening of population-coupled synaptic activity *in vivo* in a mouse model of amyloid-beta pathology.**’ In addition to this, we have also run a series of experiments and analysis to test for associations between amyloid levels and pathophysiology across the circuit measures we have made. The relevant sections in the results and discussion are covered below.

First, we ran new analysis and found those mice with the greatest plaque load to have increased cellular GCaMP activity in both excitatory and inhibitory neurons (**Fig.S1w**). We also found excitatory neurons to show increased activity at all measured distances (**Fig.S1x**), but only those inhibitory neurons within 20 μ m showed increased activity (**Fig.S1x**). The relevant text and figure now run as follows:

Results

Given that the increased activity of excitatory and inhibitory neurons was driven by increases in calcium event amplitude (**Fig.S1o-q**), we next tested the extent to which increases in amplitude were associated with plaque load at the animal level (**Fig.S1w**). We found that higher plaque load was associated with increased amplitude in both excitatory and inhibitory neurons, while *App*^{NL-G-F} mice with low plaque densities were comparable to WT mice (**Fig.S1w**). Excitatory neurons showed increased amplitude at all measured distances (**Fig.S1x**). In contrast, inhibitory cells only showed increased activity within 20 μ m of a plaque (**Fig.S1x**).

Supplementary Fig. 1 | Dysregulated resting-state activity at excitatory and inhibitory neurons in a mouse model of amyloidosis.

[...] **w**, Heatmap showing normalised (to WT) excitatory (top) and inhibitory (bottom) neuron amplitude in 3-8m *App*^{NL-G-F} mice with increasing plaque load. **x**, Heatmaps showing difference (*App*^{NL-G-F}-WT) in the z-scored (to WT) amplitude values of excitatory (top) and inhibitory (bottom) neurons located at increasing distances from methoxy-X04-labelled plaques in *App*^{NL-G-F} mice. [...]

Second, we investigated whether regional differences in amyloid pathology corresponded to patterns of neuronal hyperactivity in mesoscopic calcium data, with posterior regions (visual cortex) being more hyperactive than anterior regions (motor cortex). We found that the visual cortex showed elevated markers of amyloid (MOAB-2, methoxy-X04) and increased markers of neuronal activity (c-Fos) compared to the motor cortex (**Fig. S1j-m**). *In vivo* two-photon

imaging further confirmed higher plaque density in the visual cortex (**Fig. S1z**). The relevant text and figure now run as follows:

Results

Using immunofluorescence measures, we also observed that posterior regions (VIS), which exhibit earlier signs of hyperactivity in *App^{NL-G-F}* mice, showed increased markers of both putative neuronal activity (c-Fos) and amyloidosis (methoxy-X04 and MOAB-2) in comparison to frontal regions (MO) (**Fig.S1j-m**).

Supplementary Fig. 1 | Dysregulated resting-state activity at excitatory and inhibitory neurons in a mouse model of amyloidosis.

[...] **j**, Cortical region showing immunofluorescence labelling of c-Fos (green), methoxy-X04 (magenta), and MOAB-2 (cyan) in *App^{NL-G-F}* mice. Scale bar: 20 μm . **k-m**, Normalised (to V1) fluorescence intensity of c-Fos (**k**), methoxy-X04 (**l**), and MOAB-2 (**m**) in the visual (black) and motor (orange) cortices of 6-8m *App^{NL-G-F}* mice. [...]

Results:

Our mesoscopic imaging data suggested that frontal regions, such as the motor cortex, are more resilient to hyperactivity than posterior regions (**Fig.1a-g**, **Fig.S1a-h**). To test whether activity changes are detectable at the cellular level in these frontal regions, we repeated our cellular imaging experiments by measuring the calcium activity of excitatory and inhibitory neurons, as well as methoxy-X04 plaque density, in motor cortex at 4-6 months (**Fig.S1y**). Consistent with our mesoscopic imaging data, activity levels in both excitatory and inhibitory cells were comparable between *App^{NL-G-F}* and WT mice (**Fig.S1y**). We also found *in vivo* plaque load to be lower in motor cortex than age-matched timepoints in visual cortex (**Fig.S1z**).

Supplementary Fig. 1 | Dysregulated resting-state activity at excitatory and inhibitory neurons in a mouse model of amyloidosis.

[...] **y**, Normalised (to WT) area under the curve (AUC), mean frequency (Freq), and amplitude (Amp) of calcium events at excitatory (left) and inhibitory (right) cells per mouse in 4-6m WT (black) and *App^{NL-G-F}* (blue, red) mice. **z**, Density of methoxy-X04-positive plaques in the motor (left) and visual (right) cortex of 4-6m *App^{NL-G-F}* mice. [...]

Third, our structural synapse imaging data found the greatest reduction in GABAergic bouton density to be in regions with high plaque load (**Fig.3d**), while our functional synaptic imaging data found pathophysiology to vary with distance to plaque, with hyperactivity in the periphery and hypoactivity in more proximal regions (**Fig.3j-k**). In addition, our new analysis found

GABAergic bouton activity to correlate with amyloid load at the animal level (**Fig.S4e-f**). The relevant text and figures now run as follows:

Results

We first measured the density of GABAergic boutons *in vivo* (see **Methods**) and found that bouton density was stable at 3-4m (**Fig.3b**) but reduced by 6-8m (**Fig.3c**) in *App^{NL-G-F}* mice when compared to WT. GABAergic bouton density showed the greatest reduction in areas with high plaque load (**Fig.3d**).

Fig. 3 | Functional signatures of dysregulated bouton activity in *App^{NL-G-F}* mice.

[...] **d**, Bouton density in areas with increasing methoxy-X04-labelled plaque density. [...]

Results

Our data suggest a complex activity profile for GABAergic boutons that varies with both age and distance to plaques. More distal GABAergic boutons initially map to elevated GABAergic cellular activity at early time points but then show a decrease in activity with both time and proximity to pathology (**Fig.3i-k**).

Fig. 3 | Functional signatures of dysregulated bouton activity in *App^{NL-G-F}* mice.

[...] **i**, Example of superficial GABAergic axons expressing cyRFP (red) and a methoxy-X04-labelled plaque (magenta). Dashed circles represent plaque edge and 20 μm increments. Scale bar: 10 μm . **j-k**, Heatmaps showing difference (*App^{NL-G-F}*-WT) in the z-scored (to WT) average area under curve (AUC), frequency (Freq), and amplitude (Amp) values of GABAergic boutons located at increasing distances from methoxy-X04-labelled plaques in 3-4m (**j**) and 6-8m (**k**) *App^{NL-G-F}* mice. Values on the x-axis represent minimum distance for each bin. [...]

Results

In addition, changes in GABAergic bouton, but not spine, activity correlated with plaque load at the animal level (**Fig.S4e-f**). Together, this highlights the early-stage vulnerability of cortical GABAergic circuitry to amyloid-related pathology in this model *in vivo*.

Supplementary Fig. 4 | Functional signatures of dysregulated synaptic activity in *App^{NL-G-F}* mice.

[...] **e-f**, Correlation between GABAergic bouton (**e**) and excitatory dendritic spine (**f**) activity (norm to WT) and median plaque density per mouse in 3-8m *App^{NL-G-F}* mice. [...]

Fourth, our spatial transcriptomics analysis specifically identified molecular processes in parvalbumin and somatostatin-positive cells that covaried with local plaque load, whereas other inhibitory cell-types such as VIP and LAMP5 were more resilient. The relevant text and figure run as follows:

Results

We found that all GABAergic subclasses showed increasing levels of DEGs with high plaque load (**Fig.6k**). However, PV and SST interneurons were associated with more than double the percentage of DEGs that covaried with regional plaque load compared to VIP and LAMP5 subclasses (**Fig.6l, S6k-m, Tables S6, S12**).

Fig. 6 | Modified synaptic transmission at GABAergic neurons in the *App^{NL-G-F}* mouse.

[...] **k**, Percentage of DEGs by plaque load (based on areas with low, mid, and high plaque number) for inhibitory subclasses in *App^{NL-G-F}* mice. Comparisons against low plaque areas shown for each subclass. **l**, DEGs that covary with plaque load for PV (top) and SST (bottom) in *App^{NL-G-F}* mice. All panels DEGs: MAST, adj. p < 0.05 and abs(log2FC) > 0.25. [...] Dot plots: the size of dots is proportional to [...] the abs(Log2FC) (**l**); the colour bars show Log2FC (g, **i**, **l**). [...]

Supplementary Fig. 6 | Transcriptomic changes during amyloidosis in the *App^{NL-G-F}* mouse.

[...] **k**, Percentage of DEGs that covary with plaque load when comparing *App^{NL-G-F}* and WT across GABAergic subclasses. Comparisons against SST shown for each subclass. **l-m**, DEGs that covary with plaque load for VIP (**l**) and LAMP5 (**m**) neurons. [...]

Fifth, we added PV specific in vivo 2-photon calcium imaging data and found evidence of hyperactivity in this GABAergic subtype (**Fig.1t**), which was also associated with increased plaque load (**Fig.1u**). The relevant text and figure run as follows:

Results

The mDlx-based labelling approach targets a heterogeneous population of inhibitory neurons⁴⁵. However, our previous work³⁷ and that of others⁴⁶ found a bias toward SST neurons using this enhancer system (~ 22% of GCaMP6s-expressing neurons were PV positive and ~60 % were SST positive³⁷). Therefore, to test for the involvement of other subclasses, we next conducted 2-photon imaging experiments using a PV-specific viral targeting strategy (AAV1-S5E2p-cyRFP-GSG-P2A-HIS-GCaMP6s-WPRE)^{51,52} to label PV neurons with GCaMP6s and cyRFP in V1 (**Fig.1r**). Immunofluorescence measures validated that 99.4% (505/508 cells) of cyRFP positive neurons also expressed PV (**Fig.1s**). Using 2-photon imaging, we found evidence for an increased frequency and amplitude of calcium events in PV neurons in 4-6m *App^{NL-G-F}* mice relative to age-matched controls (**Fig.1t**). In addition, elevated PV neuron activity was associated with greater local amyloid plaque load (**Fig.1u**). We next conducted immunofluorescence experiments labelling PV, SST, and c-Fos in the visual cortex of WT and *App^{NL-G-F}* mice (**Fig.1v**). Using this approach, we found increased c-Fos expression at both PV and SST neurons in *App^{NL-G-F}* mice by 4-6m, with PV neurons showing a relatively greater increase than SST cells (**Fig.1w-x**). Together, our data suggest that both PV and SST neurons show hyperactivity, based on increased c-Fos in both GABAergic subclasses, and elevated calcium activity measured with the mDlx construct (biased toward SST) and the S5E2 construct (PV-specific).

Fig. 1 | Dysregulated resting-state activity at excitatory and inhibitory neurons during amyloidosis.

[...] **r**, Example region (V1) from *in vivo* 2-photon imaging showing expression of structural (cyRFP, red) and functional (GCaMP6s, green) markers in PV neurons as well as amyloid plaques via methoxy-X04 (magenta). Scale bar: 10 μm. Traces: Example of ΔF/F₀ calcium traces from PV neurons in 4-6m *App^{NL-G-F}* (red) and WT (black) mice during resting-state activity. Scale bars: 1.0 ΔF/F₀, 10 s. **s**, Example cortical region showing immunofluorescence

labelling of PV (blue) in cyRFP-expressing neurons (red) and a quantification of double-positive cells in V1 (bottom right). Scale bar: 20 μm . **t**, Normalised (to WT) frequency (left) and amplitude (right) of calcium events in PV neurons in 4-6m *App*^{NL-G-F} (red) and WT (black) mice. **u**, Heatmap of normalised (to WT) frequency (top) and amplitude (bottom) values of PV neurons in regions with increasing methoxy-X04 plaque density in 4-6m *App*^{NL-G-F} mice. **v**, Cortical region showing immunofluorescence labelling of c-Fos (green), PV (red), and SST (cyan). Scale bar: 10 μm . **w**, Normalised (to WT) c-Fos intensity in PV (left, red) and SST (right, cyan) neurons in 4-6m *App*^{NL-G-F} (red, cyan) and age-matched WT (black) mice. **x**, Normalised (to WT) c-Fos intensity in PV (red) and SST (cyan) neurons in 4-6m *App*^{NL-G-F} mice.

Finally, we added a section to the discussion to more carefully state that our findings are correlative, and not causative, so other A β -independent mechanisms could be involved.

Discussion

Multiple cellular and micro-circuit changes in our data correlated with amyloid plaque load. For example, our spatial transcriptomics analysis identified molecular processes in PV and SST cells that covaried with regional plaque load. In contrast, other inhibitory subclasses such as VIP and LAMP5 had far fewer DEG signatures that covaried with amyloid pathology. In addition, increased activity of both excitatory and inhibitory neurons, as well as reductions in GABAergic bouton density and calcium activity, were more evident in regions with higher plaque levels. Recent work has shown no changes in activity in superficial layers (L1-4) in *App*^{NL-G-F} mice at 1.5m, when plaque levels are very low in this mouse model⁹⁷. Consistent with this, we found that mice with low plaque load were similar to WT, suggesting an early-stage resilience which may involve homeostatic processes^{38,72}. However, it is important to note that we imaged at later timepoints and exclusively in regions where plaques were evident, to specifically study the impact of the plaque microenvironment on micro-circuit activity. This is not the case in recent work which focused on pre-plaque periods⁹⁷. Others have reported synaptic deficits in similar knock-in models in the entorhinal cortex by 2m⁷⁶, suggesting that dysfunctional synaptic activity may be detectable during very early stages of amyloid-beta pathology. Taken together, our distance and plaque density measurements suggest an association between micro-circuit dysregulation and amyloid pathology. However, this does not imply causation, and further work is needed to determine whether the effects we observe are directly driven by soluble or aggregated forms of A β , or whether other mechanisms, potentially initiated by A β , or independent of it, are involved⁹⁸.

It is interesting that they find the first calcium population changes in the visual cortex.

We thank the reviewer for their comment and agree that this is an interesting result. Visuospatial symptoms are increasingly recognized among patients with dementia (Mukherjee et al., 2020), and visual paradigms are emerging as promising functional biomarkers of amnesic dysfunction (Stothart et al., 2025). Recent positron emission tomography (PET) studies in humans have also identified the occipital lobe among the regions with the earliest A β accumulation (Lecy et al., 2024). In line with this, the increased levels of neuronal activity in our dataset appear to originate in a broad mesoscopic circuit centred around the retrosplenial and visual cortices.

Is there a correlation with increased plaque load in the visual cortex compared to other regions? Why is there no difference in the motor cortex? Is this area devoid of plaques? These are important questions, especially since they imply that the functional changes observed are A β -related.

The reviewer raises an interesting possibility that we have tested by conducting a series of experiments and analyses. First, we ran immunolabelling experiments for methoxy-X04,

MOAB-2 and c-Fos in both the visual cortex and motor cortex of App^{NL-G-F} mice (**Fig.S1j-m**). We found that the visual cortex shows higher levels of MOAB-2, methoxy-X04, and c-Fos compared to the motor cortex (**Fig.S1j-m**). The relevant figures and text are below.

Results

Using immunofluorescence measures, we also observed that posterior regions (VIS), which exhibit earlier signs of hyperactivity in App^{NL-G-F} mice, showed increased markers of both putative neuronal activity (c-Fos) and amyloidosis (methoxy-X04 and MOAB-2) in comparison to frontal regions (MO) (**Fig.S1j-m**).

Supplementary Fig. 1 | Dysregulated resting-state activity at excitatory and inhibitory neurons in a mouse model of amyloidosis.

[...] j, Cortical region showing immunofluorescence labelling of c-Fos (green), methoxy-X04 (magenta), and MOAB-2 (cyan) in App^{NL-G-F} mice. Scale bar: 20 μ m. k-m, Normalised (to V1) fluorescence intensity of c-Fos (k), methoxy-X04 (l), and MOAB-2 (m) in the visual (black) and motor (orange) cortices of 6-8m App^{NL-G-F} mice. [...]

Next, we conducted 2-photon GCaMP6s calcium imaging to measure the activity of excitatory and inhibitory neurons, as well as methoxy-X04 plaque density, in the motor cortex of wild-type and App^{NL-G-F} mice at 4-6 months. Consistent with our mesoscopic imaging, we did not find evidence for increased excitatory or inhibitory cellular activity in motor cortex (**Fig.S1y**). Plaque density in motor cortex was also lower than equivalent timepoints in visual cortex (**Fig.S1z**). Analysis of functional assemblies found a reduction in the size of E-E (**Fig.S1aa**) but not I-E (**Fig.S1ab**) assemblies. Together, these results suggest that cellular and assembly-level inhibition is relatively intact at early timepoints in the motor cortex of App^{NL-G-F} mice, and that plaque load is lower in comparison to more posterior regions. The relevant figures and text are below.

Results

Our mesoscopic imaging data suggested that frontal regions, such as the motor cortex, are more resilient to hyperactivity than posterior regions (**Fig.1a-g**, **Fig.S1a-h**). To test whether activity changes are detectable at the cellular level in these frontal regions, we repeated our cellular imaging experiments by measuring the calcium activity of excitatory and inhibitory neurons, as well as methoxy-X04 plaque density, in motor cortex at 4-6 months (**Fig.S1y**). Consistent with our mesoscopic imaging data, activity levels in both excitatory and inhibitory cells were comparable between App^{NL-G-F} and WT mice (**Fig.S1y**). We also found *in vivo* plaque load to be lower in motor cortex than age-matched timepoints in visual cortex (**Fig.S1z**). Analysis of functional assemblies found a reduction in the size of E-E (**Fig.S1aa**), but not I-E (**Fig.S1ab**) assemblies. Together, this suggests that cellular and assembly-level inhibition is relatively intact at earlier timepoints in the motor cortex of App^{NL-G-F} in comparison to more posterior regions.

Supplementary Fig. 1 | Dysregulated resting-state activity at excitatory and inhibitory neurons in a mouse model of amyloidosis.

[...] **y**, Normalised (to WT) area under the curve (AUC), mean frequency (Freq), and amplitude (Amp) of calcium events at excitatory (left) and inhibitory (right) cells per mouse in 4-6m WT (black) and *App*^{NL-G-F} (blue, red) mice. **z**, Density of methoxy-X04-positive plaques in the motor (left) and visual (right) cortex of 4-6m *App*^{NL-G-F} mice. **aa-ab**, Fraction of functionally associated neurons in 4-6m *App*^{NL-G-F} (blue, red) and WT (black) mice for assemblies comprising excitatory neurons alone (**aa**, blue), and inhibitory and excitatory neurons (**ab**, red). [...]

There is a substantial body of literature (e.g., from the Stevens lab) suggesting that in APP models, microglia approach synapses and may contribute to synaptic dysfunction. The authors report in their spatial transcriptomics results that microglia display the greatest changes, so one wonders if they play a role in the early synaptic changes reported here?

The reviewer again raises a very interesting issue. To investigate this, we first conducted an Over Representation Analysis (ORA) using the DEGs we identified at microglia cells in our spatial transcriptomic analysis. Using this approach, we found evidence for synapse-related biological processes in microglia, including positive regulation of synaptic transmission, regulation of synaptic plasticity, and regulation of synaptic structure/activity (**Fig.S6e-f**). Therefore, we next conducted a ligand receptor (LR) analysis to test for putative cellular interactions between microglia, excitatory neurons, and inhibitory neurons using our spatial transcriptomics data. Our LR interaction analysis suggested that microglia show the strongest interactions with excitatory, rather than inhibitory, neurons at early timepoints (**Fig.S6g**). At later timepoints, microglia showed weaker LR interactions with both excitatory and inhibitory neurons (**Fig.S6h**). Taken together these results suggest that microglia may play a role in the early synaptic changes reported here, although further work beyond the scope of this manuscript is required to investigate this. The relevant figure and text are below.

Results

In agreement with previous work using the CosMx™ system⁶⁶, we found evidence for the greatest number of DEGs at microglial cells across all tested ages (**Fig.S6b-c, Table S12**). Using over-representation analysis (ORA), we found DEGs at microglia to be associated with synaptic regulation in *App*^{NL-G-F}, with ligand-receptor interaction analysis finding most prominent interactions between microglia and excitatory neurons (see **Methods, Fig.S6e-h**).

Supplementary Fig. 6 | Transcriptomic changes during amyloidosis in the App^{NL-G-F} mouse.

[...] **e-f**, Biological processes associated with DEGs in MGL at 3-4m (**e**) and 6-8m (**f**). **g-h**, Percentage change in inferred ligand-receptor interaction strength (estimated based on probability weight) between MGL, INH, and EXC cells in App^{NL-G-F} vs WT mice (red denotes increased and blue denotes decreased in App^{NL-G-F}) at 3-4m (**g**) and 6-8m (**h**). [...]

Regarding Methoxy-X04 labeling, can the authors quantify how many of the total plaques (as labeled by anti-A β antibodies) are Methoxy-X04 positive in the NLGF mice? This is important to understand if the phenotype they report is related to a particular plaque type.

We thank the reviewer for their comment. To address this issue, we conducted additional experiments to label amyloid plaques in App^{NL-G-F} mice with both methoxy-X04 and MOAB-2 (**Fig.S1j**). Using this approach, we found that more than 70% of MOAB-2 positive plaques were also labelled by methoxy-X04 in all analysed mice and brain regions. The relevant text has now been incorporated into the methods section and runs as follows:

Methods

Amyloid-beta plaque quantification *in vivo*

[...] The following day, methoxy-X04 signal was imaged at 720 nm. To estimate amyloid load, plaques were manually identified based on size and morphology^{48,116}. Subsequent immunofluorescence measures (**see Immunofluorescence**) showed that more than 70% of MOAB-2 positive plaques were also labelled by methoxy-X04 in all analysed mice and brain regions.

Synapse imaging:

In Figure 1i, the authors show examples of an excitatory and an inhibitory synapse but do not discuss their correlation. Would one not expect a correlation between overall inhibitory and excitatory synaptic activity based on electrophysiological recordings to maintain synaptic excitatory/inhibitory balance?

We thank the reviewer for their comment but wish to clarify that Fig.1i in the original version of the manuscript relates to somatic calcium activity measured at excitatory and inhibitory neurons, and not synaptic activity, which is instead reported in Figures 3-4 of the original manuscript. In agreement with the reviewer's suggestion, the original version of the manuscript included four figure panels (**Fig.1m,n,p,q**) where we measured correlations between the calcium-mediated somatic activity of excitatory and inhibitory neurons (**Fig.1p-q**). This work found a reduction in the size of functional assemblies comprising excitatory and inhibitory neurons, suggesting dysfunctional E-I balance as the reviewer postulates. We have now added text to the results to better signpost that these data are from somatic measures and more clearly highlight the functional assembly level E-I analysis. The text now runs as follows:

Results

Increased resting-state activity at excitatory neurons could occur in conditions of elevated inhibitory neuronal activity if the functional connectivity between assemblies of excitatory and inhibitory neurons (E-I assemblies) is reduced²⁹. We therefore tested whether functional E-I assemblies were modified in the *App*^{NL-G-F} model (**Fig.1l-q**). To do this, we used a previously published approach to estimate the functional association of E-I assemblies based on pairwise correlations between somatic calcium traces (**Fig.1l,o**)^{44,46,49}. Using this method, positive and significant correlation values are thought to reflect mutual connectivity or shared inputs⁵⁰. First, for each excitatory neuron, we calculated the fraction of associated excitatory neurons in the local cortical region (**see Methods, Fig.1l**). We found fewer functional associations between excitatory neurons during resting-state activity by 6-8m in *App*^{NL-G-F} mice compared to age-matched controls (**Fig.1m-n**). We next calculated the fraction of excitatory neurons associated with local inhibitory neurons (**see Methods, Fig.1o**) and found fewer functional associations between excitatory and inhibitory neurons in 6-8m *App*^{NL-G-F} mice than in age-matched controls (**Fig.1p-q**). These results suggest that resting-state neuronal activity is increased during amyloidosis and functional associations between E-E and I-E assemblies are disrupted.

Fig. 1 | Dysregulated resting-state activity at excitatory and inhibitory neurons during amyloidosis.

[...] **l,o**, Example regions showing excitatory (blue circle) and inhibitory (red circle) neurons (left), and pairwise correlations between calcium traces (right) from assemblies comprised of only excitatory neurons (**l**), or both excitatory and inhibitory neurons (**o**). Scale bars: 10 μ m, 1.0 $\Delta F/F_0$, 20 s. **m,n,p,q**, Fraction of functionally associated neurons in 3-4m (**m,p**) and 6-8m (**n,q**) *App*^{NL-G-F} (blue, red) mice and age-matched WT (black) controls for assemblies comprising excitatory neurons alone (**m-n**, blue), and inhibitory and excitatory neurons (**p-q**, red). [...]

In Figures 1j and 1k, they assess the AUC of calcium signals but do not examine the frequency of these signals. It would be beneficial to analyze whether the frequency differs between groups, as they did e.g. in analyses presented in Figure 3.

We thank the reviewer for their comment and have now added the suggested analysis to investigate the amplitude and frequency of calcium events in somatic measures from excitatory and inhibitory neurons, and how these changes map to the increased activity we measured in *App*^{NL-G-F} mice when considering the AUC in Fig.1j-k.

This analysis found the increased AUC in excitatory (**Fig.S1o,q**) and inhibitory (**Fig.S1p-q**) neurons to be driven by an increase in the amplitude of calcium events at both 3-4 and 6-8 months. In excitatory neurons, the frequency of calcium events was elevated at 3-4m, but then lower than wild-type levels at 6-8m (**Fig.S1r,t**). In inhibitory neurons, the frequency of events was similar to wild-type levels at 3-4m but then dropped to levels lower than wild-type at 6-8m (**Fig.S1s-t**).

We plotted histograms of event amplitude to understand how reductions in the frequency of events at later timepoints could map to increases in the average event amplitude, and ultimately the increase in the total AUC. We found a reduction in the percentage of small amplitude events at excitatory and inhibitory neurons at 6-8m of age (**Fig.S1u-v**). This occurred with a significant increase in the percentage of large amplitude events, resulting in a significant increase in the total AUC.

We thank the reviewer for suggesting this analysis that we think has improved the manuscript, and now include the findings in the results and below:

Results

We found the increased AUC in excitatory and inhibitory neurons to be driven by an increase in the amplitude of calcium events at both 3-4 and 6-8 months (**Fig.S1o-q**). In excitatory neurons, the frequency of calcium events was elevated at 3-4m, but lower than WT levels at 6-8m (**Fig.S1r,t**). In inhibitory neurons, the frequency of events was similar to WT levels at 3-4m but then dropped to levels lower than WT at 6-8m (**Fig.S1s-t**). We investigated event-amplitude distributions to understand how reductions in the frequency of events at later timepoints could map to increases in the average event amplitude, and ultimately the total AUC. We found a reduction in the percentage of small-amplitude events and an increase in the percentage of large-amplitude events at both excitatory and inhibitory neurons at 6-8m of age (**Fig.S1u-v**), resulting in a significant increase in the total AUC.

Supplementary Fig. 1 | Dysregulated resting-state activity at excitatory and inhibitory neurons in a mouse model of amyloidosis.

[...] **o-p**, Average amplitude of calcium events measured in excitatory (**o**, blue) and inhibitory (**p**, red) neurons in *App^{NL-G-F}* mice at 3-6m (left) and 6-8m (right) and age-matched WT (black) animals. **q**, Amplitude of calcium events at excitatory (left) and inhibitory (right) cells per mouse in 3-8m in WT (black) and *App^{NL-G-F}* (blue, red) mice. **r-s**, Frequency of calcium events measured in excitatory (**r**, blue) and inhibitory (**s**, red) neurons in *App^{NL-G-F}* mice at 3-6m (left) and 6-8m (right) and age-matched WT (black) animals. **t**, Frequency of calcium events at

excitatory (left) and inhibitory (right) cells per mouse in 3-8m in WT (black) and *App*^{NL-G-F} (blue, red) mice. **u-v**, Percentage of calcium events of increasing amplitude in excitatory (**u**) and inhibitory (**v**) neurons in 6-8m in WT (black) and *App*^{NL-G-F} (blue, red) mice. [...]

The authors report that they label a “heterogeneous population of inhibitory neurons”, but it would be important to understand if this involves all interneuron subtypes to a similar degree or whether there is a preference for certain subtypes in their in vivo imaging. In the subsequent spatial transcriptomics results, they make claims that PV interneurons are affected first. The authors should clarify which interneuron subtypes are labeled and affected in their study.

We thank the reviewer for their comment. In previous work we found that, of the GCaMP-positive neurons labelled with the mDlx construct, ~ 22% were PV positive and ~60 % were SST positive (Radulescu et al., 2023). To test for the involvement of other GABAergic subclasses, we next conducted 2-photon imaging experiments and analysis using a PV-specific viral targeting strategy to label PV neurons with GCaMP6s and cyRFP (AAV1-S5E2p-cyRFP-GSG-P2A-HIS-GCaMP6s-WPRE) (Terstege et al., n.d.; Vormstein-Schneider et al., 2020) (**Fig.1r**). We first validated the construct using a PV antibody in brain slices of V1 that had been targeted with the viral construct. We found 99.4% of cyRFP positive neurons also expressed PV (**Fig.1s**). Within these PV-positive neurons, we found evidence for an increased frequency and amplitude of calcium events in *App*^{NL-G-F} mice relative to age-matched controls (**Fig.1t**). In addition, elevated PV neuron activity was associated with greater local amyloid plaque load (**Fig.S1u**). We next conducted immunofluorescence experiments labelling PV, SST and c-Fos in the visual cortex of both wild-type and *App*^{NL-G-F} mice at 4-6m (**Fig.1v**). Using this approach, we found increased c-Fos expression in both PV and SST neurons of *App*^{NL-G-F} mice relative to wild-type, with PV neurons showing a greater increase compared to SST neurons. (**Fig.1w-x**). Our data therefore suggest that both PV and SST neurons may show hyperactivity as a feature of amyloid-associate pathophysiology in the *App*^{NL-G-F} mouse model. This is based on evidence of increased c-Fos in both GABAergic subclasses, and elevated calcium activity measured with the mDLx construct (biased toward SST: ~22% PV and ~60% SST) and the S5E2 construct (PV specific). The relevant text and figures are shown below:

Results

The mDlx-based labelling approach targets a heterogenous population of inhibitory neurons⁴⁵. However, our previous work³⁷ and that of others⁴⁶ found a bias toward SST neurons using this enhancer system (~ 22% of GCaMP-expressing neurons were PV positive and ~60 % were SST positive³⁷). Therefore, to test for the involvement of other subclasses, we next conducted 2-photon imaging experiments using a PV-specific viral targeting strategy (AAV1-S5E2p-cyRFP-GSG-P2A-HIS-GCaMP6s-WPRE)^{51,52} to label PV neurons with GCaMP6s and cyRFP in V1 (**Fig.1r**). Immunofluorescence measures validated that 99.4% (505/508 cells) of cyRFP positive neurons also expressed PV (**Fig.1s**). Using 2-photon imaging, we found evidence for an increased frequency and amplitude of calcium events in PV neurons in 4-6m *App*^{NL-G-F} mice relative to age-matched controls (**Fig.1t**). In addition, elevated PV neuron activity was associated with greater local amyloid plaque load (**Fig.1u**). We next conducted immunofluorescence experiments labelling PV, SST, and c-Fos in the visual cortex of WT and *App*^{NL-G-F} mice (**Fig.1v**). Using this approach, we found increased c-Fos expression at both PV and SST neurons in *App*^{NL-G-F} mice by 4-6m, with PV neurons showing a relatively greater increase than SST cells (**Fig.1w-x**). Together, our data suggest that both PV and SST neurons show hyperactivity, based on increased c-Fos in both GABAergic subclasses, and elevated calcium activity measured with the mDLx construct (biased toward SST) and the S5E2 construct (PV-specific).

Fig. 1 | Dysregulated resting-state activity at excitatory and inhibitory neurons during amyloidosis.

[...] **r**, Example region (V1) from *in vivo* 2-photon imaging showing expression of structural (cyRFP, red) and functional (GCaMP6s, green) markers in PV neurons as well as amyloid plaques via methoxy-X04 (magenta). Scale bar: 10 μm . Traces: Example of $\Delta F/F_0$ calcium traces from PV neurons in 4-6m *App*^{NL-G-F} (red) and WT (black) mice during resting-state activity. Scale bars: 1.0 $\Delta F/F_0$, 10 s. **s**, Example cortical region showing immunofluorescence labelling of PV (blue) in cyRFP-expressing neurons (red) and a quantification of double-positive cells in V1 (bottom right). Scale bar: 20 μm . **t**, Normalised (to WT) frequency (left) and amplitude (right) of calcium events in PV neurons in 4-6m *App*^{NL-G-F} (red) and WT (black) mice. **u**, Heatmap of normalised (to WT) frequency (top) and amplitude (bottom) values of PV neurons in regions with increasing methoxy-X04 plaque density in 4-6m *App*^{NL-G-F} mice. **v**, Cortical region showing immunofluorescence labelling of c-Fos (green), PV (red), and SST (cyan). Scale bar: 10 μm . **w**, Normalised (to WT) c-Fos intensity in PV (left, red) and SST (right, cyan) neurons in 4-6m *App*^{NL-G-F} (red, cyan) and age-matched WT (black) mice. **x**, Normalised (to WT) c-Fos intensity in PV (red) and SST (cyan) neurons in 4-6m *App*^{NL-G-F} mice.

Is there a time point in NLGF mice when synaptic activity is normal?

The reviewer raises an interesting issue. We detect changes in cellular and synaptic activity in both excitatory and inhibitory neurons at 3-4 m of age, which is the earliest timepoint we investigated. Recent work has shown no changes in activity levels in superficial layers (L1-4) in *App*^{NL-G-F} mice at earlier timepoints, such as 1.5 months, when plaque levels are very low in this mouse model (Papanikolaou et al., 2025). Consistent with this, we found animals with low levels of plaque pathology to have activity levels that were similar to wild-type mice (**Fig.S1w**). However, it is important to note that, in our study, we imaged at later timepoints and always in regions where plaques were evident, in order to study the impact of the plaque microenvironment on micro-circuit activity. This is not the case in recent work which focused on pre-plaque periods (Papanikolaou et al., 2025). Others have reported synaptic deficits in similar knock-in models in the entorhinal cortex at 2 months of age, suggesting that dysfunctional synaptic activity may be detectable during very early stages of amyloid-beta pathology (Petrache et al., 2019). To address these issues, we compared activity levels from regions with the lowest plaque loads in our data to wild-type activity (**Fig. S1w**) as discussed above. We also added a section to the discussion to cover these points. The relevant text and figures are included here.

Results

Given that the increased activity of excitatory and inhibitory neurons was driven by increases in calcium event amplitude (**Fig.S1o-q**), we next tested the extent to which increases in

amplitude were associated with plaque load at the animal level (**Fig.S1w**). We found that higher plaque load was associated with increased amplitude in both excitatory and inhibitory neurons, while *App*^{NL-G-F} mice with low plaque densities were comparable to WT mice (**Fig.S1w**).

Supplementary Fig. 1 | Dysregulated resting-state activity at excitatory and inhibitory neurons in a mouse model of amyloidosis.

[...] **w**, Heatmap showing normalised (to WT) excitatory (top) and inhibitory (bottom) neuron amplitude in 3-8m *App*^{NL-G-F} mice with increasing plaque load. [...]

Discussion

Recent work has shown no changes in activity in superficial layers (L1-4) in *App*^{NL-G-F} mice at 1.5m, when plaque levels are very low in this mouse model⁹⁷. Consistent with this, we found that mice with low plaque load were similar to WT, suggesting an early-stage resilience which may involve homeostatic processes^{38,72}. However, it is important to note that we imaged at later timepoints and exclusively in regions where plaques were evident, to specifically study the impact of the plaque microenvironment on micro-circuit activity. This is not the case in recent work which focused on pre-plaque periods⁹⁷. Others have reported synaptic deficits in similar knock-in models in the entorhinal cortex by 2m⁷⁶, suggesting that dysfunctional synaptic activity may be detectable during very early stages of amyloid-beta pathology.

In Figure 2, they report primarily reductions in presynaptic markers, which is interesting because previous work from the Malinow group and others labs has shown reductions in postsynaptic markers, such as PSD-95, in both mouse models and human AD tissue. Simply attributing this difference to variations in mouse models is not convincing and does not help move the field forward. The authors should address this discrepancy by exploring possible reasons for the different findings.

We thank the reviewer for their comment and have added new experimental data and text, incorporating the suggested references, to address this issue. In addition to the postsynaptic vulnerability associated with AD-related models and AD tissue, there are several lines of evidence finding presynaptic loss and abnormal physiology in AD-related models (Barthet and Mulle, 2020; Brasnjevic et al., 2013; Rutten et al., 2005; Sanchez-Varo et al., 2021; Stephen et al., 2019; Trujillo-Estrada et al., 2014). One possibility is that the vulnerability/resilience of pre- and/or post-synaptic compartments might be specific to different regions. To test this concept in the *App*^{NL-G-F} mouse line, we added additional experiments and analysis to measure excitatory and inhibitory pre- and post-synaptic puncta in the entorhinal cortex (**Fig.S2d-f**). This is a well-studied area often considered a site of early-stage AD-related pathology (Braak and Braak, 1995; Wang et al., 2010). In this region, we found evidence for both excitatory and inhibitory synapse loss in the *App*^{NL-G-F} mouse (**Fig.S2d**). We again found inhibitory synapse loss to be driven by a reduction in VGAT, but reductions in excitatory synapse density were associated with reductions in both pre- and post-synaptic markers (**Fig.S2e-f**). As such, it appears that the *App*^{NL-G-F} mouse does show postsynaptic vulnerability in certain regions. We therefore conclude that the presynaptic vulnerability that we report in the visual cortex of *App*^{NL-G-F} mice may be due to an interplay between the type of model (which does not involve

overexpression) and regional variability in synaptic vulnerability/resilience. The relevant sections addressing this issue run as follows:

Results

We next tested the extent to which synapse loss in the *App^{NL-G-F}* mouse was also evident in other areas known to be sites of early-stage AD-related pathology such as the entorhinal cortex^{54,55}. Similar to measures in V1, we found evidence for the loss of both excitatory and inhibitory colocalised puncta in entorhinal cortex (**Fig.S2d**). Glutamatergic changes were associated with a loss of both VGLUT1 and PSD-95 (**Fig.S2e**), suggesting that the postsynaptic compartment also shows vulnerability in entorhinal cortex. However, GABAergic changes were again associated with reductions in presynaptic VGAT levels without changes in gephyrin density (**Fig.S2f**). These results suggest that presynaptic vulnerability is a common feature in the *App^{NL-G-F}* mouse across the investigated regions, and is more prominent at GABAergic compared to glutamatergic synapses.

Supplementary Fig. 2 | Presynaptic GABAergic vulnerability in *App^{NL-G-F}* mouse cortex.

[...] **d**, Normalised (to WT) colocalised glutamatergic (left) and GABAergic (right) synaptic puncta in 3-8m *App^{NL-G-F}* (blue, red) and age-matched WT (black) mice in the entorhinal cortex (ENT). **e-f**, Normalised (to WT) density of excitatory (**e**) and inhibitory (**f**) pre- (left) and post- (right) synaptic proteins in 3-8m *App^{NL-G-F}* (blue, red) and age-matched WT mice (black) in the entorhinal cortex (ENT). [...]

Discussion

AD-related pathology has been associated with the loss and dysfunction of dendritic spines⁸³⁻⁸⁷ and axonal boutons^{36,88-91}, as well as asymmetries in pre- and post-synaptic dynamics⁹². We found loss of excitatory and inhibitory presynaptic markers across all studied brain regions, supporting the earlier vulnerability of this compartment. Consistent with previous work^{26,78,93}, PSD-95 loss and functional alterations at excitatory spines were also observed in our data, although less pronounced. Importantly, we only investigated a small fraction of synaptic proteins^{4,94}, so future studies with greater multiplexing capability may reveal additional synaptic vulnerabilities. Nevertheless, our measures of synaptic puncta, structure, and function suggest presynaptic GABAergic vulnerability as an early feature of amyloid-associated pathophysiology in line with previous work^{95,96}. This conclusion is also supported by work in other knock-in mouse models and human *postmortem* samples, showing reductions in presynaptic density and function, as well as A β accumulation at inhibitory boutons^{32,76}.

They show that the levels of c-Fos are elevated in NLGF mice but VGAT is reduced at c-Fos-positive neurons, suggesting that loss of GABAergic presynaptic puncta may be associated with elevated neuronal activity levels in early-stage amyloidosis. However, this is not convincing, as there are many reasons for c-Fos changes in NLGF mice. More direct analyses of neuronal activity, perhaps combining functional assays with

post-hoc labeling, would strengthen this claim and provide clearer evidence for the relationship between VGAT reduction and elevated neuronal activity. Indeed, previous work has shown reduced or no changes in c-FOS in several APP models vs. controls, which suggest that this technique is quite variable and also brain region-dependent.

We thank the reviewer for their comment. To address this issue, we revised the analysis of our cellular and inhibitory bouton basket imaging data which tested for inhibitory bouton activity that was spatially and temporally associated with calcium activity in proximal excitatory neurons.

We again used an event-triggered averaging approach, but this time correlated the inhibitory bouton activity with the level of excitatory neuronal activity. Consistent with our c-Fos and VGAT data, we found that inhibitory bouton activity in *App^{NL-G-F}* animals was predominantly reduced around highly active excitatory neurons. We have combined these data with that of figure 5 and thank the reviewer for their helpful suggestion. The relevant text and figure now run as follows.

Results

Presynaptic GABAergic loss and dysfunction are associated with highly active excitatory neurons

[...] We next investigated the relationship between changes in the activity of GABAergic boutons and spatially proximal excitatory neurons *in vivo*. To do this, we returned to our co-expression experiments, which used the mDlx enhancer system to label a heterogeneous population of inhibitory neurons with both a functional (GCaMP6s) and a structural (cyRFP) marker^{45,46} and a second construct to label excitatory neurons with GCaMP6s and mCherry (see Methods, Fig.1h-i). Using this approach, we were able to observe putative baskets of GABAergic boutons (expressing both cyRFP and GCaMP6s) surrounding the soma of excitatory neurons (expressing GCaMP6s and mCherry) in superficial layers of visual cortex (Fig.5e). We used *in-vivo* 2-photon calcium imaging to simultaneously image, and separate, the calcium-mediated neuronal activity of excitatory neurons and their adjacent inhibitory boutons in 6-8m WT and *App^{NL-G-F}* mice (Fig.5e-f). We first used correlation-based analysis to test the functional properties of GABAergic boutons in relation to local excitatory neuronal activity (see Methods). We found the activity of GABAergic boutons at axonal baskets was less correlated with its spatially associated excitatory neuron activity in *App^{NL-G-F}* mice compared to age-matched WT animals (Fig.5g). We next used event-triggered averaging to test for changes in GABAergic bouton activity that were spatially and temporally associated with activity levels in the adjacent excitatory neuron (see Methods, Fig.5f). We identified calcium transients in excitatory neurons and generated average inhibitory bouton signals time-locked to the transients in excitatory neurons (Fig.5f). We found that, in *App^{NL-G-F}* animals, the activity of inhibitory boutons that were spatially and temporally associated with highly active excitatory neurons was reduced relative to equivalent values in WT animals (Fig.5h). These data suggest that reductions in inhibitory bouton activity are associated with elevated neuronal activity levels in early-stage amyloidosis.

Fig. 5 | Hyperactivity is associated with GABAergic dysfunction and abnormal oscillatory activity in a mouse model of amyloidosis

[...] **e**, Top left: Example of GABAergic boutons labelled with mDlx-CyRFP (red) in proximity to a EF1a-GCaMP6s-expressing excitatory neuron (green). Top right: GCaMP6s excitatory cellular signal in isolation. Bottom left: mDlx-cyRFP structural bouton signal in isolation. Bottom right: structural bouton channel (red) merged with mDlx-GCaMP6s (green) showing bouton-specific activity (white arrowheads). Scale bar 10 μm . **f**, Example traces generated using event-triggered averaging from GCaMP6s transients in the excitatory neuron, showing an average waveform from the excitatory neuron (blue) and inhibitory boutons proximal to this cell (red). Dashed line depicts the time of peak excitatory activity. Scale bars: 0.2 $\Delta F/F_0$ and 0.5 s. **g**, Correlation of GABAergic bouton activity with local excitatory neuron activity in 6-8m App^{NL-G-F} (red) and WT (black) mice. **h**, Z-scored (to WT) amplitude of GABAergic boutons spatially and temporally associated with neurons of increasing frequency in 6-8m App^{NL-G-F} (red) and age-matched WT (black) mice. [...]

In Figure 3d, it would be helpful if the authors could be more explicit about which specific areas are being analyzed.

We thank the reviewer for their comment and now include text to clarify that this is in the visual cortex.

Results

To test between these scenarios, we measured *in vivo* calcium signals from presynaptic GABAergic boutons in the amyloid plaque microenvironment using 2-photon microscopy. To do this, we again utilised the mDlx enhancer strategy to express both cyRFP and GCaMP6s in superficial axons and boutons of V1 in 3-4m and 6-8m App^{NL-G-F} and WT mice (see **Methods, Fig.3a**). Using this approach, we obtained structural and functional measures of GABAergic axonal boutons *in vivo* and spatially related them to local amyloid pathology via methoxy-X04 labelling (see **Methods**).

We first measured the density of GABAergic boutons *in vivo* (see **Methods**) and found that bouton density was stable at 3-4m (**Fig.3b**) but reduced by 6-8m (**Fig.3c**) in App^{NL-G-F} mice when compared to WT. GABAergic bouton density showed the greatest reduction in areas with high plaque load (**Fig.3d**).

Presenting analyses on a single-animal basis would also allow readers to appreciate the variability in the NLGF models.

As suggested by the reviewer, we now include data at the single animal level. We added this analysis throughout the manuscript but include an example below.

Supplementary Fig. 1 | Dysregulated resting-state activity at excitatory and

inhibitory neurons in a mouse model of amyloidosis.

[...] **n**, Area under the curve (AUC) of the $\Delta F/F_0$ trace of excitatory (left) and inhibitory (right) cells per mouse in 3-8m in WT (black) and App^{NL-G-F} (blue, red) mice. [...] **q**, Amplitude of calcium events at excitatory (left) and inhibitory (right) cells per mouse in 3-8m in WT (black) and App^{NL-G-F} (blue, red) mice. [...] **t**, Frequency of calcium events at excitatory (left) and inhibitory (right) cells per mouse in 3-8m in WT (black) and App^{NL-G-F} (blue, red) mice. [...]

They observe a surprising “normalization” in bouton frequency from 3–4 to 6–8 months, which is interesting. Why is that? Is there a subsequent reduction at later time points?

We thank the reviewer for raising this interesting point. The reviewer is correct that population level measures of calcium activity from GABAergic boutons show a reduction in frequency at 3-4 m in App^{NL-G-F} mice (**Fig.3e**) but these values are then similar to wild-type levels at 6-8 m (**Fig.3g**).

The reviewer makes the interesting suggestion that this might be a normalization-like effect. One possibility is that those boutons with very low frequency scores are lost across this time period. We suggest this is a possible scenario as bouton density is similar to wild-type levels at 3-4m (**Fig.3b**) but reduced at 6-8m (**fig.3c**). We have added a section to the discussion to address this possibility; the relevant text now runs as follows:

Discussion

Future work may use repeated longitudinal *in vivo* approaches to track the activity of individual GABAergic boutons across AD-related pathology. This approach may test if population-coupled boutons with weaker activity are ultimately lost and investigate if surviving boutons show homeostasis-like (e.g. increases in size/strength⁷⁴) or maladaptive responses.

We also include here for the reviewer an extract from a different manuscript in preparation where we longitudinally track the structure and function of GABAergic boutons in App^{NL-G-F} mice (a). After two weeks, we classify them into persistent and lost boutons and compare their activity profiles at baseline. In this study, we observed that boutons that disappear within two weeks are more likely to show low-frequency activity at baseline (b). These findings support our proposed rationale for bouton normalization between 3–4 and 6–8 months, which we attribute to a survivor bias favouring high-frequency boutons.

Manuscript in preparation:

Longitudinal activity changes in GABAergic boutons in App^{NL-G-F} mice.

a, Diagram depicting experimental design for longitudinal two-photon imaging and example axon imaged at baseline and week 2. White arrows depict persistent boutons while pink arrow shows a bouton that is lost. Scale bar: 1 μm. **b**, Probability density distributions of GABAergic bouton frequency in persistent (black) vs lost (pink) boutons in 3-6-month-old App^{NL-G-F} mice.

Spatial transcriptomics:

On the surface the spatial transcriptomics results seem to nicely support the imaging data; however, there appears to be an overextension in using this data to infer mechanistic insights into the synaptic changes. In Figure 6e, it seems that they find almost as many SST and (slightly fewer) VIP neurons as PV neurons, but in reality, PV interneurons typically constitute a larger proportion of inhibitory neurons. Why is that? What is the independent validation that they have assigned cell types/classes correctly in the UMAP plots?

We thank the reviewer for their observation and appreciate that we should have better signposted the related Statistical Table (S6) where we report the total number of cells identified for each inhibitory subclass. We identify a higher number of PV (3-4m: 2059; 6-8m: 2284) compared to SST (3-4m: 1306; 6-8m: 1483) interneurons in both age groups. The number of VIP (3-4m: 661; 6-8m: 696) and LAMP5 (3-4m: 595; 6-8m: 589) cells is smaller than that of SST cells. In addition, to clarify the inhibitory subclass identity, we have now included an expression matrix of the top 10 marker genes for each inhibitory subclass (**Fig. S6i**). This is also reported in the Statistical Table.

Supplementary Fig. 6 | Transcriptomic changes during amyloidosis in the

App^{NL-G-F} mouse.

[...] i, Normalised expression of top 10 marker genes per GABAergic cell subclass. [...]

They also claim that PV interneurons show the earliest and most pronounced changes compared with other cell types, as highlighted in the abstract. However, the figure suggests there are changes across all interneuron subtypes. Using the percentage of DEGs as the main criterion to claim that one cell type is more or earlier affected than others is not convincing. The authors need to temper these claims or provide additional experimental evidence to support the preferential vulnerability of PV interneurons.

We thank the reviewer for their comment and have revised the text to more carefully describe the key findings from the spatial transcriptomics work. We have also revised our language to temper the claims relating to PV and appreciate that we should have been more careful in reporting these findings in the abstract (now revised). We agree with the reviewer that using the percentage of DEGs as the sole criterion is unconvincing and have revised the text to clarify that our analysis strategy combined both DEG counts and over-representation analysis (ORA).

In the text we state that, 'We found the greatest level of DEGs at 3-4m were focused at PV neurons, followed by SST neurons, with much lower levels in LAMP5 and VIP neurons (**Fig.6f, S6j, Table S6**). We next conducted an ORA⁶⁸ to investigate the putative functions of DEGs at GABAergic subclasses (see **Methods**). At 3-4m, we found enriched biological processes exclusively in the PV population (**Fig.6g, Table S6**).'

Together, these analyses indicate that PV cells show the highest number of DEGs at early timepoints (**Fig. 6f**) and that only DEGs at PV cells are significantly associated with biological processes (notably related to synaptic function) (**Fig. 6g**). This suggests that transcriptional alterations in PV neurons are not only more numerous but also more functionally coherent in comparison to the other studied subclasses. We also now include a more systematic analysis of shared and unique DEGs at each subclass, showing that ~41% of DEGs are unique to PV cells (**Fig.S6j**). We have therefore revised the abstract to state the following:

Abstract

We propose that early-stage AD-related synaptic pathophysiology is focused at population-coupled synapses, with molecular measures implicating abnormal synaptic processing as an early-stage feature in parvalbumin-positive interneurons.

Our analysis at later timepoints found changes identified with the ORA across all studied GABAergic neuron subclasses: 'At 6-8m, changes across GABAergic subclasses were more uniform, so that all subclasses showed a similar fraction of DEGs (**Fig.6h, Table S6**) and included common biological processes associated with the regulation of synaptic plasticity, membrane potential, vesicle-mediated transport in synapses, and dendrite development (**Fig.6i, Table S6**).'

However, we found the greatest covariance of DEGs with local amyloid pathology to occur with both PV and SST neurons: 'We found that all GABAergic subclasses showed increasing levels of DEGs with high plaque load (**Fig.6k**). However, PV and SST interneurons were associated with more than double the percentage of DEGs that covaried with regional plaque load compared to VIP and LAMP5 subclasses (**Fig.6l, S6k-m, Tables S6, S12**).'

These data suggest that, although the earliest changes we can detect using ORA are in PV cells, SST cells may also be sensitive to amyloid pathology when analysis of covariance against local pathology is considered.

As suggested by the reviewer, we also added additional experimental evidence to further investigate the involvement of different GABAergic subclasses. We used a PV-specific viral targeting strategy to label PV interneurons with GCaMP6s and cyRFP (AAV1-S5E2p-cyRFP-GSG-P2A-HIS-GCaMP6s-WPRE)(Terstege et al., n.d.; Vormstein-Schneider et al., 2020) and conducted 2-photon imaging of PV positive interneurons and local methoxy-X04 positive plaques in *App*^{NL-G-F} mice and wild-type mice (**Fig.1r**). Using this approach, we found evidence for an increased amplitude and frequency of PV neurons in *App*^{NL-G-F} mice relative to age-matched controls (**Fig.1t**). We also ran an immunofluorescence experiment to determine c-Fos levels within PV and SST cells (**Fig.1v**). We found that c-Fos levels were increased in both interneuron subtypes in *App*^{NL-G-F} mice when compared to wild-type controls, with PV neurons showing a greater increase compared to SST neurons. (**Fig.1w-x**).

Our data therefore suggest that although the ORA analysis points toward the greatest synapse-related dysregulation at early-time points in PV cells, both PV and SST neurons may show hyperactivity as a feature of amyloid-associated pathophysiology in the *App*^{NL-G-F} mouse model. This is based on evidence of increased c-Fos in both GABAergic subclasses, and elevated calcium activity measured with the mDlx construct (biased toward SST) and the S5E2 construct (PV-specific). The relevant sections are covered below.

Results

We found the greatest level of DEGs at 3-4m were focused at PV neurons, followed by SST neurons, with much lower levels in LAMP5 and VIP neurons (**Fig.6f, S6j, Table S6**). We next conducted an ORA⁶⁸ to investigate the putative functions of DEGs at GABAergic subclasses

(see Methods). At 3-4m, we found enriched biological processes exclusively in the PV population (Fig.6g, Table S6). Relevant enriched categories were mostly associated with synaptic processes, including the downregulation of transcripts linked to GABAergic receptors and transmission (*Gabra2*, *Gabrb2*, *Nf1*, *Pten*), and the upregulation of genes involved in GABA synthesis (*Gad1*, *Glu1*), synaptic vesicle cycle (*ApoE*, *Bin1*, and *Sncb*), and calcium signalling (*Atp2b2* and *Calm1*) (Fig.6g, Table S6). [...] Together, these results suggest that early synaptic processing-related changes in PV neurons are followed by more widespread alterations across all tested GABAergic subclasses.

Supplementary Fig. 6 | Transcriptomic changes during amyloidosis in the *App^{NL-G-F}* mouse.

[...] j, Percentage of DEGs shared across different GABAergic subclasses. Empty overlap denotes 0%. [...]

Results

The mDlx-based labelling approach targets a heterogenous population of inhibitory neurons⁴⁵. However, our previous work³⁷ and that of others⁴⁶ found a bias toward SST neurons using this enhancer system (~ 22% of GCaMP-expressing neurons were PV positive and ~60 % were SST positive³⁷). Therefore, to test for the involvement of other subclasses, we next conducted 2-photon imaging experiments using a PV-specific viral targeting strategy (AAV1-S5E2p-cyRFP-GSG-P2A-HIS-GCaMP6s-WPRE)^{51,52} to label PV neurons with GCaMP6s and cyRFP in V1 (Fig.1r). Immunofluorescence measures validated that 99.4% (505/508 cells) of cyRFP positive neurons also expressed PV (Fig.1s). Using 2-photon imaging, we found evidence for an increased frequency and amplitude of calcium events in PV neurons in 4-6m *App^{NL-G-F}* mice relative to age-matched controls (Fig.1t). In addition, elevated PV neuron activity was associated with greater local amyloid plaque load (Fig.1u). We next conducted immunofluorescence experiments labelling PV, SST, and c-Fos in the visual cortex of WT and *App^{NL-G-F}* mice (Fig.1v). Using this approach, we found increased c-Fos expression at both PV and SST neurons in *App^{NL-G-F}* mice by 4-6m, with PV neurons showing a relatively greater increase than SST cells (Fig.1w-x). Together, our data suggest that both PV and SST neurons show hyperactivity, based on increased c-Fos in both GABAergic subclasses, and elevated calcium activity measured with the mDlx construct (biased toward SST) and the S5E2 construct (PV-specific).

Fig. 1 | Dysregulated resting-state activity at excitatory and inhibitory neurons during amyloidosis.

[...] **r**, Example region (V1) from *in vivo* 2-photon imaging showing expression of structural (cyRFP, red) and functional (GCaMP6s, green) markers in PV neurons as well as amyloid plaques via methoxy-X04 (magenta). Scale bar: 10 μm . Traces: Example of $\Delta F/F_0$ calcium traces from PV neurons in 4-6m *App*^{NL-G-F} (red) and WT (black) mice during resting-state activity. Scale bars: 1.0 $\Delta F/F_0$, 10 s. **s**, Example cortical region showing immunofluorescence labelling of PV (blue) in cyRFP-expressing neurons (red) and a quantification of double-positive cells in V1 (bottom right). Scale bar: 20 μm . **t**, Normalised (to WT) frequency (left) and amplitude (right) of calcium events in PV neurons in 4-6m *App*^{NL-G-F} (red) and WT (black) mice. **u**, Heatmap of normalised (to WT) frequency (top) and amplitude (bottom) values of PV neurons in regions with increasing methoxy-X04 plaque density in 4-6m *App*^{NL-G-F} mice. **v**, Cortical region showing immunofluorescence labelling of c-Fos (green), PV (red), and SST (cyan). Scale bar: 10 μm . **w**, Normalised (to WT) c-Fos intensity in PV (left, red) and SST (right, cyan) neurons in 4-6m *App*^{NL-G-F} (red, cyan) and age-matched WT (black) mice. **x**, Normalised (to WT) c-Fos intensity in PV (red) and SST (cyan) neurons in 4-6m *App*^{NL-G-F} mice.

They have 8 mice per group and can they please clarify if groups were sex-balanced in the transcriptomics analyses?

We thank the reviewer for their comment. Yes, we have used mixed sex across our genotypes and have added text to the Supplementary Tables to address this issue.

Conclusion:

While the manuscript presents interesting novel data and offers some insights into synaptic and network dysfunction in an APP mouse model, several issues need to be addressed to strengthen the validity and impact of the findings. Providing more direct evidence for the role of A β , improving data analyses and statistics, and ensuring that the main conclusions are fully supported by the data would significantly improve the manuscript.

We thank the reviewer for their comments and have added the experimental work and analysis as suggested. We would like to thank the reviewer for their suggestions which we think have greatly improved the manuscript.

Reviewer #2 (Remarks to the Author):

This manuscript by Melgosa-Ecenarro et al. examined cell-type-specific single-synapse pathophysiology associated with amyloidosis in an Alzheimer's disease (AD) using a second-generation knock-in AD mouse model, AppNL-G-F. In cortical regions with dysregulated meso-scale activity (elevated resting-state activity) at early stages of amyloidosis, by using in vivo two-photon calcium imaging the authors found disrupted functional associations between excitatory and inhibitory neuron ensembles and selective functional weakening of GABAergic presynaptic activity proximal to plaques and showing strong coupling with population activity. Using spatial transcriptomics, they further discovered that parvalbumin (PV) interneurons among all subclasses of GABAergic neurons have the earliest transcriptomic changes that contribute to the dysregulation of synaptic processes. This study has thus provided important insights for understanding microcircuit-level mechanisms underlying aberrant network function in AD-related pathology, highlighting an early vulnerability of PV inhibitory neurons. The study is comprehensive, and the data are in general convincing. However, there are multiple issues that need to be clarified.

Major points:

1. Fig. 2d-e, the authors separately counted VGUT/VGAT and PSD95/GEPH puncta. Are they suggesting that some presynaptic functions (or structures) are lost without much affecting the postsynaptic function/structure? Please explain here.

We thank the reviewer for their comment and should clarify that we measured colocalised VGLUT/PSD95 (excitatory) and VGAT/GEPH (inhibitory) puncta. We have added text to the results and discussion to address this issue. We do see evidence for presynaptic loss, ahead of major changes in post-synaptic features in visual cortex. Previous work has suggested that presynaptic elements can be lost earlier than post-synaptic elements in certain models of AD-related pathology (Stephen et al., 2019). The relevant section of the discussion, together with the new section on synaptic marker quantification in the entorhinal cortex, are included below:

Results

We next tested the extent to which synapse loss in the *App^{NL-G-F}* mouse was also evident in other areas known to be sites of early-stage AD-related pathology such as the entorhinal cortex^{54,55}. Similar to measures in V1, we found evidence for the loss of both excitatory and inhibitory colocalised puncta in entorhinal cortex (**Fig.S2d**). Glutamatergic changes were associated with a loss of both VGLUT1 and PSD-95 (**Fig.S2e**), suggesting that the postsynaptic compartment also shows vulnerability in entorhinal cortex. However, GABAergic changes were again associated with reductions in presynaptic VGAT levels without changes in gephyrin density (**Fig.S2f**). These results suggest that presynaptic vulnerability is a common feature in the *App^{NL-G-F}* mouse across the investigated regions, and is more prominent at GABAergic compared to glutamatergic synapses.

Supplementary Fig. 2 | Presynaptic GABAergic vulnerability in *App^{NL-G-F}* mouse cortex.

[...] **d**, Normalised (to WT) colocalised glutamatergic (left) and GABAergic (right) synaptic puncta in 3-8m *App^{NL-G-F}* (blue, red) and age-matched WT (black) mice in the entorhinal cortex (ENT). **e-f**, Normalised (to WT) density of excitatory (**e**) and inhibitory (**f**) pre- (left) and post- (right) synaptic proteins in 3-8m *App^{NL-G-F}* (blue, red) and age-matched WT mice (black) in the entorhinal cortex (ENT). [...]

Discussion

AD-related pathology has been associated with the loss and dysfunction of dendritic spines^{83–87} and axonal boutons^{36,88–91}, as well as asymmetries in pre- and post-synaptic dynamics⁹². We found loss of excitatory and inhibitory presynaptic markers across all studied brain regions, supporting the earlier vulnerability of this compartment. Consistent with previous work^{26,78,93}, PSD-95 loss and functional alterations at excitatory spines were also observed in our data, although less pronounced. Importantly, we only investigated a small fraction of synaptic proteins^{4,94}, so future studies with greater multiplexing capability may reveal additional synaptic vulnerabilities. Nevertheless, our measures of synaptic puncta, structure, and function suggest presynaptic GABAergic vulnerability as an early feature of amyloid-associated pathophysiology in line with previous work^{95,96}. This conclusion is also supported by work in other knock-in mouse models and human *postmortem* samples, showing reductions in presynaptic density and function, as well as A β accumulation at inhibitory boutons^{32,76}.

2. Fig. 2f, please show example zoomed-in images with both cFos-EYFP and VGAT signals in wildtype and mutant mice.

We have added example images to address this issue, which are shown below.

Fig. 5 | Hyperactivity is associated with GABAergic dysfunction and abnormal oscillatory activity in a mouse model of amyloidosis

a, Cartoon of experimental approach depicting viral-mediated c-Fos-based activity tagging, with example images showing immunofluorescence labelling of VGAT (red) and c-Fos-EYFP-positive (green) cell (left) and dendritic spines (right) in 3-8m WT (top) and *App^{NL-G-F}* (bottom) mice. Scale bars: 5 μ m (left), 2 μ m (right). [...]

3. Fig. 2h, it is not clear how c-Fos intensity was Z-scored.

We thank the reviewer for their comment. Cellular c-Fos intensity scores in *App^{NL-G-F}* animals were z-scored using the mean and standard deviation of the age-matched wild-type values. We have now added text to the methods section to better communicate this approach. We would also like to highlight that we have moved figure 2h to become 5c.

Methods

To enable pooling and comparisons, c-Fos and synaptic measurements were z-scored using the mean and standard deviation of the corresponding region- and age-matched WT distribution.

4. Line 239, “...or, alternatively, persistent synapses may exhibit compensatory changes in strength to offset local synapse loss”. Wouldn’t a general increase in the density of presynaptic puncta be expected if it is a compensatory change to offset some local synapse loss?

The reviewer makes an interesting point relating to synaptic functional compensation following structural loss. In previous work we have shown that loss of postsynaptic dendritic spines can be associated with compensatory increases in the size/strength of the remaining local dendritic spines in adult wild-type animals (Barnes et al., 2022, 2017). However, there is very little work testing for similar changes in presynaptic axons. We now highlight this in the discussion. The relevant text now runs as follows:

Discussion

Future work may use repeated longitudinal *in vivo* approaches to track the activity of individual GABAergic boutons across AD-related pathology. This approach may test if population-coupled boutons with weaker activity are ultimately lost and investigate if surviving boutons show homeostasis-like (e.g. increases in size/strength⁷⁴) or maladaptive responses.

5. Line 259, “GABAergic boutons with reduced and increased activity could be found within the same axon...”, please clarify compared to what reduced and increased activity could be found. Does this only reflect the fact that there were natural variations among different boutons within the same axon in terms of calcium activity rather than “bouton-specific” alterations?

We thank the reviewer for their comment. To clarify, we z-scored the activity of each bouton against the wild-type average and standard deviation, to determine the extent to which the activity of a given bouton was increased or decreased relative to the wild-type average. Using this approach, we could approximate the extent to which the changes we observed were greater than the natural variation seen in the wild-type measures. We have added additional text to the methods to clarify this issue. The text now runs as follows:

Methods

Traces from each bouton and axon were extracted and analysed using custom-written MATLAB code. Traces were normalised to the background fluorescence (300 frame running average) to calculate $\Delta F/F_0$ _bouton and $\Delta F/F_0$ _axon (Fig.S3a). To isolate bouton-specific responses, the $\Delta F/F_0$ _axon was subtracted from each $\Delta F/F_0$ _bouton, and a 20% threshold was applied (Fig.S3a). Bouton activity was z-scored to the mean and standard deviation of the age-matched WT population.

6. Fig. 4 and S4 shows an overall decrease in the postsynaptic activity of excitatory synapses of excitatory neurons. However, this result is inconsistent with Fig.1 which shows a global increase in the network activity at 6-8m. Please explain.

We thank the reviewer for their comment. The reviewer is correct that the total activity level as estimated by the area under the curve (AUC) of the calcium trace at dendritic spines is similar to wild-type levels at 3-4 m (Fig.S4a) but reduced in *App*^{NL-G-F} mice at 6-8m (Fig.S4b). This is driven by a reduction in the amplitude of spine events at 6-8m (Fig.4e). In contrast, the cellular activity of excitatory neurons is elevated (Fig.1j). The increased somatic cellular activity is likely driven by conditions of reduced inhibition, and could be reconciled with the spine activity findings by considering the frequency of calcium events in spines, which is increased at both 3-4m (Fig.4b) and 6-8m (Fig4d) in *App*^{NL-G-F}. We have now added a section to the discussion to cover these possibilities. The relevant text now runs as follows:

Discussion

Early work using amyloid overexpression models reported decreases in excitatory synaptic function^{25–27}. Recent studies in *App*^{NL-G-F} mice^{76,77} and younger APP/PS1 mice⁷⁸ found evidence for impaired glutamate reuptake and increased synaptic excitation. In agreement, we observed increased calcium-event frequency in excitatory dendritic spines. At later timepoints, this increase in frequency occurred alongside reductions in event amplitude, resulting in a net decrease in spine activity (AUC). The greatest reductions in spine activity occurred in population-coupled spines proximal to plaques. This suggests that both excitatory synaptic hyper- and hypo-activity may co-exist during different stages of amyloidosis. However, our data suggest that such excitatory synaptic changes are preceded and exceeded by reductions in GABAergic bouton activity, potentially disrupting the local excitatory/inhibitory balance and contributing to microcircuit hyperactivity. We observed VGAT reductions around c-Fos-positive excitatory cells together with a functional uncoupling of E-I assemblies and GABAergic baskets, supporting impaired inhibition as an early mediator of amyloid-associated hyperactivity^{79–82}.

7. Imaging was performed under anesthesia, which can affect cortical activity and neuronal synchrony. Discussion on this caveat should be added.

We have added a caveat to the discussion to address this, and additional data in awake animals which support our findings relating to increased neuronal activity levels in posterior regions of *App*^{NL-G-F} mice when compared to wild-types (**Fig.5i-p**). The relevant text and figures are shown below.

Discussion

For instance, previous work has found that circuit dysfunction in AD models can be influenced by anaesthesia¹⁰⁶. Here, we found hyperactivity in posterior cortical regions (VIS and RSC) to be present in both awake and anaesthetised conditions. In contrast, we did not find robust evidence of hyperactivity in frontal regions of anesthetised or awake *App*^{NL-G-F} mice. Further work is required to understand the extent to which brain-state and behavioural features modulate AD-related pathophysiology.

Results

To address this issue, we combined widefield 1-photon mesoscopic calcium imaging with simultaneous EEG recordings in awake *Thy1-GCaMP6s* mice and *App*^{NL-G-F}*xThy1-GCaMP6s* crosses (see **Methods, Fig.5i**). In agreement with our earlier mesoscopic calcium imaging work (**Fig.1d-e**), we found increased resting-state calcium-mediated neuronal activity only in the posterior cortical regions (VIS and RSC) of awake *App*^{NL-G-F} mice when compared to controls (**Fig.5j, S5d**). We then used event-triggered averaging of mesoscopic calcium events from visual cortex and measured temporally associated local field potential (LFP) recordings from a superficial electrode (**Fig.5k**). We investigated the power of EEG frequency bands during the pre-event baseline and the rising phase of the calcium event (**Fig.5k-l**). Following our imaging results, we observed increased power in *App*^{NL-G-F} mice, spanning all measured frequency bands in the baseline phase immediately before the onset of a calcium event (**Fig.5m**).

Fig. 5 | Hyperactivity is associated with GABAergic dysfunction and abnormal oscillatory activity in a mouse model of amyloidosis

[...] **i**, Schematic (top) and example traces (bottom) of simultaneous superficial local field potential recordings (LFP, light grey) and 1-photon mesoscopic imaging of calcium-mediated activity (black) in awake mice. Scale bars: 2.0 $\Delta F/F_0$, 400 nV, and 5 s. **j**, Normalised frequency, amplitude, and total activity (freq x amp) averaged per mouse in awake *Thy-1-GCaMP6s* (black) and *App^{NL-G-F} x Thy-1-GCaMP6s* (blue) visual cortex. **k**, Example LFP trace (top) associated with calcium transient (bottom). Red dashed lines denote time periods used for baseline and rising phase. **l**, Time-frequency power plot using wavelet convolution and log values converted in WT (top) and *App^{NL-G-F}* (bottom) mice. Dashed lines denote baseline and rising phase described in **k**. **m**, Mean Welch's power spectral density (PSD) estimates in 10 Hz bands during the baseline shown in **k-l**. [...]

Supplementary Fig.5 | *App^{NL-G-F}* mice show increased activity in posterior cortical regions.

[...] **d**, Normalised calcium-mediated activity (freq x amp) per mouse in visual (VIS), retrosplenial (RSC), somatosensory (SS) and motor (MO) areas at 6-8m in *Thy-1-GCaMP6s* (black) and *App^{NL-G-F} x Thy-1-GCaMP6s* (blue) awake mice.

Reviewer #3 (Remarks to the Author):

Re: Melgosa-Ecenarro et al. , early-stage amyloidosis selectively weakens population-coupled synaptic activity in vivo.

The results presented here represent a nice advancement in our understanding of global circuit changes in an amyloidosis model. Also, as stated in the manuscript, synapse loss is a hallmark of cognitive decline in Alzheimer's disease (AD), but it represents only a partial effect, with perhaps 30–40% of synapses lost during the disease's progression. This late-stage manifestation likely follows earlier synaptic

dysfunction phases, which may vary between global impacts on all synapses in amyloid-affected areas or targeted effects on specific synapse subsets. The authors observe early increases in resting state activity in the visual cortex, and later increases in the retrosplenial and somatosensory cortices, with increases in both inhibitory and excitatory neuron populations and interestingly, they observed reductions in both VGLUT1 and VGAT signals. They also found that greatest number of Differentially Expressed Genes were in PV neurons. The work contains intriguing findings and is well-written, but several concerns were noted.

Main points:

-Here the authors found broad changes in cortical circuit activity using 1P mesoscale GECI imaging in APP-KI mice. Interestingly, they found early/more robust increases in circuit activity in the posterior cortical regions, including V1, but resilience in the frontal/M1 regions. The authors also observed simultaneously increased activity of both excitatory and inhibitory neurons in the V1 region using 2P imaging (except not in their axons?). There are some concerns arising here, specifically:

1) Prior studies found that interneuron activity either increased or decreased in an amyloidosis model depending on the interneuron subtype (Algamal et al. 2022, Communications Biology). B/c the authors are averaging across all GABAergic types here, this could be misleading as to mechanisms at the single-cell level. Recommend repeating the V1 experiment using PV and/or Sst-specific tools.

We thank the reviewer for their comment, and have conducted the suggested experiments. In previous work we found that, of the GCaMP-positive neurons labelled with the mDlx construct, ~22% were PV positive and ~60% were SST positive (Radulescu et al., 2023). To test for the involvement of other GABAergic subclasses, and as suggested by the reviewer, we next conducted 2-photon imaging experiments and analysis using a PV-specific viral targeting strategy to label PV neurons with GCaMP6s and cyRFP (AAV1-S5E2p-cyRFP-GSG-P2A-HIS-GCaMP6s-WPRE) (Terstege et al., n.d.; Vormstein-Schneider et al., 2020) (**Fig.1r**). We first validated the construct using a PV antibody in brain slices of V1 that had been targeted with the viral construct. We found 99.4% of cyRFP positive neurons also expressed PV (**Fig.1s**). Within these PV-positive neurons, we found evidence for an increased frequency and amplitude of calcium events in 4-6m *App^{NL-G-F}* mice relative to age-matched controls (**Fig.1t**). In addition, elevated PV neuron activity was associated with greater local amyloid plaque load (**Fig.S1u**). We next conducted immunofluorescence experiments labelling PV, SST and c-Fos in the visual cortex of both wild-type and *App^{NL-G-F}* mice (**Fig.1v**). Using this approach, we found increased c-Fos expression in both PV and SST neurons in 4-6m *App^{NL-G-F}* mice when compared to age-matched wild-type controls, with PV neurons exhibiting a greater increase than SST neurons (**Fig.1w-x**). Our data therefore suggest that both PV and SST neurons may show hyperactivity as a feature of amyloid-associated pathophysiology in the *App^{NL-G-F}* mouse model. This is based on evidence of increased c-Fos in both GABAergic subclasses, and elevated calcium activity measured with the mDlx construct (biased toward SST: ~22% PV and ~60% SST) and the S5E2 construct (PV specific). The relevant text and figures are shown below:

Results

The mDlx-based labelling approach targets a heterogeneous population of inhibitory neurons⁴⁵. However, our previous work³⁷ and that of others⁴⁶ found a bias toward SST neurons using this enhancer system (~22% of GCaMP-expressing neurons were PV positive and ~60% were SST positive³⁷). Therefore, to test for the involvement of other subclasses, we next conducted 2-photon imaging experiments using a PV-specific viral targeting strategy (AAV1-S5E2p-cyRFP-GSG-P2A-HIS-GCaMP6s-WPRE)^{51,52} to label PV neurons with GCaMP6s and cyRFP in V1 (**Fig.1r**). Immunofluorescence measures validated that 99.4% (505/508 cells) of cyRFP

positive neurons also expressed PV (**Fig.1s**). Using 2-photon imaging, we found evidence for an increased frequency and amplitude of calcium events in PV neurons in 4-6m App^{NL-G-F} mice relative to age-matched controls (**Fig.1t**). In addition, elevated PV neuron activity was associated with greater local amyloid plaque load (**Fig.1u**). We next conducted immunofluorescence experiments labelling PV, SST, and c-Fos in the visual cortex of WT and App^{NL-G-F} mice (**Fig.1v**). Using this approach, we found increased c-Fos expression at both PV and SST neurons in App^{NL-G-F} mice by 4-6m, with PV neurons showing a relatively greater increase than SST cells (**Fig.1w-x**). Together, our data suggest that both PV and SST neurons show hyperactivity, based on increased c-Fos in both GABAergic subclasses, and elevated calcium activity measured with the mDlx construct (biased toward SST) and the S5E2 construct (PV-specific).

Fig. 1 | Dysregulated resting-state activity at excitatory and inhibitory neurons during amyloidosis.

[...] **r**, Example region (V1) from *in vivo* 2-photon imaging showing expression of structural (cyRFP, red) and functional (GCaMP6s, green) markers in PV neurons as well as amyloid plaques via methoxy-X04 (magenta). Scale bar: 10 μ m. Traces: Example of $\Delta F/F_0$ calcium traces from PV neurons in 4-6m App^{NL-G-F} (red) and WT (black) mice during resting-state activity. Scale bars: 1.0 $\Delta F/F_0$, 10 s. **s**, Example cortical region showing immunofluorescence labelling of PV (blue) in cyRFP-expressing neurons (red) and a quantification of double-positive cells in V1 (bottom right). Scale bar: 20 μ m. **t**, Normalised (to WT) frequency (left) and amplitude (right) of calcium events in PV neurons in 4-6m App^{NL-G-F} (red) and WT (black) mice. **u**, Heatmap of normalised (to WT) frequency (top) and amplitude (bottom) values of PV neurons in regions with increasing methoxy-X04 plaque density in 4-6m App^{NL-G-F} mice. **v**, Cortical region showing immunofluorescence labelling of c-Fos (green), PV (red), and SST (cyan). Scale bar: 10 μ m. **w**, Normalised (to WT) c-Fos intensity in PV (left, red) and SST (right, cyan) neurons in 4-6m App^{NL-G-F} (red, cyan) and age-matched WT (black) mice. **x**, Normalised (to WT) c-Fos intensity in PV (red) and SST (cyan) neurons in 4-6m App^{NL-G-F} mice.

Discussion

Our data suggest that early molecular changes associated with synapse dysfunction may occur first in PV cells, but that cellular hyperactivity is likely a feature of both PV and SST interneurons during amyloid-associated pathology. Several studies have implicated PV- and SST-positive interneurons in early-stage amyloidosis²⁹. However, their involvement appears to be brain region- and timepoint-dependent, with evidence pointing to a complex interplay of potentially compensatory and pathogenic roles across disease progression^{75,101-104}.

2) Why are GABAergic bouton activity levels decreasing while soma level activity increases? It appears the axonal data 'i.e. activity' changes are related to proximity to plaques. Is this due to action potential failures in axons close to plaques, effects on presynaptic Ca²⁺ channels near plaques, other reasons (non AP-related signaling), etc.? Just saying presynaptic 'activity' is changing is inadequate. Understanding this seems critical to the author's interpretations. Another possibility is that different inhibitory axons (i.e., PV or Sst axons) signal differently (as shown previously for their somas) in an amyloidosis situation.

We thank the reviewer for their comment. As the reviewer quite rightly states, we see a complex activity profile for GABAergic boutons that varies with both time and the proximity to plaques. Boutons close to plaques exhibit hypoactivity, whilst distal boutons initially show hyperactivity at 3-4 m (Fig.3j). As such, the more distal boutons do initially map to the inhibitory cellular activity at this earlier time point but show a decrease in activity with both time and proximity to pathology (Fig.3j-k). To understand how the elevated cellular activity could map to reductions in bouton activity, we examined the event-amplitude distributions of boutons with low activity levels (AUC: z-score <0). We found a shift in the distribution of calcium-event sizes, so that small amplitude events in *App*^{NL-G-F} mice were more frequent, while medium- and high-amplitude events became more infrequent than in wild-type mice (Fig.S3e-g). Therefore, our data suggest that, while inhibitory cells show greater calcium responses in *App*^{NL-G-F} mice, these might recruit an increased frequency of small-amplitude events at downstream GABAergic boutons. As the reviewer states, one possibility is that this is due to changes in presynaptic calcium channels, which we find to be differentially expressed in both PV and SST neurons (Fig.6i). Below we include the relevant sections in the results and discussion relating to this comment.

Results

Our data suggest a complex activity profile for GABAergic boutons that varies with both age and distance to plaques. More distal GABAergic boutons initially map to elevated GABAergic cellular activity at early time points but then show a decrease in activity with both time and proximity to pathology (Fig.3i-k). To understand how the elevated cellular activity could map to reductions in bouton activity, we examined the event-amplitude distributions of boutons with low activity levels (*App*^{NL-G-F}). We found a shift in the distribution of calcium-event sizes, so that small amplitude events in *App*^{NL-G-F} mice were more frequent, while medium- and high-amplitude events became more infrequent than in WT mice (Fig.S3e-g).

Supplementary Fig. 3 | Functional signatures of dysregulated GABAergic bouton activity in *App*^{NL-G-F} mice.

[...] e-g, Fraction of small- (<0.45) (e), medium- (0.45-0.65) (f), and high- (>0.65) (g) amplitude calcium events in low-activity (AUC z-score<0) GABAergic boutons in 3-8m *App*^{NL-G-F} (red) and WT (black) mice. [...]

Discussion

Possible molecular mechanisms of synaptic and micro-circuit vulnerability

Our data suggest that, while interneuron somas show signs of hyperactivity in *App*^{NL-G-F} mice, activity levels in GABAergic boutons decrease with time and proximity to plaques. These changes in activity can be reconciled by considering bouton-level event distributions, which suggest that, while inhibitory cells show greater calcium activity in *App*^{NL-G-F} mice, these somatic events may recruit small-amplitude synaptic events at downstream GABAergic boutons. Future work may investigate the underlying mechanisms. For example, presynaptic calcium channels and/or other synaptic machinery may be directly or indirectly disrupted by amyloidosis⁴. In support of this, we found early-stage changes in molecular processes involving GABAergic synaptic transmission at PV interneurons, suggesting abnormal synaptic processing in this GABAergic subclass during early amyloidosis^{4,6}. We found DEGs linked to GABAergic synaptic transmission (*Gabra2*, *Gabrb2*, *Nf1*, *Pten*), GABA synthesis (*Gad1*, *Glu1*), synaptic vesicle cycle (*ApoE*, *Bin1*, *Snca*) and calcium signalling (*Atp2b2*, *Calm1*). Impairments in the expression and function of some of these molecules have previously been reported in mouse models and AD patient tissue, and their pharmacological modulation has shown therapeutic potential^{29,107,108}. Further work is required to investigate how the early molecular changes that we highlight link to the downregulation of GABAergic synaptic protein expression and activity *in vivo*. Such work may focus on PV and SST interneurons, as these subclasses showed the greatest number of DEGs that covaried with local amyloid load. In addition, studies using unbiased multi-omics approaches and synaptic compartment-specific analyses will be crucial to further elucidate the nature of the mechanistic processes involved¹⁰⁹. A major technical challenge, however, lies in linking *in vivo* functional recordings with molecular measures at the single-synapse level.

3) I think the authors have a fantastic opportunity to also look at the single-cell excitatory and inhibitory behavior (somatically) in the 'resilient' frontal/motor region. Perhaps global changes are stable but are coming at the cost of substantial homeostatic changes (and thus stress on the system) at the single-cell level. Thus, it would certainly increase the impact to consider these experiments in that region as well.

We thank the reviewer for their comment and have added additional experiments as suggested to address this point. First, we ran immunolabelling experiments for methoxy-X04, MOAB-2 and c-Fos in both the visual cortex and motor cortex of *App*^{NL-G-F} mice (**Fig.S1i-l**). We analysed the plaque distribution across different regions and found that the visual cortex shows higher levels of MOAB-2, methoxy-X04, and c-Fos levels compared to the motor cortex (**Fig.S1j**).

Next, as suggested by the reviewer, we conducted 2-photon GCaMP6s calcium imaging to measure the activity of excitatory and inhibitory neurons, as well as methoxy-X04 plaque density, in the motor cortex of wild-type and *App*^{NL-G-F} mice at 4-6 months. Consistent with our mesoscopic imaging, we did not find evidence for increased excitatory or inhibitory cellular activity in motor cortex at this earlier timepoint (**Fig.S1y**). Plaque density in motor cortex was also lower than equivalent timepoints in visual cortex (**Fig.S1z**). Analysis of functional assemblies found a reduction in the size of E-E (**Fig.S1aa**), but not I-E (**Fig.S1ab**) assemblies. Together, these results suggest that cellular and assembly-level inhibition is relatively intact at earlier timepoints in the motor cortex of *App*^{NL-G-F} mice, and that plaque load is lower in motor cortex in comparison to more posterior regions. The relevant figures and text are below.

Results

Using immunofluorescence measures, we also observed that posterior regions (VIS), which exhibit earlier signs of hyperactivity in *App*^{NL-G-F} mice, showed increased markers of both putative neuronal activity (c-Fos) and amyloidosis (methoxy-X04 and MOAB-2) in comparison to frontal regions (MO) (**Fig.S1j-m**).

Supplementary Fig. 1 | Dysregulated resting-state activity at excitatory and inhibitory neurons in a mouse model of amyloidosis.

[...] j, Cortical region showing immunofluorescence labelling of c-Fos (green), methoxy-X04 (magenta), and MOAB-2 (cyan) in App^{NL-G-F} mice. Scale bar: 20 μ m. k-m, Normalised (to V1) fluorescence intensity of c-Fos (k), methoxy-X04 (l), and MOAB-2 (m) in the visual (black) and motor (orange) cortices of 6-8m App^{NL-G-F} mice. [...]

Results

Our mesoscopic imaging data suggested that frontal regions, such as the motor cortex, are more resilient to hyperactivity than posterior regions (Fig.1a-g, Fig.S1a-h). To test whether activity changes are detectable at the cellular level in these frontal regions, we repeated our cellular imaging experiments by measuring the calcium activity of excitatory and inhibitory neurons, as well as methoxy-X04 plaque density, in motor cortex at 4-6 months (Fig.S1y). Consistent with our mesoscopic imaging data, activity levels in both excitatory and inhibitory cells were comparable between App^{NL-G-F} and WT mice (Fig.S1y). We also found *in vivo* plaque load to be lower in motor cortex than age-matched timepoints in visual cortex (Fig.S1z). Analysis of functional assemblies found a reduction in the size of E-E (Fig.S1aa), but not I-E (Fig.S1ab) assemblies. Together, this suggests that cellular and assembly-level inhibition is relatively intact at earlier timepoints in the motor cortex of App^{NL-G-F} in comparison to more posterior regions.

Supplementary Fig. 1 | Dysregulated resting-state activity at excitatory and inhibitory neurons in a mouse model of amyloidosis.

[...] y, Normalised (to WT) area under the curve (AUC), mean frequency (Freq), and amplitude (Amp) of calcium events at excitatory (left) and inhibitory (right) cells per mouse in 4-6m WT (black) and App^{NL-G-F} (blue, red) mice. z, Density of methoxy-X04-positive plaques in the motor (left) and visual (right) cortex of 4-6m App^{NL-G-F} mice. aa-ab, Fraction of functionally associated neurons in 4-6m App^{NL-G-F} (blue, red) and WT (black) mice for assemblies comprising excitatory neurons alone (aa, blue), and inhibitory and excitatory neurons (ab, red). [...]

For the synapse quantification, these are informative experimental designs. However, it is unclear why the authors have now chosen to lump all data from 3-8 months into these datasets. Are there correlations emerging with age here- authors mention 'earlier changes' in Fig. S2a, but no data related seems to be in the figure?

We thank the reviewer for highlighting this issue. As suggested by the reviewer, we now include a separate age-based analysis for 3-4 and 6-8 months. This shows early-stage loss of inhibitory colocalised synaptic puncta at 3-4m, whilst excitatory colocalised puncta are similar to wild-type levels (Fig. S2a). At later time-points, both excitatory and inhibitory synaptic

measures are reduced (**Fig. S2b**). The relevant figure panels are shown below and integrated into the results section:

Results

In App^{NL-G-F} mice, the normalised density of both glutamatergic and GABAergic colocalised synaptic puncta was reduced relative to WT controls (**Fig.2c-d**), with GABAergic puncta showing earlier changes (**Fig.S2a-b**).

Fig. 2 | Presynaptic GABAergic vulnerability in App^{NL-G-F} mouse cortex.

[...] **c-d**, Normalised (to WT) colocalised glutamatergic (**c**) and GABAergic (**d**) synaptic puncta in 3-8m App^{NL-G-F} (blue, red) and age-matched WT (black) mice. [...]

Supplementary Fig. 2 | Presynaptic GABAergic vulnerability in App^{NL-G-F} mouse cortex.

a-b, Normalised (to WT) colocalised excitatory (left) and inhibitory (right) synaptic puncta in App^{NL-G-F} (blue, red) and age-matched WT (black) mice at 3-4m (**a**) and 6-8m (**b**). [...]

Why would overexpression of c-fos in WT and APP mice result in differential amounts- is this truly activity-dependent? Is it surprising that c-fos activity levels can be sustained in an elevated fashion for months (i.e., in APP mice)? Wouldn't the system downregulate that signaling at some point?

We thank the reviewer for their comment. We would like to clarify that in our viral approach we used the c-Fos promoter to express EYFP protein, aiming to use fluorescence intensity as a reporter to estimate c-Fos expression levels. c-Fos is an immediate early gene, and its expression is thought to be reflective of neuronal activity across a 30–120 minute window (Barros et al., 2015; Hudson, 2018; Lara Aparicio et al., 2022). Given that it is thought to be activity dependent, it has been widely used as a putative marker of neuronal activity (Barth et al., 2004; Lara Aparicio et al., 2022). Several other papers have shown c-Fos to be elevated in mouse models of AD-related pathology (Poirier et al., 2011; Ray et al., 2024). We also find c-Fos levels to be elevated, and this is consistent with our calcium imaging data, which shows increased neuronal activity levels.

What is the threshold for a c-Fos positive neuron? Is c-Fos distributed uniformly in the neurites?

We thank the reviewer for raising this issue. We have added text to the methods to cover thresholds, as well as a figure to the supplement to show the neurite distribution.

Methods

To estimate the levels of c-Fos expression in V1 and RSC, we quantified EYFP fluorescence intensity in virally injected brains. Pyramidal neurons exhibiting c-Fos expression (defined as 10% above background) were identified and included in the analysis. Ten-pixel-thick lines were manually drawn around c-Fos-positive cells to extract EYFP fluorescence intensity profiles, which were normalised to background values (25th percentile) using custom-written MATLAB code. [...] For each cell, c-Fos expression was estimated as the average pixel intensity in each normalised trace. EYFP-expressing dendritic stretches (defined as 10% above background) were identified at various depths of the z-stacks (**Fig. 5a, right**). Ten-pixel-thick lines were then manually drawn to extract fluorescence intensity profiles, which revealed uniform c-Fos expression along the dendritic length (**Fig. S5a**).

Supplementary Fig.5 | *App^{NL-G-F}* mice show increased activity in posterior cortical regions.

a, c-Fos fluorescence normalised to dendrite mean, along relative dendrite length. [...]

The simultaneous imaging of excitatory neurons and presynaptic putative basket contacts is a very impressive experiment. However, it is unclear what functional property is being tested (relatedly, the authors rely on ‘see Methods’ far too much throughout the manuscript... tell us what is being tested in the results) and what is the relevance? One cannot infer changes in inhibitory synaptic strength from a change in Ca²⁺ amplitude alone. Also jitter of PV synapses with Ca²⁺ requires some ground truthing to see if temporal resolution is there (look at previous patch paired recordings perhaps)?

We thank the reviewer for their comment. To address these issues, we took several steps. First, we removed the temporal jitter analysis and temporal analysis of bouton dynamics. We next revised the analysis of our cellular and inhibitory bouton basket imaging data to test for inhibitory bouton activity that was spatially and temporally associated with calcium activity in proximal excitatory neurons (**Fig.5e-f**). We first correlated the inhibitory bouton activity with the level of excitatory neuronal activity (**Fig.5g**). We then used an event-triggered averaging approach to identify calcium transients in excitatory neurons and measure the average inhibitory bouton signals associated with these events (**Fig.5f,h**). Consistent with our c-Fos and VGAT data (**Fig.5c**), we found that, in *App^{NL-G-F}* animals, the activity of inhibitory boutons that were spatially and temporally associated with highly active excitatory neurons was statistically reduced (**Fig.5h**). We have combined these data with that of figure 5 and attempted to better describe the methodological steps taken in the main results. The relevant text and figure now run as follows.

Results

Presynaptic GABAergic loss and dysfunction are associated with highly active excitatory neurons

[...] We next investigated the relationship between changes in the activity of GABAergic boutons and spatially proximal excitatory neurons *in vivo*. To do this, we returned to our co-expression experiments, which used the mDlx enhancer system to label a heterogeneous population of inhibitory neurons with both a functional (GCaMP6s) and a structural (cyRFP) marker^{45,46} and a second construct to label excitatory neurons with GCaMP6s and mCherry (see Methods, Fig.1h-i). Using this approach, we were able to observe putative baskets of GABAergic boutons (expressing both cyRFP and GCaMP6s) surrounding the soma of excitatory neurons (expressing GCaMP6s and mCherry) in superficial layers of visual cortex (Fig.5e). We used *in-vivo* 2-photon calcium imaging to simultaneously image, and separate, the calcium-mediated neuronal activity of excitatory neurons and their adjacent inhibitory boutons in 6-8m WT and *App*^{NL-G-F} mice (Fig.5e-f). We first used correlation-based analysis to test the functional properties of GABAergic boutons in relation to local excitatory neuronal activity (see Methods). We found the activity of GABAergic boutons at axonal baskets was less correlated with its spatially associated excitatory neuron activity in *App*^{NL-G-F} mice compared to age-matched WT animals (Fig.5g). We next used event-triggered averaging to test for changes in GABAergic bouton activity that were spatially and temporally associated with activity levels in the adjacent excitatory neuron (see Methods, Fig.5f). We identified calcium transients in excitatory neurons and generated average inhibitory bouton signals time-locked to the transients in excitatory neurons (Fig.5f). We found that, in *App*^{NL-G-F} animals, the activity of inhibitory boutons that were spatially and temporally associated with highly active excitatory neurons was reduced relative to equivalent values in WT animals (Fig.5h). These data suggest that reductions in inhibitory bouton activity are associated with elevated neuronal activity levels in early-stage amyloidosis.

Fig. 5 | Hyperactivity is associated with GABAergic dysfunction and abnormal oscillatory activity in a mouse model of amyloidosis

[...] **e**, Top left: Example of GABAergic boutons labelled with mDlx-CyRFP (red) in proximity to a EF1a-GCaMP6s-expressing excitatory neuron (green). Top right: GCaMP6s excitatory cellular signal in isolation. Bottom left: mDlx-cyRFP structural bouton signal in isolation. Bottom right: structural bouton channel (red) merged with mDlx-GCaMP6s (green) showing bouton-specific activity (white arrowheads). Scale bar 10 μm . **f**, Example traces generated using event-triggered averaging from GCaMP6s transients in the excitatory neuron, showing an average waveform from the excitatory neuron (blue) and inhibitory boutons proximal to this cell (red). Dashed line depicts the time of peak excitatory activity. Scale bars: 0.2 $\Delta\text{F}/\text{F}_0$ and 0.5 s. **g**, Correlation of GABAergic bouton activity with local excitatory neuron activity in 6-8m *App*^{NL-G-F} (red) and WT (black) mice. **h**, Z-scored (to WT) amplitude of GABAergic boutons spatially and temporally associated with neurons of increasing frequency in 6-8m *App*^{NL-G-F} (red) and age-matched WT (black) mice. [...]

In the results and/or discussion, the authors should address the fact that GABAergic circuit dysfunction and, thus, hyperexcitability is likely a feature of *App*^{NL-G-F} at very

early time points (i.e., 1-2 months in Petrache et al. 2019 already cited) and has also been noted in very early APP/Abeta models (i.e., 3-4 weeks post expression; Goettemoeller et al. 2024; also Bushe & Kole 2012, showing hyperexcitability before extensive plaques). This is (importantly) complemented here by the authors' finding of little correlation between resting state activity and proximity to X04 staining (although perhaps not for the axonal regions).

We thank the reviewer for their point and agree that GABAergic circuit dysfunction, particularly at the level of the synapse, is likely an early-stage feature of the *App*^{NL-G-F} model. For example, we observe changes in GABAergic bouton activity and GABAergic loss at 3-4m. We also see evidence for excitatory and inhibitory cellular hyperactivity at this time. While activity levels in GABAergic boutons vary with distance to plaque, this relationship is not as clear for somatic recordings. This raises the issue that somatic recordings (Ephys and imaging) are often blind to the true distance of the total cellular morphology to amyloid proximity. This is because an axon or dendrite from a cell, which is distal to the cell body, may be impacted by amyloid. As such, for GABAergic somatic measures at the animal level, we find significant activity changes are only detectable when mice have at least moderate plaque load (**Fig.S1w**). We now include the references suggested by the reviewer in the discussion (see below).

Discussion

Nevertheless, our measures of synaptic puncta, structure, and function suggest presynaptic GABAergic vulnerability as an early feature of amyloid-associated pathophysiology in line with previous work^{95,96}. This conclusion is also supported by work in other knock-in mouse models and human *postmortem* samples, showing reductions in presynaptic density and function, as well as A β accumulation at inhibitory boutons^{32,76}.

Minor points:

-Line 79-80: there are many papers, including the 1.0 and 2.0 (KI) amyloid pathology mice, which show preferential pathophysiology (mostly presynaptic) in specific synapse types-notably PV interneurons...would recommend adding these additional citations in context(results and/or discussion)- Chen, Mody 2018 eNeuro; Sos, Mody 2020 PLOS ONE; Kumar, Rangaraju 2024 Nature Comm.) which likely help with interpretations of the data here.

We thank the reviewer for their comment and have added the suggested references to the discussion.

Discussion

Our spatial transcriptomic analyses found that, although DEGs are detectable across different inhibitory populations, some subclasses such as PV and SST interneurons show greater changes in initial disease stages and stronger putative molecular covariation with plaque-rich environments. Our data suggest that early molecular changes associated with synapse dysfunction may occur first in PV cells, but that cellular hyperactivity is likely a feature of both PV and SST interneurons during amyloid-associated pathology. Several studies have implicated PV- and SST-positive interneurons in early-stage amyloidosis²⁹. However, their involvement appears to be brain region- and timepoint-dependent, with evidence pointing to a complex interplay of potentially compensatory and pathogenic roles across disease progression^{75,101-104}. In contrast, LAMP5 and VIP cells showed lower levels of DEGs that covaried with plaque load. Interestingly, recent work has suggested that LAMP5 and VIP neurons are more active during certain brain states (e.g. locomotion), whilst PV interneuron activity better maps to resting-state synchronised oscillations⁶⁷. Thus, one possibility is that early-stage PV interneuron dysfunction may contribute to amyloid-associated hyperactivity by failing to modulate resting-state activity levels⁶⁹. In support of this, dysfunction in PV

interneurons associated with network, plasticity, and memory impairments has been widely linked to AD^{29,69}.

-Line 89: “which occurs with a functional uncoupling of E-I assemblies.” Please add a reference here; it is an important point.

We thank the reviewer for spotting this and have added a reference to this line.

Introduction

In vivo 2-photon cellular imaging revealed increased resting-state activity at excitatory (E) and inhibitory (I) neurons, with follow-up measures implicating both parvalbumin- (PV) and somatostatin-positive (SST) interneurons. These changes occur with a functional uncoupling of E-I assemblies³⁷, measured as a reduction in the number of E-E and E-I neuron pairs showing correlated activity.

-EF1a is not known to be a promoter specific to excitatory neurons, or neurons alone. It is remarkably surprising that the authors did not find overlap between cells expressing their mDlx driven and EF1a driven constructs. Fortunately, this does not appear to affect most of the claims in the paper, thus making this a minor point.

We thank the reviewer for noting this.

REFERENCES

- Barnes, S.J., Franzoni, E., Jacobsen, R.I., Erdelyi, F., Szabo, G., Clopath, C., Keller, G.B., Keck, T., 2017. Deprivation-Induced Homeostatic Spine Scaling In Vivo Is Localized to Dendritic Branches that Have Undergone Recent Spine Loss. *Neuron* 96, 871-882. <https://doi.org/10.1016/j.neuron.2017.09.052>
- Barnes, S.J., Keller, G.B., Keck, T., 2022. Homeostatic regulation through strengthening of neuronal network-correlated synaptic inputs. *eLife* 11, e81958. <https://doi.org/10.7554/eLife.81958>
- Barros, V.N., Mundim, M., Galindo, L.T., Bittencourt, S., Porcionatto, M., Mello, L.E., 2015. The pattern of c-Fos expression and its refractory period in the brain of rats and monkeys. *Front. Cell. Neurosci.* 9, 72. <https://doi.org/10.3389/fncel.2015.00072>
- Barth, A.L., Gerkin, R.C., Dean, K.L., 2004. Alteration of neuronal firing properties after in vivo experience in a FosGFP transgenic mouse. *J. Neurosci. Off. J. Soc. Neurosci.* 24, 6466–6475. <https://doi.org/10.1523/JNEUROSCI.4737-03.2004>
- Barthet, G., Mulle, C., 2020. Presynaptic failure in Alzheimer's disease. *Prog. Neurobiol.* 194, 101801. <https://doi.org/10.1016/j.pneurobio.2020.101801>
- Braak, H., Braak, E., 1995. Staging of Alzheimer's disease-related neurofibrillary changes. *Neurobiol. Aging* 16, 271–278; discussion 278-284. [https://doi.org/10.1016/0197-4580\(95\)00021-6](https://doi.org/10.1016/0197-4580(95)00021-6)
- Brasnjevic, I., Lardenoije, R., Schmitz, C., Van Der Kolk, N., Dickstein, D.L., Takahashi, H., Hof, P.R., Steinbusch, H.W.M., Rutten, B.P.F., 2013. REGION-SPECIFIC NEURON AND SYNAPSE LOSS IN THE HIPPOCAMPUS OF APPSL/PS1 KNOCK-IN MICE. *Transl. Neurosci.* 4, 8–19. <https://doi.org/10.2478/s13380-013-0111-8>
- Feng, G., Mellor, R.H., Bernstein, M., Keller-Peck, C., Nguyen, Q.T., Wallace, M., Nerbonne, J.M., Lichtman, J.W., Sanes, J.R., 2000. Imaging neuronal subsets in transgenic mice expressing multiple spectral variants of GFP. *Neuron* 28, 41–51. [https://doi.org/10.1016/s0896-6273\(00\)00084-2](https://doi.org/10.1016/s0896-6273(00)00084-2)
- Hudson, A.E., 2018. Genetic reporters of neuronal activity: c-Fos and G-CaMP6. *Methods Enzymol.* 603, 197–220. <https://doi.org/10.1016/bs.mie.2018.01.023>
- Lara Aparicio, S.Y., Laureani Fierro, Á. de J., Aranda Abreu, G.E., Toledo Cárdenas, R., García Hernández, L.I., Coria Ávila, G.A., Rojas Durán, F., Aguilar, M.E.H., Manzo Denes, J., Chi-Castañeda, L.D., Pérez Estudillo, C.A., 2022. Current Opinion on the Use of c-Fos in Neuroscience. *NeuroSci* 3, 687–702. <https://doi.org/10.3390/neurosci3040050>
- Lecy, E.E., Min, H.-K., Apgar, C.J., Maltais, D.D., Lundt, E.S., Albertson, S.M., Senjem, M.L., Schwarz, C.G., Botha, H., Graff-Radford, J., Jones, D.T., Vemuri, P., Kantarci, K., Knopman, D.S., Petersen, R.C., Jack, C.R., Lee, J., Lowe, V.J., 2024. Patterns of Early Neocortical Amyloid- β Accumulation: A PET Population-Based Study. *J. Nucl. Med.* 65, 1122–1128. <https://doi.org/10.2967/jnumed.123.267150>
- Mukherjee, S., Mez, J., Trittschuh, E.H., Saykin, A.J., Gibbons, L.E., Fardo, D.W., Wessels, M., Bauman, J., Moore, M., Choi, S.-E., Gross, A.L., Rich, J., Loudon, D.K.N., Sanders, R.E., Grabowski, T.J., Bird, T.D., McCurry, S.M., Snitz, B.E., Kambh, M.I., Lopez, O.L., De Jager, P.L., Bennett, D.A., Keene, C.D., Larson, E.B., Crane, P.K., 2020. Genetic data and cognitively defined late-onset Alzheimer's disease subgroups. *Mol. Psychiatry* 25, 2942–2951. <https://doi.org/10.1038/s41380-018-0298-8>
- Papanikolaou, A., Graykowski, D., Lee, B.I., Yang, M., Ellingford, R., Zünkler, J., Bond, S.A., Rowland, J.M., Rajani, R.M., Harris, S.S., Sharp, D.J., Busche, M.A., 2025. Selectively vulnerable deep cortical layer 5/6 fast-spiking interneurons in Alzheimer's disease models in vivo. *Neuron* 0. <https://doi.org/10.1016/j.neuron.2025.04.010>
- Petrache, A.L., Rajulawalla, A., Shi, A., Wetzels, A., Saito, T., Saido, T.C., Harvey, K., Ali, A.B., 2019. Aberrant Excitatory–Inhibitory Synaptic Mechanisms in Entorhinal Cortex Microcircuits During the Pathogenesis of Alzheimer's Disease. *Cereb. Cortex* 29, 1834–1850. <https://doi.org/10.1093/cercor/bhz016>
- Poirier, G.L., Amin, E., Good, M.A., Aggleton, J.P., 2011. Early-onset dysfunction of retrosplenial cortex precedes overt amyloid plaque formation in Tg2576 mice. *Neuroscience* 174, 71–83. <https://doi.org/10.1016/j.neuroscience.2010.11.025>

- Radulescu, C.I., Doostdar, N., Zabouri, N., Melgosa-Ecenarro, L., Wang, X., Sadeh, S., Pavlidi, P., Airey, J., Kopanitsa, M., Clopath, C., Barnes, S.J., 2023. Age-related dysregulation of homeostatic control in neuronal microcircuits. *Nat. Neurosci.* 1–13. <https://doi.org/10.1038/s41593-023-01451-z>
- Ray, A., Loghinov, I., Ravindranath, V., Barth, A.L., 2024. Early hippocampal hyperexcitability and synaptic reorganization in mouse models of amyloidosis. *iScience* 27, 110629. <https://doi.org/10.1016/j.isci.2024.110629>
- Rutten, B.P.F., Van der Kolk, N.M., Schafer, S., van Zandvoort, M.A.M.J., Bayer, T.A., Steinbusch, H.W.M., Schmitz, C., 2005. Age-Related Loss of Synaptophysin Immunoreactive Presynaptic Boutons within the Hippocampus of APP751SL, PS1M146L, and APP751SL/PS1M146L Transgenic Mice. *Am. J. Pathol.* 167, 161–173. [https://doi.org/10.1016/S0002-9440\(10\)62963-X](https://doi.org/10.1016/S0002-9440(10)62963-X)
- Sanchez-Varo, R., Sanchez-Mejias, E., Fernandez-Valenzuela, J.J., De Castro, V., Mejias-Ortega, M., Gomez-Arboledas, A., Jimenez, S., Sanchez-Mico, M.V., Trujillo-Estrada, L., Moreno-Gonzalez, I., Baglietto-Vargas, D., Vizuete, M., Davila, J.C., Vitorica, J., Gutierrez, A., 2021. Plaque-Associated Oligomeric Amyloid-Beta Drives Early Synaptotoxicity in APP/PS1 Mice Hippocampus: Ultrastructural Pathology Analysis. *Front. Neurosci.* 15, 752594. <https://doi.org/10.3389/fnins.2021.752594>
- Stephen, T.-L., Tamagnini, F., Piegsa, J., Sung, K., Harvey, J., Oliver-Evans, A., Murray, T.K., Ahmed, Z., Hutton, M.L., Randall, A., O'Neill, M.J., Jackson, J.S., 2019. Imbalance in the response of pre- and post-synaptic components to amyloidopathy. *Sci. Rep.* 9, 14837. <https://doi.org/10.1038/s41598-019-50781-1>
- Stothart, G., Alderman, S., Hermann, O., Creavin, S., Coulthard, E.J., 2025. A passive and objective measure of recognition memory in mild cognitive impairment using Fastball memory assessment. *Brain Commun.* 7, fcac279. <https://doi.org/10.1093/braincomms/fcac279>
- Terstege, D.J., Ren, Y., Ahn, B.Y., Seo, H., Adigun, K., Galea, L.A.M., Sargin, D., Epp, J.R., n.d. Impaired parvalbumin interneurons in the retrosplenial cortex as the cause of sex-dependent vulnerability in Alzheimer's disease. *Sci. Adv.* 11, eadt8976. <https://doi.org/10.1126/sciadv.adt8976>
- Trujillo-Estrada, L., Dávila, J.C., Sánchez-Mejias, E., Sánchez-Varo, R., Gomez-Arboledas, A., Vizuete, M., Vitorica, J., Gutiérrez, A., 2014. Early neuronal loss and axonal/presynaptic damage is associated with accelerated amyloid- β accumulation in A β PP/PS1 Alzheimer's disease mice subiculum. *J. Alzheimers Dis. JAD* 42, 521–541. <https://doi.org/10.3233/JAD-140495>
- Vormstein-Schneider, D., Lin, J.D., Pelkey, K.A., Chittajallu, R., Guo, B., Arias-Garcia, M.A., Allaway, K., Sakopoulos, S., Schneider, G., Stevenson, O., Vergara, J., Sharma, J., Zhang, Q., Franken, T.P., Smith, J., Ibrahim, L.A., Mastro, K.J., Sabri, E., Huang, S., Favuzzi, E., Burbridge, T., Xu, Q., Guo, L., Vogel, I., Sanchez, V., Saldi, G.A., Gorissen, B.L., Yuan, X., Zaghoul, K.A., Devinsky, O., Sabatini, B.L., Batista-Brito, R., Reynolds, J., Feng, G., Fu, Z., McBain, C.J., Fishell, G., Dimidschstein, J., 2020. Viral manipulation of functionally distinct interneurons in mice, non-human primates and humans. *Nat. Neurosci.* 23, 1629–1636. <https://doi.org/10.1038/s41593-020-0692-9>
- Wang, X., Michaelis, M.L., Michaelis, E.K., 2010. Functional Genomics of Brain Aging and Alzheimer's Disease: Focus on Selective Neuronal Vulnerability. *Curr. Genomics* 11, 618–633. <https://doi.org/10.2174/138920210793360943>
- Zarhin, D., Atsmon, R., Ruggiero, A., Baeloha, H., Shoob, S., Scharf, O., Heim, L.R., Buchbinder, N., Shinikamin, O., Shapira, I., Styr, B., Braun, G., Harel, M., Sheinin, A., Geva, N., Sela, Y., Saito, T., Saido, T., Geiger, T., Nir, Y., Ziv, Y., Slutsky, I., 2022. Disrupted neural correlates of anesthesia and sleep reveal early circuit dysfunctions in Alzheimer models. *Cell Rep.* 38, 110268. <https://doi.org/10.1016/j.celrep.2021.110268>

REVIEWERS' COMMENTS ROUND2

Reviewer #1 (Remarks to the Author):

I thank the authors for addressing my comments and questions with additional experiments, analysis and clearer language. Still, several central claims need tempering to avoid overstating causality beyond what the data support. Throughout the Abstract, Results, Discussion and Conclusions, the text implies that early synaptic pathophysiology is driven by or results from selective vulnerability of “strongly population-coupled inhibitory presynaptic inputs” and that PV interneurons are the direct locus of this effect. Because all bouton imaging comes from mDlx-labelled, SST-biased inhibitory axons (not PV-specific axons) this wording goes beyond what the data show.

In addition, the manuscript currently implies temporal sequences in the transcriptomics (PV changes “followed by” broader GABAergic changes) that cannot be inferred from cross-sectional data.

There also remain issues with regard to analysis definitions and statistical correctness. The population-coupling metric needs clearer specification because the current definition could introduce autocorrelation. Likewise, the manuscript does not explicitly state whether synaptic statistics treat boutons/spines as independent samples or whether per-animal summaries or mixed-effects models were used. If synapses are treated as independent n's, this constitutes pseudo-replication.

Minor points:

“However, more recent investigations have reported compelling evidence for GABAergic changes associated with neuronal hyperactivity in early stages of AD” - here the authors only cite themselves and a transcriptomics study but not the original studies that have shown this association more directly

“Several studies have implicated PV- and SST-positive interneurons in early-stage amyloidosis” - the authors are encouraged to cite the primary papers and not only their own review paper

Authors: We thank Reviewer #1 for their comments, we have added primary literature to the highlighted sections as suggested. As suggested, we have tried to carefully state that the functional changes occur in GABAergic boutons, whilst our spatial transcriptomics data suggest abnormal synaptic processing is occurring in PV neurons, we ensure that no language directly links these two observations as suggested by the reviewer. We have also added text to the discussion to explicitly state this point, and comment on the limitations of cross-sectional studies. The text now runs as follows:

‘Further work is required to investigate how the early molecular changes that we highlight link to the downregulation of GABAergic synaptic protein expression and critically in vivo synaptic activity. We measured in vivo synaptic activity separately and could not definitively link changes to any specific inhibitory neuronal subclass. Such work may focus on PV and SST interneurons, as these subclasses showed the greatest number of DEGs that covaried with local amyloid load. In addition, studies using unbiased multi-omics approaches and synaptic compartment-specific analyses will be crucial to further elucidate the nature of the mechanistic processes involved¹¹. A major technical challenge, however, lies in linking in vivo functional recordings with molecular measures at the single-synapse level, and understanding the evolution of such changes over time which can be challenging in cross-sectional studies.’

We have added a reference to the population coupling methods section to better explain the approach used. For many of the analyses in the paper we ran testing at the animal or region level, to test for an association with varying amyloid pathology.

Reviewer #2 (Remarks to the Author):

This manuscript by Melgosa-Ecenarro et al., has improved a lot through the revision, by adding multiple sets of new experiments. The reviewer only identified some small issues before publication.

1. In Introduction, line 71, “while early-stage amyloid-related changes have been associated with neuronal hyperactivity...”, a highly relevant paper should be cited here (Zhang et al., Front Aging Neurosci 2023, doi: 10.3389/fnagi.2023.1213379), which reviews amyloid-associated hyperactivity, synaptic loss and excitation-inhibition imbalance, etc.
2. Figure 1m,n,p,q, how was the fraction of associated neurons quantified? What was the criterion for determining whether two neurons are associated? I could not find this information in the manuscript.

Authors: We thank the reviewer for their positive comments. We have now added the suggested reference and clarified how the fraction of associated neurons was quantified. The relevant methods text runs as follows: *‘E-I neuronal assemblies were estimated using positive and significant pairwise correlations between thresholded calcium signals based on previously published approaches^{38,45,47,50,117}. For each neuron of interest, network size was estimated as the fraction of correlated neurons out of all neurons in each cortical region⁴⁵.’*

Reviewer #3 (Remarks to the Author):

This manuscript provides a compelling, technically sophisticated and comprehensive investigation into early synaptic and circuit-level dysfunction using App-KI model of amyloid pathology. Together, the data converge on a central conclusion that synaptic pathophysiology is not global, but rather selectively targets population-coupled inhibitory presynaptic boutons and excitatory dendritic spines, with PV interneurons are disproportionately affected. The population assembly data is a highly novel for the field.

The manuscript was already visually beautiful and analytically rigorous. It is now strengthened by extensive revisions and new experiments detailed in the rebuttal. The authors responded thoroughly to reviewer concerns and provided added analyses, controls, and clarifications that greatly enhance the clarity and impact of the work. This includes interpretation of mesoscopic signals, new analyses, anesthesia-related concerns were addressed with new experiments- providing increased confidence results were not artifact related to anesthesia. The rebuttal is detailed and supported by substantial new experimental additions in general, resulting in a markedly strengthened manuscript.

Authors: We thank the reviewer for their positive comments.